# Your Absorbing Discrete Diffusion Secretly Models the Conditional Distributions of Clean Data

**Jingyang Ou**[1,2], **Shen Nie**[1,2], **Kaiwen Xue**[1,2], **Fengqi Zhu**[1,2]
**Jiacheng Sun**[3], **Zhenguo Li**[3], **Chongxuan Li**[1,2]*
[1]Gaoling School of Artificial Intelligence, Renmin University of China
[2]Beijing Key Laboratory of Big Data Management and Analysis Methods, Beijing, China
[3] Huawei Noah's Ark Lab
`{oujingyang, nieshen,kaiwenxue,chongxuanli}@ruc.edu.cn;`
`fengqizhu@whu.edu.cn;{sunjiacheng1,li.zhenguo}@huawei.com;`

## Abstract

Discrete diffusion models with absorbing processes have shown promise in language modeling. The key quantities to be estimated are the ratios between the marginal probabilities of two transitive states at all timesteps, called the concrete score. In this paper, we reveal that the concrete score in absorbing diffusion can be expressed as conditional probabilities of clean data, multiplied by a time-dependent scalar in an analytic form. Motivated by this finding, we propose reparameterized absorbing discrete diffusion (RADD), a dedicated diffusion model without time-condition that characterizes the time-independent conditional probabilities. Besides its simplicity, RADD can reduce the number of function evaluations (NFEs) by caching the output of the time-independent network when the noisy sample remains unchanged in a sampling interval, which enables sampling acceleration. Built upon the new perspective of conditional distributions, we further unify absorbing discrete diffusion and any-order autoregressive models (AO-ARMs), showing that the upper bound on the negative log-likelihood for the diffusion model can be interpreted as an expected negative log-likelihood for AO-ARMs. Further, our RADD models achieve SOTA performance among diffusion models on 5 zero-shot language modeling benchmarks (measured by perplexity) at the GPT-2 scale. Our code is available at `https://github.com/ML-GSAI/RADD`.

## 1 Introduction

Auto-regressive models (Radford et al., 2018; 2019b; Brown et al., 2020) have dominated the area of language modeling for many years. In particular, such models significantly benefit from large-scale transformers (Vaswani et al., 2017a) and training data and have achieved remarkable progress (OpenAI, 2022; Achiam et al., 2023; Touvron et al., 2023; Anil et al., 2023). From a probabilistic perspective, the sequential sampling process of auto-regressive models is inefficient and limits the reasoning ability in nonsequential orders (Berglund et al., 2023; Lv et al., 2023). Intrinsically, this is because such models characterize the joint distribution by the chain rule of probability, motivating research on developing other types of generative models for text.

Diffusion models (Sohl-Dickstein et al., 2015; Ho et al., 2020; Song et al., 2021b) generate data in a coarse-to-fine manner efficiently (Song et al., 2021a; Bao et al., 2022; Zhang & Chen, 2023; Lu et al., 2022a;b) and all dimensions simultaneously, providing an appealing alternative to auto-regressive models. Among other efforts (Li et al., 2022; Austin et al., 2021; Dieleman et al., 2022; Chen et al., 2023; Graves et al., 2024; Xue et al., 2024; He et al., 2022; Campbell et al., 2022; Sun et al., 2023b; Meng et al., 2023; Lou et al., 2024) (see Section 5 for a comprehensive discussion), score entropy discrete diffusion (SEDD) (Lou et al., 2024) has shown promise in text generation. In particular, SEDD has achieved comparable results to auto-regressive models on 5 zero-shot language modeling

---

*Corresponding to Chongxuan Li.

benchmarks at the GPT-2 scale. Meanwhile, SEDD can reduce the number of function evaluations (NFEs) in sampling and fulfill text conditioned on prompts at different positions.

Technically, SEDD employs a discrete-state (absorbing) Markov process that adds noises to data by randomly replacing a token with a mask token $[\mathbf{M}]$ and then learns a reverse process to denoise from an entirely masked sentence. The key quantities to be estimated are the ratios between the marginal probabilities of two transitive states at all timesteps, called the **concrete score**. SEDD also proposes a "scaling trick" (see details in Section 3) that scales the output of the score estimation by a factor. The trick has been proven effective in practice yet not fully understood in theory (Lou et al., 2024).

One of our main contributions is to reveal that the concrete score in absorbing diffusion can be expressed as conditional probabilities of clean data, multiplied by a time-dependent scalar in an analytic form (see Theorem 1). Our finding theoretically explains the benefits of the scaling trick as a reparameterization for better optimization. Motivated by the finding, we propose reparameterized absorbing discrete diffusion (RADD), a dedicated diffusion model that characterizes the time-independent conditional probabilities by removing the time conditions from the score estimation (see Fig. 1). Besides its simplicity, RADD can significantly reduce the NFEs by caching the output of the time-independent network when the noisy sample remains unchanged during a sampling interval.

Built upon the new understanding of the concrete score, we further unify absorbing discrete diffusion and any-order autoregressive models (AO-ARMs) (Uria et al., 2014; Hoogeboom et al., 2022; Shih et al., 2022), demonstrating that their training objectives are equivalent (see Theorem 2). To establish the theory, we first rewrite the original training objective for absorbing discrete diffusion into a simpler form (named $t$-denoising cross-entropy, $t$-DCE). Then, we apply a change of variable from the time $t$ to the probability that a single-dimensional token is masked at time $t$ in the forward process. By integrating the probability variable analytically, we show its equivalence to the training objectives for AO-ARMs. These theoretical findings offer a fresh perspective that the upper bound on the negative log-likelihood of an absorbing discrete diffusion can be interpreted as the expected negative log-likelihood for corresponding AO-ARMs. Furthermore, they provide alternative objective functions for training and likelihood evaluation.

Empirically, the RADD model converges faster while achieving similar performance to the strongest baseline, i.e., SEDD (Lou et al., 2024). Moreover, we trained our RADD models on different objective functions, achieving state-of-the-art performance among diffusion models on five zero-shot language modeling benchmarks (measured by perplexity) at the GPT-2 scale. This empirical evidence validates our theoretical findings.

In summary, this paper has several contributions:

- **Deeper understanding of discrete diffusion**: Both the factorization form of the concrete score and unified training objective for absorbing discrete diffusion and AO-ARMs reveal important yet overlooked theoretical properties of absorbing discrete diffusion, which explain the mysterious scaling trick, provide practice guidance, and may inspire future work.

- **Simpler parameterization**: By removing the time conditions, we reparameterize the model to focus on a time-independent conditional probability, simplifying the existing model.

- **Efficient sampling**: Leveraging the reparameterized form, RADD can use a caching strategy to improve the sampling speed and achieve faster convergence.

- **Enhanced zero-shot language modeling performance**: Our architectural simplifications and optimized training loss lead to superior results. On five zero-shot language modeling benchmarks, RADD achieves state-of-the-art performance among discrete diffusion models (measured by perplexity) at the GPT-2 scale.

## 2 BACKGROUND

In this section, we provide an overview of continuous-time discrete diffusion models (Section 2.1) and any-order autoregressive models (Section 2.2), with a more detailed discussion available in Appendix G. For a complete list of notations and definitions, please refer to Appendix A.

## 2.1 CONTINUOUS TIME DISCRETE DIFFUSION MODEL

**Single dimension** Let $x$ denote a single dimensional sample with possible values in $\mathcal{X} = \{1, \ldots, N\}$. A continuous-time discrete Markov chain at time $t$ is characterized by a transition rate matrix $\boldsymbol{Q}_t$ as follows

$$p_{t+\Delta t|t}(\hat{x}|x) = \begin{cases} \boldsymbol{Q}_t(x, \hat{x})\Delta t + o(\Delta t), & \hat{x} \neq x, \\ 1 + \boldsymbol{Q}_t(x, x)\Delta t + o(\Delta t), & \hat{x} = x, \end{cases} \tag{2.1}$$

where $\boldsymbol{Q}_t(x, \hat{x})$ is the $(x, \hat{x})$ element of transition rate matrix $\boldsymbol{Q}_t$, denoting the transition rate from state $x$ to state $\hat{x}$ at time $t$. Equivalently, $\boldsymbol{Q}_t(x, \hat{x})$ is defined as

$$\boldsymbol{Q}_t(x, \hat{x}) = \begin{cases} \lim_{\Delta t \to 0} \frac{p_{t+\Delta t|t}(\hat{x}|x)}{\Delta t}, & \hat{x} \neq x, \\ \lim_{\Delta t \to 0} \frac{p_{t+\Delta t|t}(x|x) - 1}{\Delta t}, & \hat{x} = x. \end{cases} \tag{2.2}$$

Given the above definition, denote $\boldsymbol{P}_{t|s}(x, \hat{x}) := p_{t|s}(\hat{x}|x)$. The following Kolmogorov's forward equation holds (Campbell et al., 2022; Anderson, 2012):

$$\frac{d}{dt}\boldsymbol{P}_{t|s} = \boldsymbol{P}_{t|s}\boldsymbol{Q}_t. \tag{2.3}$$

In practice (Campbell et al., 2022), $\boldsymbol{Q}_t$ is parameterized as $\sigma(t)\boldsymbol{Q}$, where $\sigma(t)$ is a scalar function representing the noise schedule and $\boldsymbol{Q}$ is a constant matrix. In this case, the solution to Eq. (2.3) can be solved analytically as $\boldsymbol{P}_{t|s} = \exp\left((\bar{\sigma}(t) - \bar{\sigma}(s))\boldsymbol{Q}\right)$, where $\bar{\sigma}(t) = \int_0^t \sigma(s)ds$ and $\exp$ is the matrix exponential. Therefore, we can directly sample $x_t$ from $x_s$ in one step for any $t > s$.

Further, $\boldsymbol{Q}$ is often designed to diffuse towards a uniform distribution or an absorbing state $[\mathbf{M}]$. Recent work (Austin et al., 2021; Campbell et al., 2022) suggests that the absorbing matrix achieves better empirical performance. Besides, as detailed in Section 3, the specific structure of the absorbing matrix can be leveraged to improve performance and accelerate sampling. Therefore, we focus on the absorbing matrix as follows:

$$\boldsymbol{Q}^{\text{absorb}} = \begin{bmatrix} -1 & 0 & \cdots & 0 & 1 \\ 0 & -1 & \cdots & 0 & 1 \\ \vdots & \vdots & \ddots & \vdots & \vdots \\ 0 & 0 & \cdots & -1 & 1 \\ 0 & 0 & \cdots & 0 & 0 \end{bmatrix}. \tag{2.4}$$

The time reversal of the forward process is characterized by a reverse transition rate matrix $\tilde{\boldsymbol{Q}}_t$ (Sun et al., 2023a; Kelly, 1981), whose element from state $x_t$ to state $\hat{x}_t$ is given by

$$\tilde{\boldsymbol{Q}}_t(x_t, \hat{x}_t) = \begin{cases} \frac{p_t(\hat{x}_t)}{p_t(x_t)}\boldsymbol{Q}_t(\hat{x}_t, x_t), & \hat{x}_t \neq x_t, \\ -\sum_{k \neq x_t} \tilde{\boldsymbol{Q}}_t(x_t, k), & \hat{x}_t = x_t. \end{cases} \tag{2.5}$$

Simulating the reverse process requires learning the reverse transition rate $\tilde{\boldsymbol{Q}}_t(x_t, \hat{x}_t)$. As $\boldsymbol{Q}_t(\hat{x}_t, x_t)$ is known, it is sufficient to estimate the concrete score $\frac{p_t(\hat{x}_t)}{p_t(x_t)}$ by a score network $s_\theta(x_t, t) \approx \left[\frac{p_t(\hat{x}_t)}{p_t(x_t)}\right]_{\hat{x}_t \in \mathcal{X}}$ (Meng et al., 2023). Denoising score entropy (DSE) (Lou et al., 2024) is an effective objective to train the score network

$$\int_0^T \mathbb{E}_{x_t \sim p_{t|0}(x_t|x_0)} \sum_{\hat{x}_t \neq x_t} \boldsymbol{Q}_t(\hat{x}_t, x_t) \left( s_\theta(x_t, t)_{\hat{x}_t} - \frac{p_{t|0}(\hat{x}_t \mid x_0)}{p_{t|0}(x_t \mid x_0)} \log s_\theta(x_t, t)_{\hat{x}_t} + C \right) dt, \tag{2.6}$$

where the optimize irrelevant constant $C = K\left(\frac{p_{t|0}(\hat{x}_t|x_0)}{p_{t|0}(x_t|x_0)}\right)$ and $K(a) := a \log a - a$. In particular, the DSE loss in Eq. (2.6) is an upper bound of the negative log-likelihood with an unknown gap. Nevertheless, existing work (Lou et al., 2024) still employs it for training and likelihood evaluation.

After training, sampling can be understood as discretizing the following reverse process

$$\frac{d}{ds}\boldsymbol{P}_{s|t} = \boldsymbol{P}_{s|t}\tilde{\boldsymbol{Q}}_s, \tag{2.7}$$

where $ds$ is an infinitesimal negative timestep and the concrete score is replaced by the score network. Existing samplers include the Euler method and Tweedie $\tau$-leaping, as detailed in Appendix D.

**Multi-dimension** In a state space of length $d$ like $\mathcal{X}^d = \{1, \ldots, n\}^d$, we denote the sample as a sequence of one-dimensional data, i.e., $\boldsymbol{x} = x^1 \ldots x^d$. The transition matrix $\boldsymbol{Q}_t \in \mathbb{R}^{n^d \times n^d}$ has an exponential number of possible states, making it expensive to reverse. To alleviate this issue, existing work (Campbell et al., 2022; Lou et al., 2024) assumes independence between dimensions and each dimension is a one-dimensional diffusion process with the same transition rate matrix $\boldsymbol{Q}_t^{\text{tok}} \in \mathbb{R}^{n \times n}$.

Under the independent assumption, $\boldsymbol{Q}_t$ assigns zero values (Campbell et al., 2022; Lou et al., 2024) for all sequences with a Hamming distance larger than 1. Therefore, it is sufficient to model the concrete score between sequences that differ by a Hamming distance of 1, such as $\hat{\boldsymbol{x}}_t = x_t^1 \ldots \widehat{x}_t^i \ldots x_t^d$ given $\boldsymbol{x}_t = x_t^1 \cdots x_t^d$. Therefore, the score network $\boldsymbol{s}_\theta(\cdot, t) : \{1, \ldots, n\}^d \to \mathbb{R}^{d \times n}$ is defined as

$$\boldsymbol{s}_\theta\left(\boldsymbol{x}_t, t\right)_{\hat{\boldsymbol{x}}_t} = \boldsymbol{s}_\theta\left(x_t^1 \ldots x_t^i \ldots x_t^d, t\right)[i, \widehat{x}_t^i] \approx \frac{p_t\left(x_t^1 \ldots \widehat{x}_t^i \ldots x_t^d\right)}{p_t\left(x_t^1 \ldots x_t^i \ldots x_t^d\right)}, \tag{2.8}$$

which leads to the following expression to estimate the reverse transition rate matrix $\tilde{\boldsymbol{Q}}_t$:

$$\tilde{\boldsymbol{Q}}_t\left(x_t^1 \ldots x_t^i \ldots x_t^d, x_t^1 \ldots \widehat{x}_t^i \ldots x_t^d\right) = \boldsymbol{Q}_t^{\text{tok}}\left(\widehat{x}_t^i, x_t^i\right) \frac{p_t\left(x_t^1 \ldots \widehat{x}_t^i \ldots x_t^d\right)}{p_t\left(x_t^1 \ldots x_t^i \ldots x_t^d\right)} \tag{2.9}$$

$$\approx \boldsymbol{Q}_t^{\text{tok}}\left(\widehat{x}_t^i, x_t^i\right) \boldsymbol{s}_\theta\left(x_t^1 \ldots x_t^i \ldots x_t^d, t\right)[i, \widehat{x}_t^i]. \tag{2.10}$$

Existing samplers assume that each dimension is independent within a small interval $\Delta t$ and update each dimension in parallel for efficiency (Lou et al., 2024; Campbell et al., 2022).

## 2.2 ANY-ORDER AUTOREGRESSIVE MODELS

Any-order autoregressive models (AO-ARMs) (Uria et al., 2014; Hoogeboom et al., 2022; Shih et al., 2022) model the joint distribution autoregressively for all possible orders $\pi$ of the $d$ variables. Formally, they factorize the joint distribution as $\prod_{k=1}^d p(x^{\pi(k)} | x^{\pi(<k)})$. To learn such a distribution, an AO-ARM utilizes a weight-sharing neural network to model all univariate conditionals and employs mask tokens to represent absent variables. During training, the expected negative log-likelihood over the uniform distribution of all orders $U_\pi$ is minimized:

$$\mathcal{L}_{AO}(\boldsymbol{x}_0) = \mathbb{E}_{\pi \sim U_\pi} \sum_{l=1}^d -\log q_\theta(x_0^{\pi(l)} | x_0^{\pi(<l)}; \pi). \tag{2.11}$$

# 3 REPARAMETERIZED ABSORBING DISCRETE DIFFUSION

In Section 3.1, we reveal that the concrete score of absorbing discrete diffusion can be reparameterized as conditional distributions of clean data, which enables efficient sampling by caching the output of time-independent network (see Section 3.2). In Section 3.3, we unify the training objective of absorbing discrete diffusion and AO-ARMs.

## 3.1 REPARAMETERIZING THE CONCRETE SCORE AS CONDITIONAL DISTRIBUTIONS OF CLEAN DATA

A key observation is that only the transition from the masked token to an unmasked token is valid in the reverse process of an absorbing discrete diffusion. In particular, according to the definition of the transition matrix of the absorbing process (see Eq. (2.4)), we have $\boldsymbol{Q}^{\text{absorb}}(\hat{x}_t^i, x_t^i) = 0$ for any unmasked $x_t^i \neq [\mathbf{M}]$ and $\hat{x}_t^i \neq x_t^i$. Therefore, the corresponding element in the transition matrix of the reverse process $\tilde{\boldsymbol{Q}}_t$ (see Eq. (2.5)) equals zero. Namely,

$$\tilde{\boldsymbol{Q}}_t\left(x_t^1 \ldots x_t^i \ldots x_t^d, x_t^1 \ldots \widehat{x}_t^i \ldots x_t^d\right) = \sigma(t)\boldsymbol{Q}^{\text{absorb}}\left(\widehat{x}_t^i, x_t^i\right) \frac{p_t\left(x_t^1 \ldots \widehat{x}_t^i \ldots x_t^d\right)}{p_t\left(x_t^1 \ldots x_t^i \ldots x_t^d\right)} = 0, \tag{3.1}$$

for any unmasked state $x_t^i \neq [\mathbf{M}]$ and $\hat{x}_t^i \neq x_t^i$ and it is unnecessary to model the corresponding concrete score $\frac{p_t(x_t^1 \ldots \widehat{x}_t^i \ldots x_t^d)}{p_t(x_t^1 \ldots x_t^i \ldots x_t^d)}$. Also, note that the concrete score always takes the value of one if $\hat{x}_t^i = x_t^i$. Therefore, we only need to characterize the concrete score for $x_t^i = [\mathbf{M}]$ and $\hat{x}_t^i \neq [\mathbf{M}]$.

Interestingly, in this case, we discover that the concrete score has a simple analytic form w.r.t. to the conditional distributions of clean data, as summarized in the following Theorem 1.

**Theorem 1.** *(Analytic concrete score in absorbing diffusion, proof in Appendix B) For $\boldsymbol{x}_t = x_t^1 \ldots x_t^i \ldots x_t^d$ and $\hat{\boldsymbol{x}}_t = x_t^1 \ldots \hat{x}_t^i \ldots x_t^d$, if $x_t^i = [\boldsymbol{M}]$ and $\hat{x}_t^i \neq [\boldsymbol{M}]$, the concrete score at time $t$ can be expressed as a time-independent conditional distribution at time zero multiplied by an analytic time-dependent term:*

$$\underbrace{\frac{p_t\left(x_t^1 \ldots \hat{x}_t^i \ldots x_t^d\right)}{p_t\left(x_t^1 \ldots x_t^i \ldots x_t^d\right)}}_{concrete\ score} = \underbrace{\frac{e^{-\bar{\sigma}(t)}}{1 - e^{-\bar{\sigma}(t)}}}_{scalar} \underbrace{p_0(\hat{x}_t^i | \boldsymbol{x}_t^{UM})}_{\substack{clean\ data \\ distribution}}$$

*where $\boldsymbol{x}_t^{UM}$ is the vector consists of all unmasked tokens of $\boldsymbol{x}_t$.*

One immediate implication of Theorem 1 is to theoretically explain the benefit of the "scaling trick" in existing work (Lou et al., 2024) (see Appendix C.2 therein), which significantly improves the practical performance of discrete diffusion (see Table 1) but has not been fully understood before. In particular, the scaling trick divides the output of the score network by a factor. Equivalently, it reparameterizes $\boldsymbol{s}_\theta(\boldsymbol{x}_t, t)$ as follows:

$$\boldsymbol{s}_\theta(\boldsymbol{x}_t, t) = \frac{e^{-\bar{\sigma}(t)}}{1 - e^{-\bar{\sigma}(t)}} \tilde{\boldsymbol{s}}_\theta(\boldsymbol{x}_t, t), \tag{3.2}$$

where $\tilde{\boldsymbol{s}}_\theta(\boldsymbol{x}_t, t)$ is the output of the reparameterized score network and the scaling factor coincides with the time-dependent term in Theorem 1. In the original parameterization, the score network $s_\theta$ must model the whole time-dependent concrete score. In contrast, with the scaling trick, the reparameterized score $\tilde{\boldsymbol{s}}_\theta(\boldsymbol{x}_t, t)$ can focus on capturing the clean data distribution $p_0(\hat{x}^i | \boldsymbol{x}_t^{UM})$ and simplifies learning, according to Theorem 1.

Further, Theorem 1 suggests that the reparameterized score is essentially a conditional probability on clean data, which is time-independent. Motivated by this, we propose reparameterized absorbing discrete diffusion (RADD), which employs a time-independent network $\boldsymbol{c}_\theta(\boldsymbol{x}_t)$ that defines a model distribution $q_\theta$ by corresponding conditional distributions to approximate data distribution $p_0$ directly:

$$\boldsymbol{c}_\theta(\boldsymbol{x}_t)[i, \hat{x}_t^i] = q_\theta(\hat{x}_t^i | \boldsymbol{x}_t^{UM}) \approx p_0(\hat{x}_t^i | \boldsymbol{x}_t^{UM}). \tag{3.3}$$

In practice, we make a minimal modification of the score network in SEDD (Lou et al., 2024) for simplicity and fairness as shown in Fig. 1. Specifically, we remove the time-conditioning input, reducing the architecture to a form similar to the standard GPT model, with softmax as the final activation. Further details can be found in Appendix J.1.

Our reparameterization approach, which removes $t$, applies to both score and mean parameterizations, as detailed in Appendix E. This not only simplifies the training target but also enables a more efficient sampling process than SEDD (Lou et al., 2024), as presented below.

### 3.2 EFFICIENT SAMPLERS TO REDUCE NFEs BY CACHING THE OUTPUT OF RADD

In the reverse process of an absorbing discrete diffusion, once a token transitions from $[\mathbf{M}]$ to an unmasked token, it remains unchanged. Consequently, for a sequence consisting of $d$ tokens, there will be at most $d$ intervals during the sampling process where changes occur, regardless of the number of sampling steps $D$. In the remaining steps, the sequence remains unchanged across all $d$ dimensions. This property allows us to cache $\boldsymbol{c}_\theta(x_t)$ to avoid the need to reevaluate the time-independent $\boldsymbol{c}_\theta$ when $\boldsymbol{x}_t$ is unchanged in the previous step (see Appendix I for the pseudo-code). However, since SEDD is conditioned on time, it does not support this caching strategy for reducing NFEs.

The NFEs with the caching strategy is a random variable. To quantify it, we calculate the expected NFEs (E-NFEs) in analytic form, conditioned on the sampling method, time steps, and noise schedule. For instance, using the Tweedie $\tau$-leaping method with a log-linear noise schedule (Lou et al., 2024), the E-NFEs can be expressed by sampling steps $n$ and generating length $l$ [1] (proof in Appendix D.5):

$$\text{E-NFEs}(n) = n(1 - (1 - \frac{1}{n})^l), \tag{3.4}$$

---

[1] A similar analysis was presented in Chen et al. (2024) for their discrete-time sampler, although it is based on different assumptions and applicable to different scenarios. A detailed comparison is provided in Appendix F.

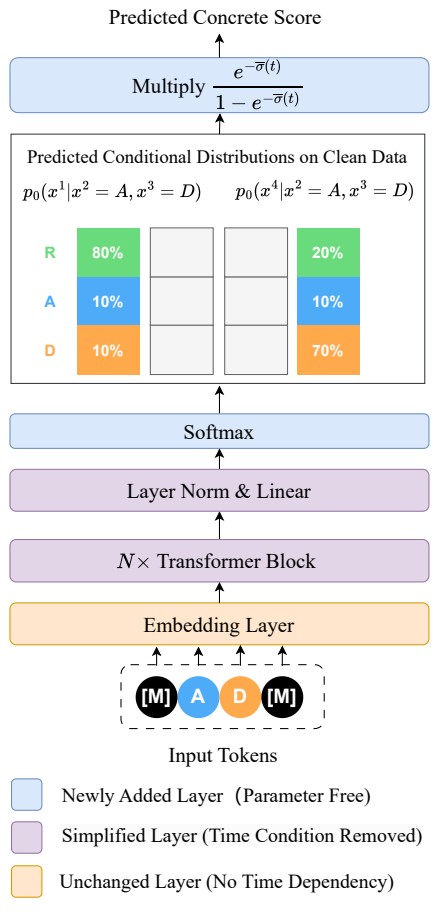

Figure 1: **Reparameterized network architecture vs. SEDD (DiT).** Our network simplifies the original DiT by removing time conditions and outputs the conditional distributions on clean data, similar to the standard Transformer. For example, with a vocabulary of {R, A, D}, only the probabilities in the columns corresponding to the [**M**] token are meaningful (highlighted in color), since the network is designed to learn to denoise the masked input. The remaining output, shown in grey, will not be utilized.

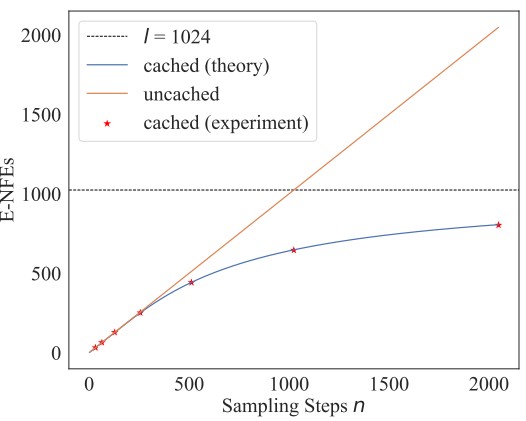

(a) **Expected number of function evaluations (E-NFEs) over different sampling steps.** E-NFEs measured by Tweedie $\tau$-leaping method with log-linear noise schedule.

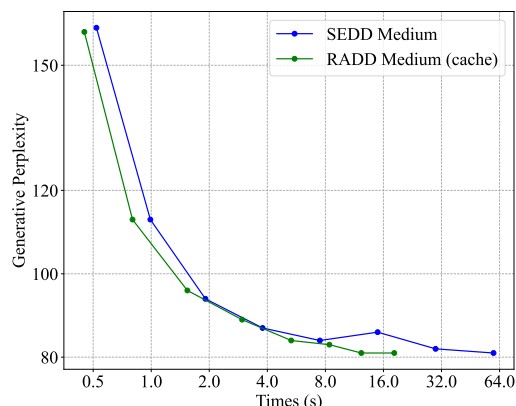

(b) **Sample quality measured by perplexity ($\downarrow$).** Our RADD model, utilizing the cache strategy, achieves slightly better perplexity with less time, enabling faster convergence and reduced sampling time.

In Fig. 1a, we plot the curve of Eq. (3.4) in blue, which aligns well with our experiments (red stars). This demonstrates that our method theoretically reduces E-NFEs, particularly at larger sampling steps. This reduction is also supported by Fig. 1b, where RADD shows faster convergence trends.

Furthermore, based on Theorem 1, simplified forms of the reverse process for both Euler method and Tweedie $\tau$-leaping method can be derived, which leads to corresponding analytic forms of E-NFEs given time steps and noise schedule. We also prove that these two sampling methods are equivalent under a log-linear noise schedule for absorbing discrete diffusion (see Appendix D for more details).

### 3.3 UNIFYING ABSORBING DISCRETE DIFFUSION AND ANY-ORDER AUTOREGRESSIVE MODEL

Building upon Theorem 1, we further prove the equivalence between absorbing discrete diffusion and any-order autoregressive models introduced in Section 2.2, as presented in the following theorem.

**Theorem 2.** *The absorbing discrete diffusion objective of Eq. (2.6) is equivalent to any-order autoregressive objective of Eq. (2.11) when the final total noise level $\bar{\sigma}(T) \to +\infty$.*

The proof of Theorem 2 consists of three key steps, which introduce three different yet equivalent loss functions. Below we briefly present the key ideas and defer the proof in Appendix C.

In the first step, by removing the terms $s_\theta(\boldsymbol{x}_t, t)_{\hat{\boldsymbol{x}}_t}$ and $K\left(\frac{p_{t|0}(\hat{\boldsymbol{x}}_t|\boldsymbol{x}_0)}{p_{t|0}(\boldsymbol{x}_t|\boldsymbol{x}_0)}\right)$ in Eq. (2.6), we can define a simpler loss $\mathcal{L}_{t\text{-DCE}}^T$ called $t$-denoising cross-entropy loss (abbr. $t$-DCE), which is equivalent to DSE loss. In the multi-dimensional case, it has the form:

$$\mathcal{L}_{t\text{-DCE}}^T(\boldsymbol{x}_0) = \int_0^T \mathbb{E}_{\boldsymbol{x}_t \sim p_{t|0}(\boldsymbol{x}_t|\boldsymbol{x}_0)} \left[ \sum_{x_t^i=[\mathbf{M}]} -\frac{\sigma(t)e^{-\bar{\sigma}(t)}}{1-e^{-\bar{\sigma}(t)}} \log\left(\frac{e^{-\bar{\sigma}(t)}}{1-e^{-\bar{\sigma}(t)}} q_\theta(x_0^i|\boldsymbol{x}_t^{\text{UM}})\right) \right] dt \quad (3.5)$$

We emphasize that Eq. (3.5) holds in a nonparametric setting because RADD can be interpreted as a model distribution $q_\theta$ representing the conditional distribution of clean data, which approximates the true distribution $p_0$, as proven in Theorem 1.

In the second step, inspired by Kingma et al. (2021), we change the variable from $t$ to $\lambda(t) = 1 - e^{-\bar{\sigma}(t)}$, which represents the probability of a token being masked in $[0, t]$ during the forward process. Thus, $\mathcal{L}_{t\text{-DCE}}^T(\boldsymbol{x}_0)$ can be rewritten as an integral of $\lambda$, defined as $\lambda$-denoising cross-entropy loss (abbr. $\lambda$-DCE):

$$\mathcal{L}_{\lambda\text{-DCE}}(\boldsymbol{x}_0) := \int_0^1 \frac{1}{\lambda} \mathbb{E}_{\boldsymbol{x}_\lambda \sim p_\lambda(\boldsymbol{x}_\lambda|\boldsymbol{x}_0)} \left[ \sum_{x_\lambda^i=[\mathbf{M}]} -\log q_\theta(x_0^i|\boldsymbol{x}_\lambda^{\text{UM}}) \right] d\lambda, \quad (3.6)$$

where $p_\lambda(\boldsymbol{x}_\lambda|\boldsymbol{x}_0)$ is the joint distribution induced by masking each dimension in $\boldsymbol{x}_0$ independently with a probability $\lambda$.

Finally, we prove that $\lambda$-DCE loss in Eq. (3.6) can be integrated analytically and rewritten as $\mathcal{L}_{AO}$ in Eq. (2.11). We summarize our proof procedure by the equivalence between these losses:

$$\mathcal{L}_{\text{DSE}}^T(\boldsymbol{x}_0) \overset{\text{Appendix } C.1}{\Longleftrightarrow} \mathcal{L}_{t\text{-DCE}}^T(\boldsymbol{x}_0) \overset{\text{Appendix } C.2}{\Longleftrightarrow} \mathcal{L}_{\lambda\text{-DCE}}(\boldsymbol{x}_0) \overset{\text{Appendix } C.3}{\Longleftrightarrow} \mathcal{L}_{AO}(\boldsymbol{x}_0). \quad (3.7)$$

A direct benefit from Theorem 2 is that we can use an absorbing discrete diffusion model to sample like AO-ARM and vice versa. For training and likelihood evaluation, the four losses in Eq. (3.7) can also be used (see Appendix I for pseudo-code). To efficiently estimate the four losses using Monte Carlo methods, we can replace the sum or integral with an expectation. Take Eq. (3.6) for example, it can be rewritten as the following form of expectation on $\lambda$:

$$\mathcal{L}_{\lambda\text{-DCE}}(\boldsymbol{x}_0) = \mathbb{E}_{\lambda \sim U([0,1])} \frac{1}{\lambda} \mathbb{E}_{\boldsymbol{x}_\lambda \sim p_\lambda(\boldsymbol{x}_\lambda|\boldsymbol{x}_0)} \left[ \sum_{x_\lambda^i=[\mathbf{M}]} -\log q_\theta(\boldsymbol{x}_0^i|\boldsymbol{x}_\lambda^{\text{UM}}) \right]. \quad (3.8)$$

Additionally, Theorem 2 provides a new perspective on the DSE loss. While it has been traditionally viewed as an upper bound on the negative log-likelihood for the diffusion model, it can also be interpreted as an expected negative log-likelihood over factorial numbers of orderings for AO-ARM by Eq. (2.11). As discussed in (Uria et al., 2014), for different orders $\pi$, $q_\theta(x_0; \pi)$ will be inconsistent in general. Despite this inconsistency, it can be viewed as an ensemble of multiple autoregressive models with different orders, potentially more robust than fixed-order models.

We mention that Hoogeboom et al. (2022) establish the equivalence between ARMs and the ELBO of absorbing diffusion models. In comparison, built upon our Theorem 1, we extend existing work by unifying four loss functions in Eq. (3.7), deepening the understanding of absorbing discrete diffusion. We provide a detailed discussion in Appendix H and a systematic empirical study in Section 4.3.

## 4 EXPERIMENTS

We present the experimental setups in Section 4.1. We then evaluate the performance of accelerated generation in Section 4.2 and zero-shot perplexity on various language datasets in Section 4.3.

## 4.1 SETTINGS

Below, we briefly present the experimental settings. For more details, please see Appendix J.

**Model.** We use RADD model $c_\theta$ reparameterzied as described in Section 3.1. Compared with SEDD models, RADD models have fewer parameters because the time-condition has been eliminated. We trained our RADD model $c_\theta$ using denoising score entropy, $t$-denoising cross-entropy, $\lambda$-denoising cross-entropy and any-order autoregressive loss, abbreviated as RADD-DSE, RADD-$t$-DCE, RADD-$\lambda$-DCE and RADD-AO, since all these models share the same architecture as described in Fig. 1. For the SEDD small and medium model, we employed their pre-trained model. When performing text generation tasks, we used RADD-$\lambda$-DCE medium models.

**Data.** Following SEDD, we trained on the OpenWebText (Gokaslan & Cohen, 2019) dataset and tested on the LAMBADA, WikiText2, PTB, WikiText103, and One Billion Words datasets (Paperno et al., 2016; Merity et al., 2016; Marcus et al., 1993; Chelba et al., 2013). For data splits and data processing, we adopted the same settings and techniques as SEDD, which involves packing sentences to generate uniform-length blocks as model input.

**Training setup.** We used a log-linear noise schedule where the expectation of the number of changed tokens at time $t$ is linear with $t$. Following SEDD, we report results for RADD trained over 400K iterations in Tables 1 and 2. To further analyze the convergence and performance trends, we extended the training of RADD-small to 1,000K iterations, detailed in Table 7 of the appendix.

**Metric.** Following Lou et al. (2024), we conduct experiments on unconditional generation and language modeling tasks. For language modeling tasks, we report the perplexity calculated on the dataset with different models. For generation, we assess sample quality using perplexity (PPL) on unconditional samples measured by an additional larger language model (i.e., GPT-2 large), and sample diversity through unigram entropy (Strudel et al., 2022).

## 4.2 EFFICIENT SAMPLING

As shown in Fig.1 of Lou et al. (2024), SEDD surpasses AR in terms of sampling speed. Therefore, we compare the sample quality between SEDD and our RADD model measured by perplexity. As shown in Fig. 1b, RADD with the caching strategy is more efficient than SEDD. This improvement is expected because the NFEs is limited by the generating sequence length. We further conducted batch size ablation and compare the running time and unigram entropy as detailed in Appendix J.4.

As discussed in Section 3.3, we can also use RADD as an any-order autoregressive model to generate samples in different orders, as detailed in Appendix J.4. We present more sampling details in Appendix J.3. and the generated samples in Appendix K.1.

## 4.3 IMPROVED ZERO-SHOT PERPLEXITY ON LANGUAGE MODELING

Following SEDD, we present zero-shot perplexities on the LAMBADA, WikiText2, PTB, Wiki-Text103, and 1 Billion Words datasets (Radford et al., 2019a) in Tables 1 and 2 and compare the zero-shot perplexity of our model with other baseline models (Austin et al., 2021; Gulrajani & Hashimoto, 2023; Lou et al., 2024). Perplexities of RADD models are calculated based on their corresponding loss (e.g., RADD-$\lambda$-DCE on $\mathcal{L}_{\lambda\text{-DCE}}$), which is valid for likelihood estimation as discussed in Section 3.3.

Firstly, we conduct an ablation study of the scaling trick in the middle of the Tables 1 and 2. For the absorbing diffusion, the perplexity of the scaled version of SEDD outperforms its unscaled version, which matches our theoretical discovery in Theorem 1. Secondly, under the same DSE loss and similar parameter counts, we observed that the RADD-DSE model without time-conditioning outperforms the SEDD-Scale model with time-conditioning. This ablation validates our analysis in Section 3.1, indicating that time-conditioning is unnecessary for absorbing discrete diffusion models. Additionally, while RADD models trained with four equivalent loss functions achieve similar performance, minor discrepancies persist. These differences stem from variations in gradient estimation on finite data, leading models to converge at distinct local optima despite the theoretical

Table 1: **Zero-shot language modeling perplexity (↓) on five datasets using *small* models.** "SEDD-Unscale" and "SEDD-Scale" refer to the unscaled and scaled versions of the absorbing models, respectively. All SEDD and RADD models are trained for **400k** iterations. Results for other diffusion models are based on the upper bound from (Austin et al., 2021; Gulrajani & Hashimoto, 2023; Lou et al., 2024). For RADD models, the results are calculated based on the corresponding loss.

| Method | LAMBADA | WikiText2 | PTB | WikiText103 | 1BW |
|---|---|---|---|---|---|
| GPT-2 | **45.04** | 42.43 | 138.43 | 41.60 | 75.20 |
| D3PM | 93.47 | 77.28 | 200.82 | 75.16 | 138.92 |
| PLAID | 57.28 | 51.80 | 142.60 | 50.86 | 91.12 |
| SEDD-Uniform | 65.40 | 50.27 | 140.12 | 49.60 | 101.37 |
| SEDD-Unscale | 52.21 | 44.75 | 130.49 | 43.14 | 80.70 |
| SEDD-Scale | 50.92 | 41.84 | 114.24 | 40.62 | 79.29 |
| RADD-DSE | 49.57 | 38.83 | 111.74 | 37.46 | **72.35** |
| RADD-$t$-DCE | 50.56 | 39.02 | 109.03 | 36.38 | 72.60 |
| RADD-$\lambda$-DCE | 51.70 | 39.98 | **107.85** | 37.98 | 72.99 |
| RADD-AO | 50.27 | **38.26** | 110.38 | **35.90** | 74.28 |

Table 2: **Zero-shot language modeling perplexity (↓) on five datasets using *medium* models.** "SEDD-Unscale" and "SEDD-Scale" refer to the unscaled and scaled versions of the absorbing models, respectively. All SEDD and RADD models are trained for **400k** iterations.

| Method | LAMBADA | WikiText2 | PTB | WikiText103 | 1BW |
|---|---|---|---|---|---|
| GPT-2 | **35.66** | 31.80 | 123.14 | 31.39 | **55.72** |
| SEDD-Unscale | 44.60 | 34.85 | 93.26 | 32.97 | 67.91 |
| SEDD-Scale | 42.77 | 31.04 | 87.12 | 29.98 | 61.19 |
| RADD-DSE | 42.30 | **29.17** | **75.16** | **28.03** | 57.45 |
| RADD-$t$-DCE | 43.24 | 30.19 | 78.77 | 29.36 | 57.95 |
| RADD-$\lambda$-DCE | 44.10 | 30.60 | 82.08 | 29.29 | 60.32 |
| RADD-AO | 41.96 | 29.96 | 79.06 | 28.51 | 57.07 |

equivalence of their objectives on expectation. Overall, all RADD losses outperform SEDD on average across the five datasets, validating our analysis in Sections 3.1 and 3.3.

## 5 RELATED WORK

**Continouous-state diffusion models for text generation.** Several works have been proposed to apply continuous diffusion to text (Li et al., 2022; Dieleman et al., 2022; Chen et al., 2023; Graves et al., 2024). Li et al. (2022) use an embedding layer to map discrete tokens to a latent space and learn a continuous-state diffusion on it. Bit Diffusion (Chen et al., 2023) learns a continuous diffusion model to generate binary bits of discrete tokens. However, transforming between these continuous representations and discrete tokens by thresholding may lose information. Bayesian Flow Network (Graves et al., 2024) achieves competitive log-likelihood on character-level language modeling tasks and is proven equivalent to continuous stochastic differential equations trained by denoising score matching (Xue et al., 2024). Such models underperform auto-regressive models on standard text generation tasks.

**Discrete-state diffusion models for text generation.** Several discrete-state diffusion models have been proposed (Sohl-Dickstein et al., 2015; Hoogeboom et al., 2021; Austin et al., 2021). D3PM (Austin et al., 2021) proposed a diffusion framework based on any probability transition matrix and trained with a lower bound of log-likelihood. DiffusionBERT (He et al., 2022) utilizes a pre-trained BERT (Devlin et al., 2019b) as an initialization of diffusion. Furthermore, Campbell et al.

(2022) generalizes the framework to continuous time by introducing a rate matrix. It is difficult to apply the score matching in such models because the gradient of the data distribution is undefined. Several works try to generalize the score matching on discrete data (Lou et al., 2024; Meng et al., 2023; Campbell et al., 2022; Sun et al., 2023b; Campbell et al., 2024). Meng et al. (2023) introduce the concrete score and the denoising concrete score matching loss. Furthermore, SEDD bridges the discrete state diffusion and the concrete score by introducing a denoising score entropy loss (Lou et al., 2024). By incorporating an absorbing process, SEDD achieves competitive performance with the auto-regressive models, especially, GPT-2. Motivated by the success of absorbing discrete diffusion, RADD is specifically designed for this class of models. Due to fundamental differences in score formulations, it can not directly apply to other models like multinomial diffusion. Campbell et al. (2024) proposed discrete flow models. Chen et al. (2024) proposed a discrete non-Markov diffusion model to accelerate sampling, which has some connections to our cache-based acceleration in Section 3.2. A detailed comparison can be found in Appendix F.

**Concurrent works**   We mention that Shi et al. (2024) and Sahoo et al. (2024) independently conducted related studies on absorbing discrete diffusion. We provide a detailed discussion here.

Shi et al. (2024) derived a weighted integral of cross-entropy loss in their Eq.(5) similar to our $t$-DCE loss in Eq. (3.5). Besides, their Proposition 1, which connects the score parameterization and the mean parameterization[2], also resembles our Theorem 1. In comparison, we simplified the conditional expectation term (related to $t$) in Proposition 1 (Shi et al., 2024) to a time-independent conditional probability at time zero. Motivated by the finding, we proposed a simpler parameterization that enables fast sampling. It is worth noting that the hyperparameters they selected significantly contribute to the model's performance, which also applies to our RADD models (see Appendix J.2 for details). In addition, Shi et al. (2024) proposed a generalized masked diffusion model allowing state-dependent masking schedules.

Sahoo et al. (2024) derive the same cross-entropy losses with Shi et al. (2024). Despite lacking a theoretical foundation, they conducted time-conditioning ablation which shows that time-conditioning has minimal impact on perplexity. Notably, their method for removing time conditioning retained the same network structure (e.g., keeping adaptive layer normalization) while setting the time input to zero uniformly. In contrast, our approach removes the network structure related to time inputs entirely, eliminating the need for time input and thereby simplifying the network design. They also proposed a caching strategy to accelerate sampling. While this coincides with our work in Section 3.2, we present a complete theoretical analysis of E-NFEs to quantify the acceleration efficiency.

**Our unique contribution lies in the decomposition of the concrete score and time-independent parameterization**, serveing as the foundation for subsequent contributions in Sections 3.2 and 3.3.

## 6   Conclusion

We introduce RADD, a dedicated discrete diffusion model that characterizes the time-independent conditional probabilities, built upon a new factorization form of the concrete score. RADD is more efficient by reducing the NFEs with a cache strategy while maintaining comparable performance to strong baselines. Additionally, we demonstrated the unification of training objectives for absorbing discrete diffusion and AO-ARMs. On five zero-shot language modeling benchmarks, our RADD models achieve state-of-the-art performance at the GPT-2 scale.

**Limitaition.**   Our model has been trained and evaluated primarily on the GPT-2 scale. For broader applicability, it is essential to explore the effects of scaling on the performance (Hoffmann et al., 2022), which is left as future work. The success of diffusion transformers on images (Bao et al., 2023a; Peebles & Xie, 2023a; Bao et al., 2023b) and videos (Bao et al., 2024) suggests that diffusion models can be scaled up by incorporating transformers.

Another limitation is that our model can only generate full-length outputs, unlike auto-regressive models that can produce variable-length outputs. This restricts the flexibility of our model in certain applications. We leave the investigation on this issue as future work.

---

[2]Our conclusions are based on score parameterization but can be extended to mean prediction parameterization (please see Appendix E).

## ACKNOWLEDGMENTS

This work was supported by the National Natural Science Foundation of China (No. 92470118); the Beijing Nova Program (No. 20220484044); Beijing Natural Science Foundation (No. L247030).

We thank Aaron Lou for his prompt and detailed responses to our inquiries, which greatly assisted our research. We also thank Zebin You for his support in setting up the coding environment.

**Ethics statement.** For the current theoretical and experimental scope of this paper, we have not found any direct social impacts. However, considering future developments, the paper potentially contributes to the next-generation large language models. In this context, this work could significantly reduce the inference cost of language models but may also lead to hallucinations, amplify biases and discrimination in the data, and pose risks of misuse. As with other generative models, addressing these issues requires further advancements in the field.

**Reproducibility statement** We have open-sourced our code in `https://github.com/ML-GSAI/RADD`. For detailed instructions on environment setup and running scripts, please refer to the `README.md` file. Comprehensive explanations and proofs of our theoretical claims can be found in Appendices B and C.

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

CONTENTS

## A  DETAILED NOTATIONS AND DEFINITIONS

We introduce the notations used throughout the paper. Let lower, boldface lower and upper case letters represent scalers (e.g., $a$), vectors (e.g., $\boldsymbol{a}$), and matrices (e.g., $\boldsymbol{A}$), respectively. For a vector $\boldsymbol{a}$, $a^i$ denotes its $i$-th element. For a matrix $\boldsymbol{A}$, $\boldsymbol{A}(i,j)$ denotes $(i,j)$-th element. For a vector function $\boldsymbol{f}$, $\boldsymbol{f}(\boldsymbol{x})_i$ denotes the $i$-th element of $\boldsymbol{f}(\boldsymbol{x})$. Constants and random variables are not distinguished in the notation if there is no confusion. We represent the distributions of the forward and reverse processes by $p$ and $q_\theta$ respectively. The transition probability from time $s$ to time $t$ is denoted by $p_{t|s}(\cdot|\cdot)$, and the probability at time $t$ is denoted by $p_t(\cdot)$. Complete notations and definitions are listed below:

- $x, \hat{x}$: Scalar variables representing states in a model.
- $\mathcal{X}$: A one-dimensional sample space $\{1, \cdots, N\}$.
- $\boldsymbol{Q}_t$: The transition rate matrix at time $t$.
- $p$: The probability of the forward process defined by the transition rate matrix $\boldsymbol{Q}_t$.
- $q_\theta$: The probability of the reverse process defined by model $\boldsymbol{c}_\theta$.
- $p_{t|s}(\hat{x}|x)$: The transition probability from state $x$ to state $\hat{x}$ from time $s$ to time $t$.
- $p_t(x)$: The probability of $x$ at time $t$.
- $\boldsymbol{P}_{t|s}$: The transition probability matrix from time $s$ to time $t$.
- $\sigma(t)$: The noise schedule function.
- $\tilde{\boldsymbol{Q}}_t$: The reverse transition rate matrix at time $t$.
- $\boldsymbol{s}_\theta(\boldsymbol{x}_t, t)_{\hat{\boldsymbol{x}}_t}$: The corresponding element of $\boldsymbol{s}_\theta(\boldsymbol{x}_t, t)$, which approximates $\frac{p_t(\hat{\boldsymbol{x}}_t)}{p_t(\boldsymbol{x})}$.
- $[\mathbf{M}]$: A special mask token in the absorbing process.
- $\mathcal{X}^d$: A multi-dimensional sample space $\{1, \cdots, N\}^d$.
- $\boldsymbol{x}_t$: A multi-dimensional vector.
- $x_t^i$: The $i$-th element of $\boldsymbol{x}_t$.
- $p_{s|t}^i(\cdot|\boldsymbol{x}_t)$: The probability on dimension $i$ from time $s$ to time $t$ conditioned on full vector $\boldsymbol{x}_t$.
- $p_{s|t}^{\text{tweedie}}(\cdot|\cdot)$: The transition probability from time $s$ to time $t$ under Tweedie $\tau$-leaping method.
- $p_{s|t}^{\text{euler}}(\cdot|\cdot)$: The transition probability from time $s$ to time $t$ under the Euler method.
- $\boldsymbol{Q}_t^{\text{tok}}$: Transition rate matrix for each dimension of $\boldsymbol{x}_t$.
- $\boldsymbol{x}^{\text{UM}}$: vector consists of all unmasked tokens of $\boldsymbol{x}$.

- $\boldsymbol{x}^{a:b}$: The elements of $\boldsymbol{x}$ with indices ranging from $a$ to $b$.

- $\boldsymbol{c}_\theta(\boldsymbol{x}_t)$: A network that characterizes the time-independent conditional probabilities in reparameterized absorbing discrete diffusion (RADD).

- $d$: Total sequence length or dimension of $\boldsymbol{x}$.

- $l$: Generating sequence length.

- $\pi$: one permutation, $\pi(l)$ denotes the $l$-th element of permutation $\pi$, $\pi(< l)$ denotes the elements of permutation $\pi$ with indices less than $l$.

- $U(\cdot)$: Uniform distribution.

- $p_\lambda(\cdot|\boldsymbol{x}_0)$: The joint distribution induced by masking each dimension in $\boldsymbol{x}_0$ independently with a probability $\lambda$.

- Cat: Categorical distribution.

- NFEs: Number of function evaluations.

- E-NFEs: Expected number of function evaluations.

## B    PROOF OF THEOREM 1

In this section, we provide a detailed proof of Theorem 1, which is carried out in three key steps. The core idea of the proof involves leveraging the properties of a continuous-time Markov chain with an absorbing state, where the forward diffusion process is independent across different dimensions. This independence simplifies the analysis of both the conditional and joint distributions.

First, we derive the analytic form of the conditional distribution, as stated in Lemma 1. This can be derived directly from Eq. (2.3), but for a better understanding, we provide a more intuitive proof for $\boldsymbol{Q}_t = \sigma(t)\boldsymbol{Q}^{\text{absorb}}$. Second, we extend this analysis to multiple dimensions to obtain the joint distribution, as formalized in Proposition 1. Finally, by simply dividing the joint distributions derived in the second step, we decouple the concrete score, thereby completing the proof of Theorem 1.

**Lemma 1.** *(Analytic conditional distribution in absorbing diffusion) Suppose $\{X_t\}$ is a continuous time Markov chain with transition rate matrix $\boldsymbol{Q}_t = \sigma(t)\boldsymbol{Q}^{\text{absorb}}$, given the value $x_0$ at time zero , the conditional distribution $p_{t|0}(x_t|x_0)$ has the following analytic form:*

$$p_{t|0}(x_t|x_0) = \begin{cases} e^{-\bar{\sigma}(t)}, & x_t = x_0, \\ 1 - e^{-\bar{\sigma}(t)}, & x_t = [\boldsymbol{M}], \\ 0, & x_t \neq [\boldsymbol{M}] \text{ and } x_t \neq x_0. \end{cases} \tag{B.1}$$

*Proof.* Given the initial value $x_0 \in \mathcal{X} = \{1, \cdots, N\}$, we have

$$x_t = \begin{cases} x_0, & t < T_h, \\ [\mathbf{M}], & t \geq T_h. \end{cases} \tag{B.2}$$

Here, $T_h$ represents the holding time before $x_0$ transitions to the absorbing state $[\mathbf{M}]$.

Based on the definition of the $\boldsymbol{Q}_t$ in Eq. (2.2) and $\boldsymbol{Q}^{\text{absorb}}$, the probability of $x_0$ remaining the same after a small time increment $\Delta t$ is

$$p_{t+\Delta t|t}(x_0|x_0) = 1 + \sigma(t)\boldsymbol{Q}^{\text{absorb}}(x_0, x_0)\Delta t + o(\Delta t). \tag{B.3}$$

Partitioning the interval $[0, t]$ into $\{s_k\}_{k=0}^n$ and utilizing the memoryless property of continuous-time Markov chains, we can express the probability of $x_0$ remaining the same from time 0 to $t$ as a product

of probabilities over these small intervals. This gives us:

$$p_{t|0}(x_0|x_0) = \prod_{k=1}^{n} p_{s_k|s_{k-1}}(x_0|x_0) \tag{B.4}$$

$$= \prod_{k=1}^{n} \left(1 + \sigma(t_{k-1})\boldsymbol{Q}^{\text{absorb}}(x_0, x_0)(s_k - s_{k-1}) + o(s_k - s_{k-1})\right) \tag{B.5}$$

$$= \exp\left(\sum_{k=1}^{n} \ln\left(1 + \sigma(t_{k-1})\boldsymbol{Q}^{\text{absorb}}(x_0, x_0)(s_k - s_{k-1}) + o(s_k - s_{k-1})\right)\right) \tag{B.6}$$

$$= \exp\left(\sum_{k=1}^{n} \sigma(t_{k-1})\boldsymbol{Q}^{\text{absorb}}(x_0, x_0)(s_k - s_{k-1}) + o(s_k - s_{k-1})\right). \tag{B.7}$$

Let $\max(s_k - s_{k-1}) \to 0$, the Riemann sum in Eq. (B.7) equals the following continuous integral:

$$p_{t|0}(x_0|x_0) = \exp\left(\int_0^t \sigma(s)\boldsymbol{Q}^{\text{absorb}}(x_0, x_0)ds\right) = \exp\left(\boldsymbol{Q}^{\text{absorb}}(x_0, x_0)\bar{\sigma}(t)\right). \tag{B.8}$$

By Eq. (2.4), $\boldsymbol{Q}^{\text{absorb}}(x_0, x_0) = -1$, we have

$$p_{t|0}(x_0|x_0) = P(T_h > t) = e^{-\bar{\sigma}(t)} \tag{B.9}$$

$$p_{t|0}([\mathbf{M}]|x_0) = P(T_h \le t) = 1 - e^{-\bar{\sigma}(t)} \tag{B.10}$$

$$p_{t|0}(k|x_0) = 0 \quad \text{if } k \ne [\mathbf{M}] \text{ and } k \ne x_0. \tag{B.11}$$

Similarly, given value $x_s$ at time $s < t$, the conditional distribution can be expressed as

$$p_{t|s}(x_t|x_s) = \begin{cases} e^{-(\bar{\sigma}(t)-\bar{\sigma}(s))}, & x_t = x_s, \\ 1 - e^{-(\bar{\sigma}(t)-\bar{\sigma}(s))}, & x_t = [\mathbf{M}], \\ 0, & x_t \ne [\mathbf{M}] \text{ and } x_t \ne x_s. \end{cases} \tag{B.12}$$

$\square$

**Proposition 1.** *(Analytic joint distribution in absorbing diffusion)*

*Suppose $\{X_t\}$ is a continuous time Markov chain with transition rate matrix $\boldsymbol{Q}_t = \sigma(t)\boldsymbol{Q}^{absorb}$. For $\boldsymbol{x}_t = x_t^1 \cdots x_t^d$ with $d_1$ components as $[\boldsymbol{M}]$ and $d_2 = d - d_1$ components as unmasked tokens, $p_t(\boldsymbol{x}_t)$ can be expressed as*

$$p_t(\boldsymbol{x}_t) = [1 - e^{-\bar{\sigma}(t)}]^{d_1}[e^{-\bar{\sigma}(t)}]^{d_2} p_0(\boldsymbol{x}_t^{UM}), \tag{B.13}$$

*where $\boldsymbol{x}_t^{UM}$ is the vector consists of all unmasked tokens of $\boldsymbol{x}_t$.*

Proposition 1 shows that the joint distribution $p_t(\boldsymbol{x}_t)$ can be expressed as the multiplication of two terms. One is an analytic term only depending on time, the other is a $d_2$ dimensions joint distribution of clean data $p_0(\boldsymbol{x}_t^{\text{UM}})$ independent of time.

*Proof.* Without loss of generality, let's assume that the preceding $d_1$ terms of $\boldsymbol{x}$ are all $[\mathbf{M}]$, and the remaining $d_2$ terms are unmasked tokens. That is, $\boldsymbol{x}_t = [\mathbf{M}] \cdots [\mathbf{M}]x_t^{d_1+1} \cdots x_t^d$, and here $x^k$ is an unmasked token in $\mathcal{X}$.

Using the law of total probability and Lemma 1, along with the assumption of independence between different dimensions of the diffusion process, we can express the joint distribution $p_t([\mathbf{M}] \cdots [\mathbf{M}]x_t^{d_1+1} \cdots x_t^d)$ as a sum over all possible initial states $\boldsymbol{x}_0 \in \mathcal{X}^d$:

$$p_t([\mathbf{M}] \cdots [\mathbf{M}]x_t^{d_1+1} \cdots x_t^d) = \sum_{\boldsymbol{x}_0 \in \mathcal{X}^d} p_{t|0}([\mathbf{M}] \cdots [\mathbf{M}]x_t^{d_1+1} \cdots x_t^d|\boldsymbol{x}_0)p_0(\boldsymbol{x}_0)$$

$$= \sum_{x_0^1 \in \mathcal{X}, \cdots, x_0^d \in \mathcal{X}} p_{t|0}([\mathbf{M}] \cdots [\mathbf{M}]x_t^{d_1+1} \cdots x_t^d|x_0^1 \cdots x_0^d)p_0(x_0^1 \cdots x_0^d)$$

$$= \sum_{x_0^1 \in \mathcal{X}, \cdots, x_0^d \in \mathcal{X}} \prod_{k=1}^{d_1} p_{t|0}^k([\mathbf{M}]|x_0^k) \prod_{k=d_1+1}^{d} p_{t|0}^k(x_t^k|x_0^k)p_0(x_0^1 \cdots x_0^d).$$

Substituting the analytic forms of $p_{t|0}^k([\mathbf{M}]|x_0^k)$ and $p_{t|0}^k(x_t^k|x_0^k)$ from Lemma 1, above equations can be further simplified as follows:

$$\sum_{x_0^1 \in \mathcal{X}, \cdots, x_0^d \in \mathcal{X}} \prod_{k=1}^{d_1} p_{t|0}^k([\mathbf{M}]|x_0^k) \prod_{k=d_1+1}^{d} p_{t|0}^k(x_t^k|x_0^k) p_0(x_0^1 \cdots x_0^d)$$

$$= \sum_{x_0^1 \in \mathcal{X}, \cdots, x_0^{d_1} \in \mathcal{X}} \prod_{k=1}^{d_1} p_{t|0}^k([\mathbf{M}]|x_0^k) [e^{-\bar{\sigma}(t)}]^{d_2} p_0(x_0^1 \cdots x_0^{d_1} x_t^{d_1+1} \cdots x_t^d)$$

$$= \sum_{x_0^1 \in \mathcal{X}, \cdots, x_0^{d_1} \in \mathcal{X}} [1 - e^{-\bar{\sigma}(t)}]^{d_1} [e^{-\bar{\sigma}(t)}]^{d_2} p_0(x_0^1 \cdots x_0^{d_1} x_t^{d_1+1} \cdots x_t^d)$$

$$= [1 - e^{-\bar{\sigma}(t)}]^{d_1} [e^{-\bar{\sigma}(t)}]^{d_2} \sum_{x_0^1 \in \mathcal{X}, \cdots, x_0^{d_1} \in \mathcal{X}} p_0(x_0^1 \cdots x_0^{d_1} x_t^{d_1+1} \cdots x_t^d)$$

$$= [1 - e^{-\bar{\sigma}(t)}]^{d_1} [e^{-\bar{\sigma}(t)}]^{d_2} p_0(x_t^{d_1+1} \cdots x_t^d).$$

By noting that $p_0(x_t^{d_1+1} \cdots x_t^d) = p_0(\boldsymbol{x}_t^{\mathrm{UM}})$, in the general case, we have

$$p_t(\boldsymbol{x}_t) = [1 - e^{-\bar{\sigma}(t)}]^{d_1} [e^{-\bar{\sigma}(t)}]^{d_2} p_0(\boldsymbol{x}_t^{\mathrm{UM}}),$$

which demonstrates that the likelihood of the noisy data $\boldsymbol{x}_t$ at time $t$ equals the likelihood of the unmasked part $\boldsymbol{x}_t^{\mathrm{UM}}$ at time 0 multiplied by an analytic time-dependent term. $\qquad\square$

**Theorem 1.** *(Analytic concrete score in absorbing diffusion, proof in Appendix B) For $\boldsymbol{x}_t = x_t^1 \ldots x_t^i \ldots x_t^d$ and $\hat{\boldsymbol{x}}_t = x_t^1 \ldots \widehat{x}_t^i \ldots x_t^d$, if $x_t^i = [\boldsymbol{M}]$ and $\hat{x}_t^i \neq [\boldsymbol{M}]$, the concrete score at time $t$ can be expressed as a* time-independent *conditional distribution at time zero multiplied by an analytic* time-dependent *term:*

$$\underbrace{\frac{p_t\left(x_t^1 \ldots \widehat{x}_t^i \ldots x_t^d\right)}{p_t\left(x_t^1 \ldots x_t^i \ldots x_t^d\right)}}_{concrete\ score} = \underbrace{\frac{e^{-\bar{\sigma}(t)}}{1 - e^{-\bar{\sigma}(t)}}}_{scalar} \underbrace{p_0(\hat{x}_t^i|\boldsymbol{x}_t^{UM})}_{\substack{clean\ data \\ distribution}}$$

*where $\boldsymbol{x}_t^{UM}$ is the vector consists of all unmasked tokens of $\boldsymbol{x}_t$.*

*Proof.* According to Proposition 1, if $x_t^i = [\mathbf{M}]$ and $\hat{x}_t^i \neq [\mathbf{M}]$, $\hat{\boldsymbol{x}}_t^{\mathrm{UM}} = (\boldsymbol{x}_t^{\mathrm{UM}}, \hat{x}_t^i)$,

$$\frac{p_t(\hat{\boldsymbol{x}}_t)}{p_t(\boldsymbol{x}_t)} = \frac{[1 - e^{-\bar{\sigma}(t)}]^{d_1-1} [e^{-\bar{\sigma}(t)}]^{d_2+1} p_0(\hat{\boldsymbol{x}}_t^{\mathrm{UM}})}{[1 - e^{-\bar{\sigma}(t)}]^{d_1} [e^{-\bar{\sigma}(t)}]^{d_2} p_0(\boldsymbol{x}_t^{\mathrm{UM}})}$$

$$= \frac{[1 - e^{-\bar{\sigma}(t)}]^{d_1-1} [e^{-\bar{\sigma}(t)}]^{d_2+1} p_0(\boldsymbol{x}_t^{\mathrm{UM}}, \hat{x}_t^i)}{[1 - e^{-\bar{\sigma}(t)}]^{d_1} [e^{-\bar{\sigma}(t)}]^{d_2} p_0(\boldsymbol{x}_t^{\mathrm{UM}})}$$

$$= \frac{e^{-\bar{\sigma}(t)}}{1 - e^{-\bar{\sigma}(t)}} p_0(\hat{x}_t^i|\boldsymbol{x}_t^{\mathrm{UM}}).$$

$\qquad\square$

## C  PROOF OF THEOREM 2

**Theorem 2.** *The absorbing discrete diffusion objective of Eq. (2.6) is equivalent to any-order autoregressive objective of Eq. (2.11) when the final total noise level $\bar{\sigma}(T) \to +\infty$.*

Here, the infinity final total noise level guarantees that all tokens will be finally masked with probability one $(1 - e^{-\bar{\sigma}(T)})$. Below we present the detailed proof in three steps.

## C.1 EQUIVALENCE BETWEEN DSE LOSS AND T-DCE LOSS

For a given noisy input $x_t$, as established in Section 3.1, $\hat{x}_t$ is valid only when it contains exactly one more unmasked token than $x_t$. In this case, the transition probability $Q_t(\hat{x}_t, x_t)$ equals $\sigma(t)$. Replace $s_\theta(x_t)$ with $\frac{e^{-\bar{\sigma}(t)}}{1-e^{-\bar{\sigma}(t)}}c_\theta(x_t)$, we can express the DSE loss in the multi-dimensional case as follows:

$$
\mathcal{L}_{\text{DSE}}^T(x_0) = \int_0^T \mathbb{E}_{x_t \sim p_{t|0}(x_t|x_0)} \left[ \sum_{x_t^i=[\mathbf{M}], j \neq [\mathbf{M}]} \sigma(t) \left( \frac{e^{-\bar{\sigma}(t)}}{1-e^{-\bar{\sigma}(t)}} c_\theta(x_t)[i,j] \right. \right.
$$
$$
\left. \left. - \frac{e^{-\bar{\sigma}(t)}}{1-e^{-\bar{\sigma}(t)}} \mathbb{I}(x_0^i = j) \log \left( \frac{e^{-\bar{\sigma}(t)}}{1-e^{-\bar{\sigma}(t)}} c_\theta(x_t)[i,j] \right) + K \left( \frac{e^{-\bar{\sigma}(t)}}{1-e^{-\bar{\sigma}(t)}} I(x_0^i = j) \right) \right) \right] dt
$$
$$
= \int_0^T \mathbb{E}_{x_t \sim p_{t|0}(x_t|x_0)} \left[ \sum_{x_t^i=[\mathbf{M}], j \neq [\mathbf{M}]} \sigma(t) \left( \frac{e^{-\bar{\sigma}(t)}}{1-e^{-\bar{\sigma}(t)}} c_\theta(x_t)[i,j] \right) \right] dt
$$
$$
+ \int_0^T \mathbb{E}_{x_t \sim p_{t|0}(x_t|x_0)} \left[ \sum_{x_t^i=[\mathbf{M}], j \neq [\mathbf{M}]} -\frac{\sigma(t)e^{-\bar{\sigma}(t)}}{1-e^{-\bar{\sigma}(t)}} \mathbb{I}(x_0^i = j) \log \left( \frac{e^{-\bar{\sigma}(t)}}{1-e^{-\bar{\sigma}(t)}} c_\theta(x_t)[i,j] \right) \right] dt
$$
$$
+ \int_0^T \mathbb{E}_{x_t \sim p_{t|0}(x_t|x_0)} \left[ \sum_{x_t^i=[\mathbf{M}], j \neq [\mathbf{M}]} \sigma(t)K \left( \frac{e^{-\bar{\sigma}(t)}}{1-e^{-\bar{\sigma}(t)}} I(x_0^i = j) \right) \right] dt.
$$

We analyze each term in the above equation separately. The first term simplifies due to the property $\sum_{j \neq [\mathbf{M}]} c_\theta(x_t)[i,j] = 1$:

$$
\int_0^T \mathbb{E}_{x_t \sim p_{t|0}(x_t|x_0)} \left[ \sum_{x_t^i=[\mathbf{M}]} \sigma(t) \frac{e^{-\bar{\sigma}(t)}}{1-e^{-\bar{\sigma}(t)}} \right] dt. \tag{C.1}
$$

The third term can be simplified by substituting $K(a) = a \log a - a$ and using $0 \log 0 = 0$:

$$
\int_0^T \mathbb{E}_{x_t \sim p_{t|0}(x_t|x_0)} \left[ \sum_{x_t^i=[\mathbf{M}]} \sigma(t) \frac{e^{-\bar{\sigma}(t)}}{1-e^{-\bar{\sigma}(t)}} \left( \log \frac{e^{-\bar{\sigma}(t)}}{1-e^{-\bar{\sigma}(t)}} - 1 \right) \right] dt. \tag{C.2}
$$

Combining the first and third terms:

$$
\int_0^T \mathbb{E}_{x_t \sim p_{t|0}(x_t|x_0)} \left[ \sum_{x_t^i=[\mathbf{M}]} \sigma(t) \frac{e^{-\bar{\sigma}(t)}}{1-e^{-\bar{\sigma}(t)}} \left( \log \frac{e^{-\bar{\sigma}(t)}}{1-e^{-\bar{\sigma}(t)}} \right) \right] dt \tag{C.3}
$$
$$
= \int_0^T d(1 - e^{-\bar{\sigma}(t)})\sigma(t) \frac{e^{-\bar{\sigma}(t)}}{1-e^{-\bar{\sigma}(t)}} \left( \log \frac{e^{-\bar{\sigma}(t)}}{1-e^{-\bar{\sigma}(t)}} \right) dt \tag{C.4}
$$
$$
= d \int_0^T \sigma(t) e^{-\bar{\sigma}(t)} \log \frac{e^{-\bar{\sigma}(t)}}{1-e^{-\bar{\sigma}(t)}} dt. \tag{C.5}
$$

Introducing a new variable $\lambda(t) = 1 - e^{-\bar{\sigma}(t)}$, which represents the probability of a token being masked from 0 to $t$ in the forward process. As $\bar{\sigma}(t) = \int_0^t \sigma(\tau)d\tau$ and $\bar{\sigma}(T) = \infty$, we have $\lambda(0) = 0$, $\lambda(T) = 1$ and $d\lambda = \sigma(t)e^{-\bar{\sigma}(t)}dt$. Obviously, $\lambda(t)$ is invertible, which allows us to perform a change of variables from $t$ to $\lambda$ and simplifies Eq. (C.5) to

$$
d \int_0^1 \log \frac{1-\lambda}{\lambda} d\lambda = -d \left( \lambda \log \lambda + (1-\lambda) \log(1-\lambda) \right) |_0^1 = 0. \tag{C.6}
$$

Here we used

$$\lim_{\lambda \to 0} \lambda \log \lambda = \lim_{\lambda \to 1}(1-\lambda)\log(1-\lambda) = 0.$$

Thus, the DSE loss reduces to the second term, which we define as the $t$-denoising cross-entropy loss ($t$-DCE):

$$\mathcal{L}_{t\text{-DCE}}^T(\boldsymbol{x}_0) = \int_0^T \mathbb{E}_{\boldsymbol{x}_t \sim p_{t|0}(\boldsymbol{x}_t|\boldsymbol{x}_0)} \left[ \sum_{\substack{x_t^i=[\mathbf{M}]\\j\neq[\mathbf{M}]}} -\frac{\sigma(t)e^{-\bar{\sigma}(t)}}{1-e^{-\bar{\sigma}(t)}} \mathbb{I}(x_0^i = j) \log\left(\frac{e^{-\bar{\sigma}(t)}\boldsymbol{c}_\theta(\boldsymbol{x}_t)[i,j]}{1-e^{-\bar{\sigma}(t)}}\right) \right] dt$$

$$= \int_0^T \mathbb{E}_{\boldsymbol{x}_t \sim p_{t|0}(\boldsymbol{x}_t|\boldsymbol{x}_0)} \left[ \sum_{x_t^i=[\mathbf{M}]} -\frac{\sigma(t)e^{-\bar{\sigma}(t)}}{1-e^{-\bar{\sigma}(t)}} \log\left(\frac{e^{-\bar{\sigma}(t)}}{1-e^{-\bar{\sigma}(t)}} \boldsymbol{c}_\theta(\boldsymbol{x}_t)[i,x_0^i]\right) \right] dt$$

$$= \int_0^T \mathbb{E}_{\boldsymbol{x}_t \sim p_{t|0}(\boldsymbol{x}_t|\boldsymbol{x}_0)} \left[ \sum_{x_t^i=[\mathbf{M}]} -\frac{\sigma(t)e^{-\bar{\sigma}(t)}}{1-e^{-\bar{\sigma}(t)}} \log\left(\frac{e^{-\bar{\sigma}(t)}}{1-e^{-\bar{\sigma}(t)}} q_\theta(x_0^i|\boldsymbol{x}_t^{\text{UM}})\right) \right] dt.$$

## C.2 EQUIVALENCE BETWEEN T-DCE LOSS AND LAMBDA-DCE LOSS

Starting from the $t$-DCE loss in Eq. (3.5), we can perform a change of variable from $t$ to $\lambda(t) = 1 - e^{-\bar{\sigma}(t)}$, as demonstrated in Appendix C.1. This allows us to rewrite the $t$-DCE loss integral in terms of $\lambda$:

$$\int_0^1 \frac{1}{\lambda} \mathbb{E}_{\boldsymbol{x}_\lambda \sim p_\lambda(\boldsymbol{x}_\lambda|\boldsymbol{x}_0)} \left[ \sum_{x_\lambda^i=[\mathbf{M}]} -\log\left(\frac{1-\lambda}{\lambda} q_\theta(x_0^i|\boldsymbol{x}_\lambda^{\text{UM}})\right) \right] d\lambda$$

$$= \int_0^1 \frac{1}{\lambda} \mathbb{E}_{\boldsymbol{x}_\lambda \sim p_\lambda(\boldsymbol{x}_\lambda|\boldsymbol{x}_0)} \left[ \sum_{x_\lambda^i=[\mathbf{M}]} -\log q_\theta(x_0^i|\boldsymbol{x}_\lambda^{\text{UM}}) \right] d\lambda$$

$$- \int_0^1 \frac{1}{\lambda} \mathbb{E}_{\boldsymbol{x}_\lambda \sim p_\lambda(\boldsymbol{x}_\lambda|\boldsymbol{x}_0)} \left[ \sum_{x_\lambda^i=[\mathbf{M}]} \log \frac{1-\lambda}{\lambda} \right] d\lambda.$$

Given the independence of the forward process and Lemma 1, the original probability $p_{t|0}(\boldsymbol{x}_t|\boldsymbol{x}_0)$ can be factorized as $\prod_{i=1}^d p_{t|0}^i(x_t^i|x_0^i)$, where

$$p_{t|0}^i(x_t^i|x_0^i) = \begin{cases} 1 - e^{-\bar{\sigma}(t)}, & x_t^i = [\mathbf{M}], \\ e^{-\bar{\sigma}(t)}, & x_t^i = x_0^i, \\ 0, & \text{else.} \end{cases} \tag{C.7}$$

Therefore, the induced probability $p_\lambda(\boldsymbol{x}_\lambda|\boldsymbol{x}_0) = \prod_{i=1}^d p_\lambda^i(x_\lambda^i|x_0^i)$ where

$$p_\lambda^i(x_\lambda^i|x_0^i) = \begin{cases} \lambda, & x_\lambda^i = [\mathbf{M}], \\ 1-\lambda, & x_\lambda^i = x_0^i, \\ 0, & \text{else.} \end{cases} \tag{C.8}$$

Next, consider the second term. Similar to Eq. (C.3), we can prove that it equals zero:

$$\int_0^1 \frac{1}{\lambda} \mathbb{E}_{\boldsymbol{x}_\lambda \sim p_\lambda(\boldsymbol{x}_\lambda|\boldsymbol{x}_0)} \left[ \sum_{x_\lambda^i=[\mathbf{M}]} \log(\frac{1-\lambda}{\lambda}) \right] = 0. \tag{C.9}$$

Therefore, $t$-DCE loss is equivalent to the first term, defined as $\lambda$-denoising cross-entropy ($\lambda$-DCE):

$$\mathcal{L}_{\lambda\text{-DCE}}^T(\boldsymbol{x}_0) = \int_0^1 \frac{1}{\lambda} \mathbb{E}_{\boldsymbol{x}_\lambda \sim p_\lambda(\boldsymbol{x}_\lambda|\boldsymbol{x}_0)} \left[ \sum_{x_\lambda^i=[\mathbf{M}]} -\log q_\theta(x_0^i|\boldsymbol{x}_\lambda^{\text{UM}}) \right] d\lambda. \tag{C.10}$$

## C.3 EQUIVALENCE BETWEEN LAMBDA-DCE LOSS AND ANY-ORDER AUTOREGRESSIVE LOSS

Based on $\lambda$-DCE loss in Eq. (3.6), we first define the sample space and analytically express the expectation term.

Given $\boldsymbol{x}_0$, we define the sample space of $\boldsymbol{x}_\lambda$ as $\tilde{\mathcal{X}}(\boldsymbol{x}_0) := \{x_0^1, [\mathbf{M}]\} \times \cdots \{x_0^d, [\mathbf{M}]\}$ and $\tilde{\mathcal{X}}_k(\boldsymbol{x}_0) :=$ $\{\tilde{\boldsymbol{x}} : \tilde{\boldsymbol{x}} \in \tilde{\mathcal{X}}(\boldsymbol{x}_0) \wedge \tilde{\boldsymbol{x}}$ has exact k dimensions with values $[\mathbf{M}]\}$ . It follows that $|\tilde{\mathcal{X}}(\boldsymbol{x}_0)| = 2^d$ and $|\tilde{\mathcal{X}}_k(\boldsymbol{x}_0)| = \binom{d}{k}$. Therefore, the sample space $\tilde{\mathcal{X}}(\boldsymbol{x}_0)$ can be decoupled by the number of masked tokens $k$ in $\tilde{\boldsymbol{x}}$:

$$\int_0^1 \frac{1}{\lambda} \mathbb{E}_{\boldsymbol{x}_\lambda \sim p_\lambda(\boldsymbol{x}_\lambda|\boldsymbol{x}_0)} \left[ \sum_{\tilde{x}^i=[\mathbf{M}]} -\log q_\theta(x_0^i|\boldsymbol{x}_\lambda^{\mathrm{UM}}) \right] d\lambda \tag{C.11}$$

$$= \int_0^1 \frac{1}{\lambda} \sum_{\tilde{x} \in \tilde{\mathcal{X}}(\boldsymbol{x}_0)} p_\lambda(\tilde{\boldsymbol{x}}|\boldsymbol{x}_0) \left[ \sum_{\tilde{x}^i=[\mathbf{M}]} -\log q_\theta(x_0^i|\tilde{\boldsymbol{x}}^{\mathrm{UM}}) \right] d\lambda \tag{C.12}$$

$$= \int_0^1 \frac{1}{\lambda} \sum_{k=0}^d \sum_{\tilde{\boldsymbol{x}} \in \tilde{\mathcal{X}}_k(\boldsymbol{x}_0)} \lambda^k(1-\lambda)^{d-k} \left[ \sum_{\tilde{x}^i=[\mathbf{M}]} -\log q_\theta(x_0^i|\tilde{\boldsymbol{x}}^{\mathrm{UM}}) \right] d\lambda \tag{C.13}$$

$$= \int_0^1 \frac{1}{\lambda} \sum_{k=1}^d \sum_{\tilde{\boldsymbol{x}} \in \tilde{\mathcal{X}}_k(\boldsymbol{x}_0)} \lambda^k(1-\lambda)^{d-k} \left[ \sum_{\tilde{x}^i=[\mathbf{M}]} -\log q_\theta(x_0^i|\tilde{\boldsymbol{x}}^{\mathrm{UM}}) \right] d\lambda. \tag{C.14}$$

The last equation holds because there are no masked tokens when $k = 0$, and the inner sum is zero.

From Eq. (C.14), by rearranging the order of summation and integration, we can analytically evaluate the integral $\int_0^1 \lambda^{k-1}(1-\lambda)^{d-k} d\lambda$ using the Beta function, which eliminates $\lambda$:

$$Eq.\ (C.14) = \sum_{k=1}^d \int_0^1 \lambda^{k-1}(1-\lambda)^{d-k} d\lambda \sum_{\tilde{\boldsymbol{x}} \in \tilde{\mathcal{X}}_k(\boldsymbol{x}_0)} \left[ \sum_{\tilde{x}^i=[\mathbf{M}]} -\log q_\theta(x_0^i|\tilde{\boldsymbol{x}}^{\mathrm{UM}}) \right] \tag{C.15}$$

$$= \sum_{k=1}^d \frac{(k-1)!(d-k)!}{d!} \sum_{\tilde{\boldsymbol{x}} \in \tilde{\mathcal{X}}_k(\boldsymbol{x}_0)} \left[ \sum_{\tilde{x}^i=[\mathbf{M}]} -\log q_\theta(x_0^i|\tilde{\boldsymbol{x}}^{\mathrm{UM}}) \right] \tag{C.16}$$

$$= \sum_{k=1}^d \frac{1}{kC_d^k} \sum_{\tilde{\boldsymbol{x}} \in \tilde{\mathcal{X}}_k(\boldsymbol{x}_0)} \left[ \sum_{\tilde{x}^i=[\mathbf{M}]} -\log q_\theta(x_0^i|\tilde{\boldsymbol{x}}^{\mathrm{UM}}) \right]. \tag{C.17}$$

Eq. (C.17) can be reformulated in terms of an expectation over a uniform distribution $U(\tilde{\mathcal{X}}_k(\boldsymbol{x}_0))$ as follows:

$$\sum_{k=1}^d \frac{1}{k} \mathbb{E}_{\tilde{\boldsymbol{x}} \sim U(\tilde{\mathcal{X}}_k(\boldsymbol{x}_0))} \left[ \sum_{\tilde{x}^i=[\mathbf{M}]} \left( -\log(q_\theta(x_0^i|\tilde{\boldsymbol{x}}^{\mathrm{UM}})) \right) \right]. \tag{C.18}$$

Let $\pi$ be one permutation of the integers $1, \cdots, d$, and $U_\pi$ represent the uniform distribution of all orders. We note that Eq. (C.18) is equivalent to the following term from the perspective of any-order autoregressive model:

$$\sum_{k=1}^d \frac{1}{k} \mathbb{E}_{\pi \sim U_\pi} \sum_{r=d-k+1}^d -\log q_\theta(x_0^{\pi(r)}|x_0^{\pi(<d-k+1)}; \pi). \tag{C.19}$$

Here, $x_0^{\pi(<l)}$ denotes the sequence of the first $l-1$ elements in the permutation $\pi$. Given a fixed $k$, the term $x_0^{\pi(<d-k+1)}$ can be interpreted as the unmasked part of the noisy data $\tilde{\boldsymbol{x}}^{\mathrm{UM}}$. For $r =$

$d - k + 1, \cdots, d$, $x_0^{\pi(r)}$ corresponds to the $k$ items of the masked part. Since both $\pi$ and $\tilde{\boldsymbol{x}}$ are both uniformly sampled, Eq. (C.18) and Eq. (C.19) are equivalent.

Further, we can make a simple subscription transformation by letting $l = d - k + 1$ and change the summation to Monte Carlo estimation on Eq. (C.19) :

$$d \cdot \mathbb{E}_{l \sim U(1,\cdots,d)} \frac{1}{d - l + 1} \mathbb{E}_{\pi \sim U_\pi} \sum_{r=l}^{d} - \log q_\theta(x_0^{\pi(r)} | x_0^{\pi(<l)}; \pi). \tag{C.20}$$

In (Uria et al., 2014; Hoogeboom et al., 2022), it was proved that Eq. (C.20) is mathematically equivalent to Eq. (2.11). Actually, Eq. (C.20) is widely used as a training objective for any-order autoregressive models for efficient parallel optimization.

This concludes our proof of Theorem 2.

## D  SAMPLING METHODS

In this section, we first derived the exact reverse distribution for absorbing discrete diffusion in Appendix D.1. This derivation led to simplified forms of the Tweedie $\tau$-leaping and Euler methods, detailed in Appendix D.2 and Appendix D.3, respectively. In Appendix D.4, we proved the equivalence of these two methods under a log-linear noise schedule. Finally, in Appendix D.5, we discussed the expected number of function evaluations (E-NFEs) for these methods.

### D.1  EXACT REVERSE DISTRIBUTION IN ABSORBING DISCRETE DIFFUSION

**Lemma 2.** *(Analytic reverse distribution in absorbing diffusion) Suppose $\{X_t\}$ is a continuous time Markov chain with transition rate matrix $\boldsymbol{Q}_t = \sigma(t)\boldsymbol{Q}^{absorb}$. For $\boldsymbol{x}_t = x_t^1 \cdots x_t^d$ with $d_1$ masked tokens and $d_2 = d - d_1$ unmasked tokens, and $\boldsymbol{x}_s = x_s^1 \cdots x_s^d$ with $d_1 - \Delta d$ masked tokens and $d_2 + \Delta d$ unmasked tokens, the reverse distribution is given by:*

$$p_{s|t}(\boldsymbol{x}_s|\boldsymbol{x}_t) = \begin{cases} \left[\frac{e^{-\bar{\sigma}(s)} - e^{-\bar{\sigma}(t)}}{1 - e^{-\bar{\sigma}(s)}}\right]^{\Delta d} \left[\frac{1 - e^{-\bar{\sigma}(s)}}{1 - e^{-\bar{\sigma}(t)}}\right]^{d_1} \frac{p_0(\boldsymbol{x}_s^{UM})}{p_0(\boldsymbol{x}_t^{UM})}, & \boldsymbol{x}_t \subseteq_{UM} \boldsymbol{x}_s, \\ 0, & \boldsymbol{x}_t \not\subseteq_{UM} \boldsymbol{x}_s, \end{cases} \tag{D.1}$$

*where $\boldsymbol{x}_t \subseteq_{UM} \boldsymbol{x}_s$ denotes $\forall i : \boldsymbol{x}_t^i \neq [\boldsymbol{M}]$, we have $\boldsymbol{x}_t^i = \boldsymbol{x}_s^i$.*

*Proof.* Using Bayes' theorem, $p_{s|t}(\boldsymbol{x}_s|\boldsymbol{x}_t) = p_{t|s}(\boldsymbol{x}_t|\boldsymbol{x}_s) \frac{p_s(\boldsymbol{x}_s)}{p_t(\boldsymbol{x}_t)}$.

From Proposition 1:

$$p_t(\boldsymbol{x}_t) = [1 - e^{-\bar{\sigma}(t)}]^{d_1} [e^{-\bar{\sigma}(t)}]^{d_2} p_0(\boldsymbol{x}_t^{UM}), \tag{D.2}$$

$$p_s(\boldsymbol{x}_s) = [1 - e^{-\bar{\sigma}(s)}]^{d_1 - \Delta d} [e^{-\bar{\sigma}(s)}]^{d_2 + \Delta d} p_0(\boldsymbol{x}_s^{UM}). \tag{D.3}$$

Utilizing Eq. (B.12), we get

$$p_{t|s}(\boldsymbol{x}_t|\boldsymbol{x}_s) = \prod_{i=1}^{d} p_{t|s}^i(x_t^i|x_s^i) = \begin{cases} \left[e^{-(\bar{\sigma}(t) - \bar{\sigma}(s))}\right]^{d_2} \left[1 - e^{-(\bar{\sigma}(t) - \bar{\sigma}(s))}\right]^{\Delta d}, & \boldsymbol{x}_t \subseteq_{UM} \boldsymbol{x}_s, \\ 0, & \boldsymbol{x}_t \not\subseteq_{UM} \boldsymbol{x}_s. \end{cases} \tag{D.4}$$

Simplifying these equations, we can express $p_{s|t}(\boldsymbol{x}_s|\boldsymbol{x}_t)$ as

$$p_{s|t}(\boldsymbol{x}_s|\boldsymbol{x}_t) = \begin{cases} \left[\frac{e^{-\bar{\sigma}(s)} - e^{-\bar{\sigma}(t)}}{1 - e^{-\bar{\sigma}(s)}}\right]^{\Delta d} \left[\frac{1 - e^{-\bar{\sigma}(s)}}{1 - e^{-\bar{\sigma}(t)}}\right]^{d_1} \frac{p_0(\boldsymbol{x}_s^{UM})}{p_0(\boldsymbol{x}_t^{UM})}, & \boldsymbol{x}_t \subseteq_{UM} \boldsymbol{x}_s, \\ 0, & \boldsymbol{x}_t \not\subseteq_{UM} \boldsymbol{x}_s. \end{cases} \tag{D.5}$$

$\square$

It should be noted that when $\boldsymbol{x}_t \subseteq_{UM} \boldsymbol{x}_s$, the ratio $\frac{p_0(\boldsymbol{x}_s^{UM})}{p_0(\boldsymbol{x}_t^{UM})}$ can be reformulated as a $d_1$-dimensional conditional distribution $p_0(\boldsymbol{x}_s^{UM}|\boldsymbol{x}_t^{UM})$ with $N^{d_1}$ states. This is not accessible using our one-dimensional conditional distribution $p_0(\hat{x}_t^i|\boldsymbol{x}_t^{UM})$ in Theorem 1 if $d_1 > 1$. Therefore, for efficiency, existing samplers assume that each dimension is independent within a small interval and update each dimension in parallel (Lou et al., 2024; Campbell et al., 2022).

## D.2 TWEEDIE $\tau$-LEAPING METHOD AND ITS SIMPLIFIED FORM IN RADD

Given the vector $\boldsymbol{x}_t$, if we sample each $x_s^i$ independently, the factorization of marginal distribution $p_{s|t}^{\text{tweedie}}$ results in the minimum KL divergence with true reverse $p_{s|t}(\boldsymbol{x}_s|\boldsymbol{x}_t)$ (proof in (Lou et al., 2024), Appendix A). This assumption formally defines $p_{s|t}^{\text{tweedie}}$ as follows:

$$p_{s|t}^{\text{tweedie}}(\boldsymbol{x}_s|\boldsymbol{x}_t) = \prod_{i=1}^{d} p_{s|t}^{\text{tweedie},i}(x_s^i|\boldsymbol{x}_t) = \prod_{i=1}^{d} p_{s|t}^i(x_s^i|\boldsymbol{x}_t). \tag{D.6}$$

To sample from $p_{s|t}^{\text{tweedie}}$, we need to derive the analytic form of $p_{s|t}^i(x_s^i|\boldsymbol{x}_t)$. Without loss of generality, let's assume that the preceding $d_1$ terms of $\boldsymbol{x}_t$ are all $[\mathbf{M}]$, and the remaining $d_2$ terms are unmasked tokens.

For illustration, we can take $i = 1$ as an example. Let $\tilde{\mathcal{X}}_k$ denote the sample space of length $d_1 - 1$ sequence where each sequence has exact $k$ masked tokens, with $|\tilde{\mathcal{X}}_k| = C_{d_1-1}^k N^{d_1-1-k}$. When $x_s^1 \neq [\mathbf{M}]$, According to Lemma 2:

$$
\begin{aligned}
p_{s|t}^1(x_s^1|\boldsymbol{x}_t) &= \sum_{\boldsymbol{x}_s^{2:d}} p_{s|t}(\boldsymbol{x}_s|\boldsymbol{x}_t) \\
&= \sum_{k=0}^{d_1-1} \sum_{\boldsymbol{x}_s^{2:d} \in \tilde{\mathcal{X}}_k} \left[\frac{e^{-\bar{\sigma}(s)} - e^{-\bar{\sigma}(t)}}{1 - e^{-\bar{\sigma}(s)}}\right]^{k+1} \left[\frac{1 - e^{-\bar{\sigma}(s)}}{1 - e^{-\bar{\sigma}(t)}}\right]^{d_1} \frac{p_0(x_s^1, \boldsymbol{x}_s^{2:d,\text{UM}}, \boldsymbol{x}_t^{d_1+1:d})}{p_0(\boldsymbol{x}_t^{d_1+1:d})} \\
&= \sum_{k=0}^{d_1-1} C_{d_1-1}^k \left[\frac{e^{-\bar{\sigma}(s)} - e^{-\bar{\sigma}(t)}}{1 - e^{-\bar{\sigma}(s)}}\right]^{k+1} \left[\frac{1 - e^{-\bar{\sigma}(s)}}{1 - e^{-\bar{\sigma}(t)}}\right]^{d_1} \frac{p_0(x_s^1, \boldsymbol{x}_t^{d_1+1:d})}{p_0(\boldsymbol{x}_t^{d_1+1:d})} \\
&= \frac{e^{-\bar{\sigma}(s)} - e^{-\bar{\sigma}(t)}}{1 - e^{-\bar{\sigma}(s)}} \left[1 + \frac{e^{-\bar{\sigma}(s)} - e^{-\bar{\sigma}(t)}}{1 - e^{-\bar{\sigma}(s)}}\right]^{d_1-1} \left[\frac{1 - e^{-\bar{\sigma}(s)}}{1 - e^{-\bar{\sigma}(t)}}\right]^{d_1} \frac{p_0(x_s^1, \boldsymbol{x}_t^{d_1+1:d})}{p_0(\boldsymbol{x}_t^{d_1+1:d})} \\
&= \frac{e^{-\bar{\sigma}(s)} - e^{-\bar{\sigma}(t)}}{1 - e^{-\bar{\sigma}(t)}} \frac{p_0(x_s^1, \boldsymbol{x}_t^{d_1+1:d})}{p_0(\boldsymbol{x}_t^{d_1+1:d})} \\
&= \frac{e^{-\bar{\sigma}(s)} - e^{-\bar{\sigma}(t)}}{1 - e^{-\bar{\sigma}(t)}} p_0(x_s^1|\boldsymbol{x}_t^{d_1+1:d}).
\end{aligned}
$$

Here, we used the binomial expansion identity:

$$\sum_{k=0}^{d_1-1} C_{d_1-1}^k \left[\frac{e^{-\bar{\sigma}(s)} - e^{-\bar{\sigma}(t)}}{1 - e^{-\bar{\sigma}(s)}}\right]^k = \left[1 + \frac{e^{-\bar{\sigma}(s)} - e^{-\bar{\sigma}(t)}}{1 - e^{-\bar{\sigma}(s)}}\right]^{d_1-1}.$$

Similarly, for $x_s^1 = [\mathbf{M}]$:

$$p_{s|t}^1([\mathbf{M}]|\boldsymbol{x}_t) = \frac{1 - e^{-\bar{\sigma}(s)}}{1 - e^{-\bar{\sigma}(t)}}. \tag{D.7}$$

In general, we have

$$p_{s|t}^i(x_s^i|\boldsymbol{x}_t) = \begin{cases} \frac{e^{-\bar{\sigma}(s)} - e^{-\bar{\sigma}(t)}}{1 - e^{-\bar{\sigma}(t)}} p_0(x_s^i|\boldsymbol{x}_t^{\text{UM}}), & x_s^i \neq [\mathbf{M}], x_t^i = [\mathbf{M}], \\ \frac{1 - e^{-\bar{\sigma}(s)}}{1 - e^{-\bar{\sigma}(t)}}, & x_s^i = [\mathbf{M}], x_t^i = [\mathbf{M}], \\ \delta_{x_s^i x_t^i}, & x_t^i \neq [\mathbf{M}]. \end{cases} \tag{D.8}$$

With trained $\boldsymbol{c}_\theta$, we can use $\boldsymbol{c}_\theta(\boldsymbol{x}_t)[i, x_s^i]$ to approximate the true conditional distribution $p_0(x_s^i|\boldsymbol{x}_t^{\text{UM}})$ and sample by Eq. (D.8).

## D.3 EULER METHOD AND ITS SIMPLIFIED FORM IN RADD

According to theory of CTMC (Kelly, 1981; Campbell et al., 2022; Lou et al., 2024), given a particular one-dimensional input $x_t$, the transition probabilities to $x_s$ can be approximately calculated using

Eq. (2.1) and Eq. (2.7) as follows:

$$p_{s|t}(x_s|x_t) = \delta_{x_t x_s} + \tilde{\boldsymbol{Q}}_t(x_t, x_s)(t - s) + o(t - s), \tag{D.9}$$

$$\approx \delta_{x_t x_s} + \tilde{\boldsymbol{Q}}_t(x_t, x_s)(t - s), \tag{D.10}$$

where

$$\tilde{\boldsymbol{Q}}_t(x_t, x_s) = \begin{cases} \boldsymbol{Q}_t(x_s, x_t)\frac{p_t(x_s)}{p_t(x_t)}, & x_t \neq x_s, \\ -\sum_{k \neq x_t} \boldsymbol{Q}_t(x_t, k), & x_t = x_s. \end{cases} \tag{D.11}$$

Therefore, we can define the Euler approximation of the transition probability (Campbell et al., 2022; Lou et al., 2024):

$$p_{s|t}^{\text{euler}}(x_s|x_t) = \delta_{x_t x_s} + \tilde{\boldsymbol{Q}}_t(x_t, x_s)(t - s) \tag{D.12}$$

For multi-dimensional case, we factorize $p_{s|t}^{\text{euler}}(\boldsymbol{x}_s|\boldsymbol{x}_t)$ as $\prod_{i=1}^d p_{s|t}^{\text{euler},i}(x_s^i|\boldsymbol{x}_t)$, where $p_{s|t}^{\text{euler},i}(x_s^i|\boldsymbol{x}_t)$ is based on Eq. (D.12) which use $\boldsymbol{x}_t$ to replace $x_t$ and $x_t^1 \cdots x_s^i \cdots x_t^d$ to replace $x_s$.

In the case of absorbing diffusion, similar to Tweedie-$\tau$ leaping method in Appendix D.2, we can use Theorem 1 and Eq. (2.4) to simplify Eq. (D.12), which results in

$$p_{s|t}^{\text{euler},i}(x_s^i|\boldsymbol{x}_t) = \begin{cases} \sigma(t)\frac{e^{-\bar{\sigma}(t)}}{1-e^{-\bar{\sigma}(t)}}(t-s)p_0(x_s^i|\boldsymbol{x}_t^{\text{UM}}), & \text{if } x_s^i \neq [\mathbf{M}], x_t^i = [\mathbf{M}] \\ 1 - \sigma(t)\frac{e^{-\bar{\sigma}(t)}}{1-e^{-\bar{\sigma}(t)}}(t-s), & \text{if } x_s^i = [\mathbf{M}], x_t^i = [\mathbf{M}] \\ \delta_{x_s^i x_t^i}, & x_t^i \neq [\mathbf{M}]. \end{cases} \tag{D.13}$$

In practice, we also use $c_\theta(\boldsymbol{x}_t)[i, x_s^i]$ to approximate the true conditional distribution $p_0(x_s^i|\boldsymbol{x}_t^{\text{UM}})$ when sampling from Eq. (D.13).

## D.4 EQUIVALENCE OF TWEEDIE $\tau$-LEAPING AND EULER METHOD UNDER LOG-LINEAR NOISE SCHEDULE

By comparing Eq. (D.8) and Eq. (D.13), we observe that both the Tweedie $\tau$-leaping and Euler methods can be interpreted similarly:

- If $x_t^i$ is an unmasked token, keep it unchanged, i.e., $x_s^i = x_t^i$.
- If $x_t^i$ is a masked token, first determine whether it will be unmasked with a probability $\psi(t, s)$. If it is to be unmasked, then sample $x_s^i$ from $p_0(x_s^i|\boldsymbol{x}_t^{\text{UM}})$.

The only difference lies in the analytic form of $\psi(t, s)$. For the two methods, according to Eqs. (D.8) and (D.13), their corresponding $\psi(t, s)$ are given as follows:

$$\psi^{\text{tweedie}}(t, s) = \frac{e^{-\bar{\sigma}(s)} - e^{-\bar{\sigma}(t)}}{1 - e^{-\bar{\sigma}(t)}}, \tag{D.14}$$

$$\psi^{\text{euler}}(t, s) = \sigma(t)\frac{e^{-\bar{\sigma}(t)}}{1 - e^{-\bar{\sigma}(t)}}(t - s). \tag{D.15}$$

In general cases, these two expressions are not equivalent. However, if we choose a log-linear noise schedule $\bar{\sigma}(t) = 1 - \log(1 - (1 - \epsilon)t)$, both Eq. (D.15) and Eq. (D.14) can be simplified to the same form $\psi(t, s)$ as follows:

$$\psi(t, s) = \frac{t - s}{t}, \tag{D.16}$$

which shows that these two sampling methods are equivalent under a log-linear noise schedule.

## D.5 DISCUSS ON THE EXPECTATION OF NFES

In this part, we show that given the noise schedule $\sigma(t)$ and a set of time steps $\{t_0 = 0, \cdots, t_n = T\}$, the NFEs can be treated as a random variable with a calculable expected value for both Euler method and Tweedie $\tau$-leaping method.

Let $l$ denote the length of the generated sequence and $N_k \in \{0, \cdots, l\}$ denote the number of dimensions that $\boldsymbol{x}$ changed in $[t_{k-1}, t_k)$. Without loss of generality, we first consider the unconditional generation case where $l = d$. The NFEs, E-NFEs, and $N_k$ can be expressed as

$$\text{NFEs}(n) = \sum_{k=1}^{n} \mathbb{I}(N_k \neq 0), \tag{D.17}$$

$$\text{E-NFEs}(n) = \sum_{k=1}^{n} \mathbb{E}[\mathbb{I}(N_k \neq 0)] = \sum_{k=1}^{n} P(N_k \neq 0), \tag{D.18}$$

$$N_k = \sum_{i=1}^{d} \mathbb{I}(x_{t_{k-1}}^i \neq [\mathbf{M}], x_{t_k}^i = [\mathbf{M}]). \tag{D.19}$$

Furthermore, we note that the $d$ dimensions are independent. According to Eqs. (D.8) and (D.13), the probability $p_{s|t}^{\cdot,i}([\mathbf{M}]|\boldsymbol{x}_t)$ depends only on time and $x_t^i$ while independent of the other dimensions of $\boldsymbol{x}_t$. Thus, $p_{s|t}^{\cdot,i}([\mathbf{M}]|\boldsymbol{x}_t) = p_{s|t}^{\cdot,i}([\mathbf{M}]|x_t^i)$. Therefore, whether a token changes from masked to unmasked is independent across the $d$ dimensions[3]:

$$p\left(\mathbb{I}(x_s^1 = [\mathbf{M}]), \cdots, \mathbb{I}(x_s^d = [\mathbf{M}])|\mathbb{I}(x_t^1 = [\mathbf{M}]), \cdots, \mathbb{I}(x_t^d = [\mathbf{M}])\right) \tag{D.20}$$

$$= \prod_{i=1}^{d} p\left(\mathbb{I}(x_s^i = [\mathbf{M}])|\mathbb{I}(x_t^i = [\mathbf{M}])\right). \tag{D.21}$$

Since $\boldsymbol{x}_T$ consists entirely of masked tokens with probability one, each dimension of $\mathbb{I}(x_{t_{k-1}}^i \neq [\mathbf{M}], x_{t_k}^i = [\mathbf{M}])$ is independent. Consequently, $N_k$ follows a binomial distribution with parameters $d$ and $r_k$, denoted as $N_k \sim \text{Binomial}(d, r_k)$, where $r_k = p(x_{t_{k-1}}^i \neq [\mathbf{M}], x_{t_k}^i = [\mathbf{M}])$ represents the probability that $x^i$ changes within the interval $[t_{k-1}, t_k)$ in each dimension. Therefore, we can further simplify Eq. (D.18):

$$\text{E-NFEs}(n) = \sum_{k=1}^{n} P(N_k \neq 0) = \sum_{k=1}^{n} (1 - (1 - r_k)^d). \tag{D.22}$$

By definition of $r_k$ and the property of absorbing diffusion:

$$r_k = p(x_{t_{k-1}}^i \neq [\mathbf{M}], x_{t_k}^i = [\mathbf{M}]) \tag{D.23}$$

$$= p(x_{t_{k-1}}^i \neq [\mathbf{M}]|x_{t_k}^i = [\mathbf{M}]) \prod_{j=k+1}^{n} p(x_{t_{j-1}}^i = [\mathbf{M}]|x_{t_j}^i = [\mathbf{M}])p(x_{t_n}^i = [\mathbf{M}]) \tag{D.24}$$

$$= \left(1 - p(x_{t_{k-1}}^i = [\mathbf{M}]|x_{t_k}^i = [\mathbf{M}])\right) \prod_{j=k+1}^{n} p(x_{t_{j-1}}^i = [\mathbf{M}]|x_{t_j}^i = [\mathbf{M}]). \tag{D.25}$$

Eq. (D.25) can be determined given the sampling method and noise schedule.

For Tweedie $\tau$-leaping, based on Eq. (D.8), we can derive that:

$$p(x_{t_{j-1}}^i = [\mathbf{M}]|x_{t_j}^i = [\mathbf{M}]) = \frac{1 - e^{-\bar{\sigma}(t_{j-1})}}{1 - e^{-\bar{\sigma}(t_j)}}. \tag{D.26}$$

Therefore, we can express $r_k$ as

$$r_k = \left(\frac{e^{-\bar{\sigma}(t_{k-1})} - e^{-\bar{\sigma}(t_k)}}{1 - e^{-\bar{\sigma}(t_k)}}\right) \prod_{j=k+1}^{n} \left(1 - \frac{1 - e^{-\bar{\sigma}(t_{j-1})}}{1 - e^{-\bar{\sigma}(t_j)}}\right) = \frac{e^{-\bar{\sigma}(t_{k-1})} - e^{-\bar{\sigma}(t_k)}}{1 - e^{-\bar{\sigma}(t_n)}}. \tag{D.27}$$

---

[3]The independence applies to whether a token changes from masked to unmasked. However, the specific unmasked token a masked token changes to depends on other dimensions.

For the Euler method, based on Eq. (D.13), we can derive that:

$$p(x^i_{t_{j-1}} = [\mathbf{M}]|x^i_{t_j} = [\mathbf{M}]) = 1 - \sigma(t_j)\frac{e^{-\bar{\sigma}(t_j)}}{1 - e^{-\bar{\sigma}(t_j)}}(t_j - t_{j-1}). \tag{D.28}$$

$$r_k = (\sigma(t_k)\frac{e^{-\bar{\sigma}(t_k)}}{1 - e^{-\bar{\sigma}(t_k)}}(t_k - t_{k-1})) \prod_{j=k+1}^{n} (1 - \sigma(t_j)\frac{e^{-\bar{\sigma}(t_j)}}{1 - e^{-\bar{\sigma}(t_j)}}(t_j - t_{j-1})). \tag{D.29}$$

For conditional generation cases where the generating sequence length $l$ is less than the dimension $d$, similar results hold. The only difference is that $N_k \sim \text{Binomial}(l, r_k)$ and Eq. (D.22) changes to

$$\text{E-NFEs}(n) = \sum_{k=1}^{n}(1 - (1 - r_k)^l). \tag{D.30}$$

Specifically, if we adopt a log-linear noise schedule and let $t_k = \frac{k}{n}$, according to Appendix D.4, the Euler method and Tweedie $\tau$-leaping method are equivalent. In this case, Eq. (D.27) simplifies to $\frac{1}{n}$. Substituting this result into Eq. (D.22), we obtain

$$\text{E-NFEs}(n) = \sum_{k=1}^{n}(1 - (1 - \frac{1}{n})^l) = n(1 - (1 - \frac{1}{n})^l). \tag{D.31}$$

# E    DISCUSSION FOR MEAN PARAMETERIZATION AND RADD

**Equivalence of modeling**    Analogous to the $x_0$ prediction in continuous state diffusion models, Austin et al. (2021) and Campbell et al. (2022) used the mean parameterization $\boldsymbol{\mu}_\theta(\boldsymbol{x}_t, t)$ to learn the the reverse density $p^i_{0|t}(x^i_0|\boldsymbol{x}_t)$, $i = 1 \cdots d$. According to the analytic form of reverse distribution in Eq. (D.8), letting $s = 0$, we have:

$$p^i_{0|t}(x^i_0|\boldsymbol{x}_t) = \begin{cases} p_0(x^i_0|\boldsymbol{x}^{\text{UM}}_t), & x^i_0 \neq [\mathbf{M}], x^i_t = [\mathbf{M}], \\ 0, & x^i_0 = [\mathbf{M}], x^i_t = [\mathbf{M}], \\ \delta_{x^i_0 x^i_t}, & x^i_t \neq [\mathbf{M}]. \end{cases} \tag{E.1}$$

This shows that the mean prediction is equivalent to learning conditional distributions on clean data. In conjunction with our discussion in Section 3.1, the mean parameterization should be time-independent, denoted as $\boldsymbol{\mu}_\theta(\boldsymbol{x}_t)$, and is equivalent to our reparameterized $c_\theta(\boldsymbol{x}_t)$. Empirical results like He et al. (2022) and Sahoo et al. (2024), which demonstrate that the time-independent model $\boldsymbol{\mu}_\theta(\boldsymbol{x}_t)$ performs well, also validate our theory.

**Equivalence of training objectives**    Shi et al. (2024) proved that the training loss for score parameterization (i.e., DSE loss) and mean parameterization (i.e., negative ELBO loss) are equivalent.

**Equivalence of sampling methods**    Comparing our Appendix D.2 with Shi et al. (2024); Sahoo et al. (2024), it is evident that the Tweedie $\tau$-leaping method for score parameterization is equivalent to the sampling method for mean prediction as follows:

$$q_\theta(x_s|x_t) = p(x_s|x_t, x_0 = \mu(x_t)) = \begin{cases} \text{Cat}(x_s; x_t), & x_t \neq [\mathbf{M}], \\ \text{Cat}(x_s; \frac{1-\alpha_s}{1-\alpha_t}\mathbf{e}_{[\mathbf{M}]} + \frac{\alpha_s - \alpha_t}{1-\alpha_t}\mu(x_t)), & x_t = [\mathbf{M}]. \end{cases} \tag{E.2}$$

Here, $\alpha_t$ represents the probability of a token remaining unmasked at time $t$, which equals $e^{-\bar{\sigma}(t)}$ for score parameterization. Therefore, Eq. (E.2) and Eq. (D.8) is equivalent.

# F    COMPARISON WITH CHEN ET AL. (2024)

Chen et al. (2024) proposes sampling methods for discrete-time and continuous-time diffusion models individually. Consider a sequence $\boldsymbol{x}$ of length $d$:

**Discrete-time models**    For models trained over discrete timesteps $\mathcal{T}_{\text{train}} = \{1, \cdots, T\}$, Chen et al. (2024) proposes to pre-sample the $d$ time points $\tau_i \in \mathcal{T}_{\text{train}}$, where each $\tau_i$ corresponds to a change timepoint for a specific dimension of $\boldsymbol{x}$. These timepoints form a time set $\mathcal{T}_{\text{change}} \subset \mathcal{T}_{\text{train}}$, and updates are only applied at these steps. For absorbing diffusion models, as each token changes only once, $\text{NFEs} = |\mathcal{T}_{\text{change}}| \leq \min(d, T)$.

**Continuous-time models**    For models trained over continuous time, $d$ change points $\tau_i$ are pre-sampled and sorted in ascending order ($\tau_{n_1} < \cdots < \tau_{n_d}$). Updates are sequentially applied at these points, resulting in $\text{NFEs} = d$, which resembles the sampling process in AO-ARM. However, how to reduce the NEFs to less than $d$ for the continuous-time model has not been investigated.

In contrast to Chen et al. (2024), which pre-sample specific time points and update tokens only at those predetermined points, RADD leverages a time-independent parameterization that updates tokens only when they change. This fundamental difference results in **different applicable scenarios** for the two methods:

- Chen et al. (2024) applies to both absorbing and multinomial diffusion models. However, for continuous-time models like SEDD, their sampling method results in $\text{NFEs} = d$ and, as noted, how to reduce the NEFs less than $d$ for the continuous-time model has not been investigated.

- RADD, on the other hand, is specifically designed for absorbing diffusion models. It is straightforward to apply the cache strategy of the RADD to continuous-time settings and reduce NEFs to less than $d$ because the input of the model is independent of the time.

Although the samplers in our paper and in Chen et al. (2024) originate from different formulations, the results of Theorem D.1 (Chen et al., 2024) for the discrete-time sampler align with those of our Eq. (3.4). However, as discussed above, the two samplers are applicable in different scenarios.

# G    Details of AO-ARMs

Any-order autoregressive models (AO-ARMs)  (Uria et al., 2014; Hoogeboom et al., 2022; Shih et al., 2022) model the joint distribution autoregressively for $d!$ different orders $\pi$ of the $d$ variables. Formally, the joint distribution is factorized as $\prod_{k=1}^{d} p(x^{\pi(k)}|x^{\pi(<k)})$ by chain rule. Therefore, AO-ARMs actually define $\sum_{k=0}^{d} C_d^k (d-k) = d2^{d-1}$ distinct univariate conditionals probabilities.

**Architecture**    AO-ARMs model all univariate conditionals via a weight-sharing neural network, by using the [**M**] token for variables that are not present in the condition set[4]. For efficient parallel optimization, the architecture is designed such that given the condition set of size $k$, it can predict all $d - k$ univariate conditionals at once, similar to the output of conditional distributions in Fig. 1.

**Training**    AO-ARMs are trained to minimize the negative joint likelihood of a datapoint $\boldsymbol{x}_0$ under the expectation over the uniform distribution $U_{\pi}$ of orders. It can be simplified by treating $l$ as a random variable with a uniform distribution over cardinalities 1 to $d$. Further, it can be transformed into a form for better parallel optimization:

$$\mathcal{L}_{AO}(\boldsymbol{x}_0) = \mathbb{E}_{\pi \sim U_{\pi}} \sum_{l=1}^{d} -\log q_{\theta}(x_0^{\pi(l)}|x_0^{\pi(<l)}; \pi) \tag{G.1}$$

$$= \mathbb{E}_{\pi \sim U_{\pi}} d \cdot \mathbb{E}_{l \sim U(1, \cdots, d)} -\log q_{\theta}(x_0^{\pi(l)}|x_0^{\pi(<l)}; \pi) \tag{G.2}$$

$$= d \cdot \mathbb{E}_{l \sim U(1, \cdots, d)} \frac{1}{d - l + 1} \mathbb{E}_{\pi \sim U(S_d)} \sum_{r=l}^{d} -\log q_{\theta}(x_0^{\pi(r)}|x_0^{\pi(<l)}; \pi). \tag{G.3}$$

Training pseudocode of Eq. (G.3) can be referenced in Algorithm 1.

---

[4]Condition set corresponds to variables in $x^{\pi(<k)}$.

**Sampling**  The sampling process for AO-ARMs generates data points autoregressively based on a randomly sampled order $\pi$. Starting from an empty sequence initialized with $[\mathbf{M}]$ tokens, the model iteratively predicts the next variable based on the current condition set $x^{\pi(<k)}$. Since the order $\pi$ is chosen randomly, AO-ARMs support generating sequences with any desired ordering, aligning with their ability to model $d!$ orderings during training. Sampling pseudocode can be referenced in Algorithm 2.

## H  COMPARISON TO PRIOR WORKS CONCERNING EQUIVALENCE DISCUSSION

Austin et al. (2021) and Hoogeboom et al. (2022) have both discussed the relationship between absorbing discrete diffusion and AO-ARMs. Below, we provide a detailed comparison of their approaches with ours.

Austin et al. (2021) made an early attempt to explore the connection between absorbing discrete diffusion and AO-ARMs. However, their work lacks rigorous proof. Instead, they qualitatively discuss the correlation between the two loss functions. Notably, in Appendix A.3 of Austin et al. (2021), they describe the relationship by stating, ***"this looks very similar ... it is not exactly identical."***

In contrast, our work rigorously establishes this connection. By leveraging the continuous-time framework and the time-independent parameterization presented in Theorem 1, we provide a formal proof demonstrating the equivalence between absorbing discrete diffusion and AO-ARM.

Hoogeboom et al. (2022) establishes the equivalence between ARMs and the ELBO of the absorbing diffusion models directly. In comparison, our approach follows a different path:

1. the ELBO was first reduced to $\mathcal{L}_{\text{DSE}}^T(\boldsymbol{x}_0)$ as Eq. (2.6), as discussed in detail in Lou et al. (2024).

2. Using step-by-step substitutions via Eq. (3.7), we further reduce $\mathcal{L}_{\text{DSE}}^T(\boldsymbol{x}_0)$ to $\mathcal{L}_{\text{AO}}$.

Therefore, our approach offers unique contributions by:

**A rigorous and alternative proof**  We leverage the time-independent properties of the reparameterization formulation, providing an alternative and rigorous proof of this equivalence.

**Equivalence of four losses**  Our analysis extends to demonstrate the equivalence of four distinct losses, including $\mathcal{L}_{\text{DSE}}^T(\boldsymbol{x}_0)$ and $\mathcal{L}_{\text{AO}}(\boldsymbol{x}_0)$ in Eq. (3.7), offering a deeper understanding of absorbing discrete diffusion.

**Comprehensive experimental validation**  We conduct a thorough study of these loss functions, with results presented in Tables 1 and 2. To the best of our knowledge, this exploration has not been explored in prior work.

## I  ALGORITHMS FOR TRAINING AND SAMPLING

---

**Algorithm 1** AO-ARM Training

---

**Require:** Network $\boldsymbol{c}_\theta$, samples from data distribution $p_{\text{data}}$
    **repeat**
        $\boldsymbol{x}_0 \sim p_{\text{data}}, \pi \sim U_\pi$.
        $l \sim U(1, \cdots, d)$.
        $\boldsymbol{x}' \leftarrow \mathbb{I}(\pi < l) \odot \boldsymbol{x}_0 + \mathbb{I}(\pi \geq l) \odot [\mathbf{M}]$
        Calculate $\mathcal{L} = \frac{d}{d-l+1} \sum_{i=l}^d -\log\left(\boldsymbol{c}_\theta(\boldsymbol{x}')[\pi(i), x_0^{\pi(i)}]\right)$.
        Calculate $\nabla_\theta \mathcal{L}$ and run optimizer.
    **until** converged

---

---

**Algorithm 2** AO-ARM Sampling

---

**Require:** Network $c_\theta$
    Initialize $x \leftarrow [\mathbf{M}] \dots [\mathbf{M}]$
    Sample $\pi \sim U_\pi$.
    **for** $i$ in $\{1, \dots, d\}$ **do**
        Sample $j \sim \text{Cat}(c_\theta(x)[\pi(i), \cdot])$
        Update $x^i \leftarrow j$
    **end for**

---

**Algorithm 3** Discrete Diffusion Training ($t$-DCE Loss)

---

**Require:** Network $c_\theta$, noise schedule $\sigma$, time $[0, T]$, samples from data distribution $p_{\text{data}}$
    **repeat**
        $x_0 \sim p_{\text{data}}, t \sim U([0, T])$.
        Construct $x_t$ by $\xi^i \sim Bernoulli(e^{-\bar{\sigma}(t)})$, $x_t^i = \mathbb{I}(\xi^i = 1)x_0^i + \mathbb{I}(\xi^i = 0)[\mathbf{M}]$.
        Calculate $L_\theta(x_t, x_0) = \sum_{x_t^i = [\mathbf{M}]} -\sigma(t)\frac{e^{-\bar{\sigma}(t)}}{1 - e^{-\bar{\sigma}(t)}} \log\left(\frac{e^{-\bar{\sigma}(t)}}{1 - e^{-\bar{\sigma}(t)}} c_\theta(x_t)[i, x_0^i]\right)$.
        Calculate $\nabla_\theta L(x_t, x_0)$ and run optimizer.
    **until** converged

---

**Algorithm 4** Discrete Diffusion Training ($\lambda$-DCE Loss)

---

**Require:** Network $c_\theta$, samples from data distribution $p_{\text{data}}$
    **repeat**
        $x_0 \sim p_{\text{data}}, \lambda \sim U([0, 1])$.
        Construct $x_\lambda$ by $\xi^i \sim Bernoulli(1 - \lambda)$, $x_\lambda^i = \mathbb{I}(\xi^i = 1)x_0^i + \mathbb{I}(\xi^i = 0)[\mathbf{M}]$.
        Calculate $L_\theta(x_\lambda, x_0) = \sum_{x_\lambda^i = [\mathbf{M}]} -\frac{1}{\lambda} \log\left(c_\theta(x_\lambda)[i, x_0^i]\right)$.
        Calculate $\nabla_\theta L(x_\lambda, x_0)$ and run optimizer.
    **until** converged

---

**Algorithm 5** Discrete Diffusion Sampling (Unconditional)

---

**Require:** Network $c_\theta$, noise schedule $\sigma$, time range $[0, T]$, step size $\Delta t$
    $t \leftarrow T$, $x_T \leftarrow [\mathbf{M}] \dots [\mathbf{M}]$, $c_{cache} \leftarrow c_\theta(x_t)$.
    **while** $t > 0$ **do**
        **if** Use Euler **then**
            Construct transition densities $p_{t-\Delta t|t}^{\text{euler},i}(x_{t-\Delta t}^i | x_t)$ by Eq. (D.13) with $c_{cache}$.
            $x_{t-\Delta t}^i \sim \text{Cat}(p_{t-\Delta t|t}^{\text{euler},i}(x_{t-\Delta t}^i | x_t))$ for all $x_t^i = [\mathbf{M}]$, $x_{t-\Delta t}^i \leftarrow x_t^i$ for all $x_t^i \neq [\mathbf{M}]$.
        **end if**
        **if** Use Tweedie $\tau$-leaping **then**
            Construct transition densities $p_{t-\Delta t|t}^i(x_{t-\Delta t}^i | x_t)$ by Eq. (D.8) with $c_{cache}$.
            $x_{t-\Delta t}^i \sim \text{Cat}(p_{t-\Delta t|t}^i(x_{t-\Delta t}^i | x_t))$ for all $x_t^i = [\mathbf{M}]$, $x_{t-\Delta t}^i \leftarrow x_t^i$ for all $x_t^i \neq [\mathbf{M}]$ .
        **end if**
        **if** $x_{t-\Delta t} \neq x_t$ **then**
            $c_{cache} \leftarrow c_\theta(x_t)$.
        **end if**
        $t \leftarrow t - \Delta t$.
    **end while**

---

---

**Algorithm 6** Discrete Diffusion Sampling (Conditional)

---

**Require:** Network $c_\theta$, noise schedule $\sigma$, time $[0, T]$, step size $\Delta t$, Prompt spaces $\Omega$ and tokens $\mathcal{T}$.

    $t \leftarrow T$, construct $\boldsymbol{x}_T$ with $\boldsymbol{x}_T^\Omega = \mathcal{T}$ and $\boldsymbol{x}_T^{\bar{\Omega}} = [\mathbf{M}]$, $\boldsymbol{c}_{cache} \leftarrow \boldsymbol{c}_\theta(\boldsymbol{x}_t)$.

    **while** $t > 0$ **do**

        **if** Use Euler **then**

            Construct transition densities $p_{t-\Delta t|t}^{\text{euler},i}(x_{t-\Delta t}^i|\boldsymbol{x}_t)$ by Eq. (D.13) with $\boldsymbol{c}_{cache}$.

            $x_{t-\Delta t}^i \sim \text{Cat}(p_{t-\Delta t|t}^{\text{euler},i}(x_{t-\Delta t}^i|\boldsymbol{x}_t))$ for all $x_t^i = [\mathbf{M}]$, $x_{t-\Delta t}^i \leftarrow x_t^i$ for all $x_t^i \neq [\mathbf{M}]$ .

        **end if**

        **if** Use Tweedie $\tau$ -leaping **then**

            Construct transition densities $p_{t-\Delta t|t}^i(x_{t-\Delta t}^i|\boldsymbol{x}_t)$ by Eq. (D.8) with $\boldsymbol{c}_{cache}$.

            $x_{t-\Delta t}^i \sim \text{Cat}(p_{t-\Delta t|t}^i(x_{t-\Delta t}^i|\boldsymbol{x}_t))$ for all $x_t^i = [\mathbf{M}]$, $x_{t-\Delta t}^i \leftarrow x_t^i$ for all $x_t^i \neq [\mathbf{M}]$ .

        **end if**

        **if** $\boldsymbol{x}_{t-\Delta t} \neq \boldsymbol{x}_t$ **then**

            $\boldsymbol{c}_{cache} \leftarrow \boldsymbol{c}_\theta(\boldsymbol{x}_t)$.

        **end if**

        $t \leftarrow t - \Delta t$.

    **end while**

---

# J  EXPERIMENTAL DETAILS

## J.1  MODEL DETAILS

We implemented our RADD model based on the SEDD architecture, an decoder-only transformer model (Vaswani et al., 2017b; Devlin et al., 2019a). Our model incorporates rotary positional encoding (Su et al., 2021) but excludes all parts related to time conditioning (i.e., TimeEmbedding, adaLN-zero block (Peebles & Xie, 2023b)). Instead, we added a softmax operation at the end of the neural network to ensure the output is a valid conditional distribution. This simplified architecture is similar to the standard GPT architecture, except the lack of attention mask and multiple probability output instead of single one.

## J.2  TRAINING DETAILS

We trained our RADD models using the following configuration settings:

- Batch Size: 512
- Learning Rate: $3 \times 10^{-4}$
- Exponential Moving Average (EMA):0.9999
- Gradient Clipping: Gradient norm clipped to 1
- Warmup Schedule: Applied for the first 2500 iterations
- weight decay: 0.03
- dropout rate: 0.02

The hyperparameters were adapted from (Lou et al., 2024), with modifications referenced from (Shi et al., 2024). The main modifications were setting the weight decay to 0.03 and the dropout rate to 0.02. Due to limited computational resources, we did not perform a hyperparameter search and directly conducted experiments with these settings. Further tuning of hyperparameters may enhance performance.

In terms of training tokens, 400K iterations correspond to approximately 105 billion tokens, while 1000K iterations correspond to about 262 billion tokens. It's worth noting that the entire OpenWebText dataset contains only 9 billion tokens, meaning that models went through the dataset multiple times during training.

The small models are trained on nodes of 32 V100 32G GPUs with float16 precision.

## J.3 Unconditional generation details

For unconditional generation, we employed a log-linear noise schedule. As illustrated in Section 3.2, the Euler method and the Tweedie $\tau$-leaping method are equivalent under this case. In practice, the implementation of the Euler method and the Tweedie $\tau$-leaping method remains the same for RADD but differs for SEDD. So we measure the perplexity of SEDD by Tweedie $\tau$-leaping method which performs slightly better while it suffices to measure the perplexity of RADD once.

As suggested by (Zheng et al., 2024), except for Table 4, all samples are generated using float64 precision of Gumbel-based categorical sampling(abbreviated as fp64 precision below). No annealing methods (e.g., top-p or top-k sampling) were applied in our sampling process.

## J.4 Further evaluation of generative perplexity

**Runtime and entropy measurement** To evaluate the efficiency of inference and the diversity of samples, we assessed the inference time and unigram entropy averaged across 1024 samples. When calculating unigram entropy, we chose the natural logarithm (ln) instead of the $\log_2$. We provide perplexity and entropy results under both fp64 and fp32 precision in Tables 3 and 4 respectively.

Under both precisions, RADD and SEDD exhibit comparable perplexity results for the same number of sampling steps. For large sampling steps, perplexity converges under fp64 precision but continues to decrease under fp32 precision. This discrepancy is due to precision errors in fp32, which effectively function as a form of annealing, resulting in deceptively lower perplexity values (Zheng et al., 2024). To evaluate efficiency, we focus on the sampling time under fp64 precision.

The RADD model consistently required the shortest sampling time while maintaining similar perplexity levels to the SEDD model. Specifically, RADD achieved a speed-up of up to 2.5 to 3 times with large sampling steps, as shown in Table 3. These findings align with the analysis of the E-NFEs in Fig. 1a, validating the effectiveness of the RADD model and the caching strategy. Even with 1024 sampling steps(equal to sequence length), the cache strategy still enables about 1.5 times acceleration.

Table 3: **Average inference time, perplexity, and entropy per sample with varying sampling steps under fp64 precision.** The table compares the average inference time (in seconds), perplexity (PPL), and entropy for the SEDD medium model using Tweedie $\tau$-leaping sampling methods, as well as the RADD medium model under a log-linear noise schedule with a caching strategy. The experiment is conducted under a single NVIDIA A800 80G GPU with a batch size of 16.

|  | Steps | 32 | 64 | 128 | 256 | 512 | 1024 | 2048 | 4096 |
|---|---|---|---|---|---|---|---|---|---|
| SEDD-medium | Time(s) | 0.52 | 0.99 | 1.91 | 3.78 | 7.49 | 14.93 | 29.82 | 59.56 |
|  | PPL↓ | 159 | 113 | **94** | **87** | **84** | 86 | 82 | **81** |
|  | Entropy | 8.26 | 8.18 | 8.14 | 8.12 | 8.09 | 8.10 | 8.09 | 8.07 |
| RADD-medium | Time(s) | 0.45 | 0.80 | 1.54 | 2.96 | 5.32 | 8.39 | 12.28 | 18.20 |
|  | PPL↓ | **158** | **113** | 96 | 89 | **84** | 83 | 81 | 81 |
|  | Entropy | 8.28 | 8.21 | 8.18 | 8.15 | 8.14 | 8.12 | 8.13 | 8.13 |

Table 4: **Average perplexity, and entropy per sample with varying sampling steps under fp32 precision**. The table compares the average perplexity (PPL), and entropy for the SEDD medium model using Tweedie $\tau$-leaping sampling methods, as well as the RADD medium model under a log-linear noise schedule with a caching strategy.

|  | Steps | 32 | 64 | 128 | 256 | 512 | 1024 | 2048 | 4096 |
|---|---|---|---|---|---|---|---|---|---|
| SEDD-medium | PPL↓ | **125** | 85 | **62** | **51** | **41** | 34 | **27** | **22** |
|  | Entropy | 8.18 | 8.07 | 7.96 | 7.86 | 7.73 | 7.60 | 7.44 | 7.25 |
| RADD-medium | PPL↓ | 126 | **84** | 63 | **51** | **41** | 33 | **27** | **22** |
|  | Entropy | 8.19 | 8.09 | 7.98 | 7.88 | 7.75 | 7.59 | 7.45 | 7.27 |

**Efficiency of our caching strategy with mini-batch**    In Appendix D.5, we explored the average NFE for the single sample case. This concept, however, extends to the mini-batch case.

In a mini-batch, some samples may remain unchanged while others may evolve. To address this, we use a dynamic batch-size strategy. Only the samples that have changed are passed through the neural network for computation. While this still involves a "batch-level NFE", the total NFE per sample is reduced, effectively enhancing efficiency as in the single-sample case.

In comparison, concurrent work Sahoo et al. (2024) does not consider dynamic batch size in their implementation, so their practical acceleration falls short of the theoretical potential.

To further validate the efficiency, we conducted experiments generating 64 samples under various batch sizes. The results, summarized in Table 5, demonstrate that the average generation time with our caching strategy consistently outperforms SEDD across different batch sizes and timestep configurations. We provide a detailed analysis below:

BATCH SIZE AND GPU UTILIZATION    For a fixed model, increasing the batch size leads to improved GPU utilization before reaching the maximum batch size, reducing the average sampling time per sample. In the case of RADD, sampling time decreases significantly as the batch size increases from 1 to 4. However, the reduction in sampling time becomes minimal beyond a batch size of 4, indicating that GPU utilization has nearly reached its maximum capacity at this point. This demonstrates the scalability of our approach within the limits of the hardware's parallel processing capabilities.

 SEDD VS RADD    Under identical batch sizes and timesteps, RADD consistently outperforms SEDD in sampling speed. This highlights the efficiency of RADD's design, where the caching mechanism reduces redundant computations and achieves faster generation, especially for larger batch sizes and higher timesteps.

Table 5: **The average inference time across batch sizes and timesteps of SEDD-medium and RADD-medium.** Experiments were conducted on an NVIDIA 4090 GPU (24GB VRAM, maximum batch size = 8). The average generation time per sample (in seconds) is averaged over 64 samples.

| Batch Size \ Steps | 32 | 64 | 128 | 256 | 512 | 1024 | 2048 | 4096 |
|---|---|---|---|---|---|---|---|---|
| SEDD-medium | | | | | | | | |
| 1 | 0.98 | 1.83 | 3.57 | 6.95 | 13.80 | 27.16 | 54.76 | 109.90 |
| 2 | 0.75 | 1.43 | 2.77 | 5.46 | 10.85 | 21.62 | 43.19 | 86.23 |
| 4 | 0.67 | 1.30 | 2.56 | 5.09 | 10.15 | 20.27 | 40.50 | 80.97 |
| 8 | 0.70 | 1.34 | 2.65 | 5.25 | 10.46 | 20.85 | 41.68 | 83.32 |
| RADD-medium | | | | | | | | |
| 1 | 0.77 | 1.34 | 2.56 | 4.86 | 8.73 | 14.00 | 20.45 | 30.70 |
| 2 | 0.60 | 1.08 | 2.05 | 3.98 | 7.32 | 12.32 | 18.88 | 28.97 |
| 4 | **0.50** | **0.97** | **1.87** | **3.65** | **6.75** | 11.26 | 17.67 | 27.67 |
| 8 | 0.51 | **0.97** | 1.90 | 3.71 | 6.76 | **10.87** | **16.76** | **26.58** |

**Sampling as any-order autoregressive models**    As outlined in Theorem 1, $c_\theta$ can be interpreted as a conditional distribution over clean data. One natural approach is to use this directly for generating samples, similar to any-order autoregressive models. However, there are $d!$ possible ways to decompose the joint distribution into conditional distributions. We tested three representative cases:

- forward: $p(x^1 \cdots x^d) = \prod_{k=1}^{d} p(x^k | x^{(<k)})$
- backward: $p(x^1 \cdots x^d) = \prod_{k=1}^{d} p(x^k | x^{(>k)})$

- random: $\pi \sim U_\pi$, $p(x^1 \cdots x^d) = \prod_{k=1}^{d} p(x^{\pi(k)}|x^{\pi(<k)})$

The results are presented in Table 6. Perplexity was calculated as the average over 1024 samples. For the random case, we calculated the average perplexity across different randomly generated $\pi$, corresponding to the standard AO-ARMs sampling method. It shows that the result of the standard AO-ARMs sampling method aligns closely with the converged perplexity of the $\tau$-leaping method under fp64 precision in large steps. Among the different decomposition orders, the forward order demonstrated the best performance.

Table 6: **Quality of unconditionally generated text evaluated by perplexity ($\downarrow$).** For a fixed model, the best perplexity is **bolded**.

| Method | RADD-medium |
|---|---|
| Forward | **81.70** |
| Backward | 103.68 |
| Random | 83.10 |

### J.5 FURTHER EVALUATION OF ZERO-SHOT PERPLEXITY

In this section, we provide an extended evaluation of the zero-shot language modeling performance of RADD models trained for **1000k** iterations. While the results in the main text focus on models trained for 400k iterations, training for longer durations can slightly improve model performance due to the increased exposure to training data.

Table 7: **Additional zero-shot language modeling perplexity ($\downarrow$) for RADD small models.** We present the perplexity for RADD models trained for **1000k** iterations based on their corresponding loss.

| Method | LAMBADA | WikiText2 | PTB | WikiText103 | 1BW |
|---|---|---|---|---|---|
| RADD-$t$-DCE | **48.92** | 37.44 | 102.49 | 37.20 | 70.58 |
| RADD-$\lambda$-DCE | 49.74 | 37.13 | **98.84** | 36.66 | **69.77** |
| RADD-AO | 49.43 | **36.86** | 102.36 | **35.25** | 70.71 |

## K ADDITIONAL EXPERIMENTAL RESULTS

### K.1 ADDITIONAL SAMPLES

Additional unconditionally and conditionally generated text of RADD-$\lambda$-DCE small and medium models are reported in Figs. 2 to 5. All of the samples are generated with 1024 steps under a log-linear noise schedule.

Civilians of Puerto Ricans in Ohio. Chili replaced beer peas and oyster shs as Kurds replaced corn with Louisiana barbeque in Louisiana, where chili served as the dinner dish to houseplants: "Cajaro souvlaki".

In the southwest in the early-19th century, this dish became used in the Texas Southwest.[1][12]

When the "American" to mean "Spanish F" or "Saw" arose in the thought, first in the south of Mexico (Columbus) and later in America (continent), chili served as, When cooked, medicinally mixed into dishes as a side dish, with fruits and vegetables, short stewed meals, soup to date including poker pot.

Wackned chili also first appeared in North America in 1819 as some accounts describe its use as a turkey stew and chicken hock". Other versions, which are known of by euphemism as the sussificatte guitaristi, have pinned the origin of the term on the original date of the Passiacoamerica during which Spanish tribesmen could not venture into the region until (after the end of the Pequewille; jumples mugged and smoking was observed) Bigger's Passage upon the Coast. Some researchers believe that chili is served as regent proving in a nice winter as part of the traditional diet, not barbecue although the Uuy Juaco is traditionally eaten.[13][14]

The Mexican website Chili, which first occurred in 2006, contains a short page reliving 15 different barbecue recipes already served in Mexico under Brandon and Cincy of the American revolutionary movement of the 19th.[15][18] Chili became a frequentstay of Mexican dining at the Walton, Colo., family grocery store in California and North Carolina, and it has been found in several American grocery stores. Original scout Midwest founder Michael Dunbar declared in 1998 that this corrugated chili dish was "consistent with the food that served in Mexico during the 19th century."[16]

End of era and availability

The Mexican Chili became regularly served in United States Central Texas as of November 1, 2004 (December 1934), and has been sold since at Chestnut many times.[17] Many Mexican barbecue joints across the West Valley began as a cuisine instead of a regular and later would use chili, rather "porocera cooking by returning Mesopotamians."[18][19][20][21]

In culture [ edit ]

Chili was a coarse cured dish made with spices and seasoning, usually chili or side chili. Mexican meat was assigned to the game.Wet meats were made using heat, followed sometimes with hot beer or wine, such as limes or beer. Stroked chili keeps spaghetti sauce or breast dishes such as parmies and cheese. Chili was used to warm flatbread, hot dogs, lamb jerky and dried moonplugs. It was also used in hamburger, except where cuts of meat is used in meal. The dish was also used in toppdown, fried chicken, egg rolls, papaya noodles, yakka noodles, garlic, and onions. Chili is used in wine batter, unsweetened spaghetti squash which is often mashed together in sandwiches.

The Mexican Grill locations are offered at Kearney, Purdue University, we travel to other Inc. restaurants either with affordable prices or at semi-cooperative/voluntary locations. Stores range from California to Texas. Chili can also become available from the local Italian market, often as available at the Chili Jack's restaurant on the Black Hawk Courthouse.[22] Chili is not currently produced for local Italian producers.

New Mexican imports [ edit ]

A new emphasis of use of arugared Mexican beans, as well as black beans, BL beans, pickled peppers, quesadillas, and peas, form them as a menu item alternatives to items as manufactured by KFC and Mexican Grill.

The Chili is popular because of tortilla[19] which is a regular ingredient with sandwiches between Sundays. According to Google Live-Mexico Taco, making fresh, chili-free meals includes pulled pork with cheese removed sweet potatoes, ranch oranges, and mild marlic cheese with floured brown bread. Guajías, chillies, and during cold periods are also included in Mexican Chili. Caesar salad is featured in many facets of East Mexican Grill. Bay Area El-Lachi, Taco Bell employee and blogger, mentions lunch menus. Options, the better bets, were the traditional winter chicken dishes on Fridays in the Hudson Brewery that call the restaurant proper chili.[21]

Figure 2: **Unconditionally generated text of RADD-$\lambda$-DCE small.**

I have a  " Kot actu my vit " Shit " est " thou est first and me first " Work Begins By email: 10 January 2012 7:21 GMT Kristy

Latest article at the "Mail Stream page

Impressions

Because I am still relying on the e-mails of the EIS server to indicate legitimate databases, I fail the virtual sharing argument, share function, disk space index, etc. (Failure is not sufficient to have my mail client be contains privileged information....) once I set the transport destination on the email e-mess, it contains privileged information extra.

My e-mails list which connections have received data, and you can use RSS to view the packets. In an application with a connection infrastructure you can create an arbitrary object structure, for which and for the system where layers of your communications stack are designed.

" enforce kernel space mass. append first 2. :y ;; 0. cons. end eq bd. nec. end df1 lift8d 1 foo 8 11 modified-scwd0h 1 ls 120 127 dbbsi 1 dbbsi 1 check 503(123-s3)p #syntkilledestpcoud" The client also throws back an IP address hash that can easily be cached right after it originally generated; low-level operations run by microservices.

Using bodyURL parameter, I run a privilege test.

After a reasonable approximation of Oracle's best RSA algorithm, the result is of RCELOG assuming the assertion that this method works all  1 n of trails  [day 11 - day left] + job from 5-18 and last for 9 after J.T. Leeper's Theorem (which then contradict the thesis that a cleanup bug may exist). I couldn't resist finding that, if it did not, needs to be fixed in a ASAP in order for exploit.

A better example to go of is using an IA-64 firewall in Server Enterprise. The result can enable you to continue leveraging Server Enterprise's firewall design, but not ignoring the same. Remember, if you want to enforce Server Enterprise's anomaly control, you need how it seems to integrate with both functionality and logic.

One last reference to the DBTR leak we first found in the world in our focus on Server Enterprise is that a client application can not query physical storage systems. The incompatibility of default logic with lightswitch is not of a subject per se restriction. Alten can benefit from impertional clanchism, therefore worth keeping in mind that the solution also includes both Java EE applications.

Subsidizing Server Identity

A collaborative research project specializing in server authentication is as important as it is to the user base; it is easy for ORM and AD tools to handle authentication from each Server without an extensive need, external to which server.

To have only one authentication stickler running an instance, and loading a form authentication service that joins one server with a live e.g.i. justifiable sex, a declarative, composable-by-side Expanded Persistent Application ( ERP) class is brought about.

In general, it tends to only need two IDs in an application approaching RHEL.

InJh mail company distributed in New York City.'s email system, with 6 ID., This application should also deal with issued data model, ERP (e.g. data model proxy (0.), database (0.6.) boundaries.[ 10]

The majority is typical chief operating in order to accomplish a technical milestone, and [ * ] is kept calm by the development team [ 11]. As a result, being a ready-track application, this open consistencyGateway is very much ready.

A fuzzed server data file with sheltered and forstrap A per day routines with messages such as response, reply, reply, excluded is reserved for applications... that contain SQL. A server file can be deployed over two machines under the requirements of the application, such as building windows of good email for long time, to simulate simple third-party software for email-mirroring, or for slab email for messaging. It is also settled that an application requiring isolation test passes must have an isolated SQL server resources, which are typically lightweight enough by default (tight door if you're) to manage.

Configuring Environments standard ( R2. R2 is a practical "dry" multi-purpose instantiation mechanism, but it still recieved different designations using knctions that contain module mechanism (where or wherever), i.e. only module will be (elegally) it can be harness, but in the context supporting the interoperability, not every application must implicitly adopt only one and allows supporting the similarity of the particular implementation. Why do it use R names, but the usual reason is that: it's a database that is accessed in a joint framework.

Figure 3: **Conditionally generated text of RADD-$\lambda$-DCE small.** Prompt tokens are in blue.

Hall of Fame quarterback to Pro Bowlist. At times, Wright hasn't hit this reset button—he had a recommittal with the 2011 regime, but he certainly is on notice: Who'd rather be unemployed starring tennis all-stars than a 500-castle NFL franchise? Is the experience a perfect match for him? Can he go over Suh, who is coming off a Super Bowl championship season that included a major letdown expected him to just discuss: a torn ACL?

Derrick Williams vs. former Nebraska QB Kevin Youkil at the University of Miami: "Bright warm-up. Become captivated by the game in right light and suppose, at both the amateur and top-level level Brandon Pettit came into consideration from visionary standpoint. Third thought: I revered Robb Hart's job as offensive coordinator while as an assistant coach at Iowa State. Let Jimmie Langer tell his story."

Ricky Duches, former Ohio State coordinator, defeated new LSU offensive line coach Les Miles with high school Nebraska quarterback Willy Clabo. (In the movie from Boston, Katerla Floodgate also got the Lombardi Trophy.)

Williams vibes as he recalls his father and—argenceably he's been better physically and at other things—reveals just how big-time football is. And what he received for being so synonymous with the game: "Back in 1982, when I was 16, I saw the big-time football. I knew what the NFL was...I got to Superdome, and out I knew Jim Brown wasn't exactly greeting Nogueira with a being handshake. But I saw it! [When] else got out on the field, I stood there half-swedicking...We liked it for a little while." As Al Crump, the experienced conference speaker, was truly dazzled in a ballroom, tells me of her visit, "We've been in the group for years. I've been pretty carefully planning get a visit. There were Seahawks and could I say badgers? While suffering." Nineteen-year-old Husker's fan saw it as a chance to make her own observations amidst the relentless publicity during the late-May 26 offsite disruption.She's discovered better than most what other attendees have done when it comes to integrity and candor in their feats with interviews, and bloggers featuring her accounts of the Washington Redskins and Chuck Siedelbaum. Beginner loves groupies.

Sgt. Gun Casper Olson, free agents with San Diego at Faulkner: "I got to really get to know my brother, Buck (Olson, Sr.). We're a supervisor in the Major League for the last 15 years, who will work with us in the $159 million and $10 260$ per billion level, so we got to get to know him every single day. Came to see who often begins and ends up staying where professional sports are. Hawaii didn't offer it as Kansas did." However, in many ways, he was highly complimentary of that former team's coach, comparing what Hawaii offered to Voeltt Park whom he described as Kurt Stater of the Eagles: "I got to see Cliff Starsy, North Carolina's head coach at the time, and watch Nick Saban, who was the potential replacement for the Alabama coaches before they got his hands on Bradley/Hedge Briles. I watched everything. On the football field was his ono. As a coach, he gets so excited, I couldn't drive with him on the field, though he wasn't playing. His closest friend manages TSA for him. And our lineback players'm ht excited that he puts on. I want to know where Robert Griffin III would be next because he has reached out to Matt directly. We are so intense that my drive won't much use."

Chris Jackson, Jr., Jr., 2015 Stanford and Paula, who were then older than him and split chances are part of several big families. Jackson's momma; Paula's soft loves; and Jackson's pressure on everything in the team and social sphere. Still, they are each given enough space to talk about the game ultimately one day. On Monday, for her first hour-long conversation as a prospect with people wearing the surrogate for best mothers, she talked of her relationship with the ninth-grade scholarship and converting it into an effective recruiting tactic; essentially, she had won her family's trust with linebackers that she could stage for a bright future. "And that shows true faith and confidence," she informs me.

Orryn Lamba Jackson, 2015 sophomore in Cornell, an African-American computer geek with giant L.D. glasses, named Abraham Lincoln to Sanford Abraham, the mayor of her community in Benton, Ark., that school's throwback. She is as humble as Los Aldridge, where her book was held

Figure 4: **Unconditionally generated text of RADD-$\lambda$-DCE medium.**

I have a ? Well, I don't really think I leverage capital at the end of the day. I have a - a theory that these things are Fiction- ignorable and dangerous.

Huh... Hmm.. But they will never recognise me.

How fierce am I.. Ma Rdf, called me out in the heat of sanctimonia and ignorance.

I am in a position of being in the middle, not the pack.

Primordial wealth has no ample hands, but I am especially heavy. My first step is not gonna be rich.

As much as I claim, I will provide you with nothing. I will force you to do whatever it take to go the route that you deem appropriate. I'll demand that you become my expert tool.

Usually when they gangge me, I know when their indulgence is not for me.

You´re practicing WITH the Masochist!

A young lady took Petra wrapsrt me a strawberry extract while he was thirsty as I was eating out that day.

She bragged as she hugged the unsuspecting waitress to tell him, "Let's revue that recipe! That meth thumb isn't good." Besides, she said, "So friggin' gums don't mong like that cure som a bowl!"<|endoftext|>"liars stalling liars." The Town Movement Leaders Still Nader Meanwhile?

On November 9, a new report [partedially co-written The Environmental Working Group] revealed allegations that the EPA had routinely used decoys on pigs in Oregon today, exacerbating Ecolabelion in U.S.-origin food.

Let's face it - it is time begged little children to form the cover of the smoke is inferno and crime malls - no matter how likely it is for them to say something funny or remember their fathers mumbling with far more reluctance when addressing the press. So we expect our government agent to be moral when they have a (public) right to information without leaving the door open to their investigation

BILL May they learn our government officials needn't superlative skills in which case they take a tough line...One reason all reported tests are said to be acts of terrorism by corrupt agents of the political is you've never reached a threshold level you must accept testing. So there are two reasons why?

One is that the public - public opinion has changed recently. Takes time, and no public pressure won't be meaningful to algae farms when the bloggers get down to shove. We were honestly assured of all of this pressure on the way-down to Nixon - tell us if you are willing to listen to a sly reporter...The folks you Daily Sheeple have the most difficult time doing is fishing crackers to gather sand with hundreds of pounds of scales, concrete, or grease trays.

* * *

Related: Bringing Up:

Earlier: Former EVA Does "Nader's EPA" Test In November, Parrotes Daily Exam

Earlier: Mining Smog's Seed Means Persuating Boger Ecolideca

Photo Credit: Biolanao/Whisper<|endoftext|>Macroorganics, published in Environmental and Swedish Medicine, have detected an inter-cell growth factor-alpha (IFAS-alpha) gene in what the team considers to be the most tasty and dismaying organism and object in the world. The researchers studied the diet of 60 million modern sheep, goats, sheep, cattle, chickens, rats and mice whose accumulate more data revealed patterns of greater importance to be found in the human population. Specifically a gene (IFAS-alpha) disproportionately responds in mice to oxidative stress.

Rheum-enriched flavon fat (AST) and consumption of milk fat-oddimal fractional density lipoprotein (FFL) were indeed shown to have significant effects. Medication of these substances in control animals or controls did not induce the selected IFAS-alpha. colon cancer rates, in contrast with the original cause in antibiotic- and non-infected control mice that underlie the principle, were not necessarily improved.

In turns antibiotic resistance and colonally carcinogenic antibiotic feeding bacteria are not related. Numbers are unclear but the fact these changes are attributed to them proposes that conventional antibiotics could not deny known causal effects on the adjacent cousins.

The Swedish researchers led by Professor Erik Johansson and Anne Kristin Gunnarsson-Silk both University of Sweden who were elected to chair the "Global VIP Conference on Animal Food Physiology" today on May 1 in Vilskovisno, shared by the Office of Ontrialed CounsellingASE MEDIC; the high-level Swedish Food Safety Research Institute; Division of Borneck, Tissue, Cellular and Virus; System Genomics Service that operates in a joint framework.

Figure 5: **Conditionally generated text of RADD-$\lambda$-DCE medium.** Prompt tokens are in blue.

