# OpenReview forum: "Your Absorbing Discrete Diffusion Secretly Models the Conditional Distributions of Clean Data"
_ICLR.cc/2025/Conference — ICLR 2025 Poster_

### Official Review · Reviewer_4Udp · 2024-11-01

**Soundness:** 3
**Presentation:** 3
**Contribution:** 3
**Rating:** 6
**Confidence:** 5

**Summary:**

The paper introduces a reparameterized absorbing discrete diffusion (RADD) framework, which reinterprets the concrete score of a masked diffusion process as a conditional distribution on the clean data. This approach allows for modeling the backward process independently of the timestep t, significantly reducing the number of function evaluations required during sampling. Additionally, the paper establishes a connection between discrete diffusion and any-order autoregressive models, offering greater flexibility in choosing the training loss, sampling procedure, and likelihood evaluation.

**Strengths:**

The paper presents a theoretically sound method for reframing score entropy discrete diffusion as a denoising prediction. The insight drawn from its connection to any-order autoregressive models is particularly valuable. Furthermore, the experimental results provide strong support for the claims made in the paper.

**Weaknesses:**

Most of the theoretical results in this paper align with concurrent and prior research. For instance, time-independent parameterization is also discussed in [1]; t-DCE and tau-CDE are introduced in [1] and [2], with [2] additionally proposing a caching trick to enhance sampling efficiency.

Moreover, the connection between masked diffusion and any-order autoregressive models is established in [3, Appendix C]. It would be appropriate to give proper credit to that work.

Nevertheless, I believe this paper offers valuable contributions and insights, with experimental results that strongly support its claims.

[1] Shi, Jiaxin, et al. "Simplified and Generalized Masked Diffusion for Discrete Data." *arXiv preprint arXiv:2406.04329* (2024).

[2] Sahoo, Subham Sekhar, et al. "Simple and Effective Masked Diffusion Language Models." *arXiv preprint arXiv:2406.07524* (2024).

[3] Hoogeboom, Emiel, et al. "Autoregressive diffusion models." *arXiv preprint arXiv:2110.02037* (2021).

**Questions:**

The parametrisation of the any-order auto-regressive model (AO-ARM) is unclear to me. Given a denoising model p_\theta(x_0 | x_t) trained with the t-DCE loss for instance, could it be reparametrised as an AO-ARM? It would be helpful to include a seesion in the appendix discussing the parametrisation of AO-ARM and to provide both the training and sampling algorithm of AO-ARM in appendix F.

---

> ### Author Response · Authors · 2024-11-22
> **Response to Reviewer 4Udp**
>
> Thank you for your thoughtful feedback and for recognizing the contributions and insights of our work.
>
> ## W1: Related work
>
> Thanks for pointing out the related work, we will add the following discussions in the revision.
>
> ### W1.1: Is RADD's time-independent parameterization unique?
>
> We clarify that our time-independent parameterization is a **unique theoretical contribution** of our work. Specifically, **the parameterization in [1\*] remains time-dependent, as explicitly shown in their Equation (11)**. In contrast, our approach simplifies the expression to its most fundamental form, completely removing the explicit dependency on $t$. For further details, please refer to the response to Common Concern 3.
>
> Additionally, [2*] lacks a theoretical understanding and rigorous analysis for removing the time condition and the caching trick.
>
> For a more detailed comparison and discussion with [1*] and [2*], please see Lines 486 to 507 of our submission.
>
> ### W1.2: Comparison to prior works concerning equivalence discussion
>
> We acknowledge that [3*] establishes the equivalence between ARMs and the ELBO of the absorbing diffusion models, corresponding to the $\mathcal{L}_ {\text{DSE}}^T(\boldsymbol{x}_ 0)$ and $\mathcal{L}_ {\text{AO}}(\boldsymbol{x}_ 0)$ terms in Eq. (3.7) of our submission. However, our approach offers unique contributions by:
>
> - Leveraging the time-independent properties under the reparameterization formulation, which provides an alternative proof of this equivalence.
> - Extending the analysis to demonstrate the equivalence of **four losses** including $\mathcal{L}_ {\text{DSE}}^T(\boldsymbol{x}_ 0)$ and $\mathcal{L}_ {\text{AO}}(\boldsymbol{x}_ 0)$ in Eq. (3.7), deepening our understanding of absorbing discrete diffusion.
> - Conducting a comprehensive study of these losses, presented in Table 1, which has not been previously explored before to our knowledge.
>
> Please see more details in the common concern 2.
>
> ## Q1:Parametrisation and Algorithms of AO-ARM
>
> Yes, the denoising model trained with diffusion loss can be reparametrised as an AO-ARM. We have included a seesion in the Appendix of the revision to discuss the parametrisation and algorithms of AO-ARM for better understanding.
>
>
> [1*] Shi, Jiaxin, et al. "Simplified and Generalized Masked Diffusion for Discrete Data." arXiv preprint arXiv:2406.04329 (2024).
>
> [2*] Sahoo, Subham Sekhar, et al. "Simple and Effective Masked Diffusion Language Models." arXiv preprint arXiv:2406.07524 (2024).
>
> [3*] Hoogeboom, Emiel, et al. "Autoregressive diffusion models." arXiv preprint arXiv:2110.02037 (2021).

---

> > ### Comment · Reviewer_4Udp · 2024-11-25
> >
> > Thanks for your detailed response.
> >
> > Regarding W1.1, I appreciate the connection established between ratio parameterization (i.e., SEDD) and time-independent parameterization. I agree that this is a unique theoretical contribution. However, the theorem does not introduce a fundamentally unique improvement. Specifically, using the theoretical results from [1*, 2*], one could also parameterize a time-independent model.  But again, I still find the proposed theorem both unique and interesting, and I really appreciate it.
> >
> > Regarding W1.2. thanks for the clarification. Indeed, the comparisons on four different losses are great contribution.
> >
> > One more question regarding your answer to Q1. I appreciate the new discussion about AO-ARM in the appendix. When sampling from AO-ARM (see alg2), do you strictly follow that sampling order first and then denoise, or do you unmask all tokens first and then remask? I think these two methods make both sense, but the last one is more efficient. I am curious about your implementation and it would be great to further clarify it in the paper.

---

> > > ### Author Response · Authors · 2024-11-25
> > > **Further Clarification**
> > >
> > > We greatly appreciate your thoughtful feedback and acknowledgment of our contributions! Below, we address the remaining concerns in detail:
> > >
> > > - **Further clarification on W1.1**: In both theorem and experiment, [1*] takes $t$ as input of the neural network, and [2*] only conducted an ablation study on whether to include $t$ as input, without providing a theoretical analysis. In contrast, our paper offers a detailed explanation of **why $t$ is unnecessary for absorbing diffusion in denoising prediction**. Specifically, as demonstrated in Theorem 1, the model inherently  learns the conditional distribution on clean data, which is fully determined by the dataset. Therefore, **$t$ provides no additional information** in denoising prediction and can be removed from the input. This theoretical insight is further validated by our experiments, as detailed in our response to W3 & Q1 & Q2 (part 2) for reviewer rVfV.
> > >
> > > - **The sampling method of AO-ARM**: In our experiments, we strictly follow the method of sampling the order first and then performing denoising. This ensures that the generated samples precisely follow the distribution defined by the AO-ARM model, enabling a direct and fair comparison with diffusion-based samplers.
> > >
> > > If you have further questions or requirements, we would be happy to continue the discussion. Thank you again for your thoughtful feedback!

---

> > > > ### Comment · Reviewer_4Udp · 2024-11-25
> > > >
> > > > Thanks for your further clarification. I agree that $t$ provides no additional information, but I think there was a misunderstanding of my point. I think your theory is excellent as it establishes a clear connection between ratio parameterization and clean data prediction. However, following the derivation in [1*, 2*]. All you need is the denoised distribution $p(x_0 | x_t)$, which indicates that t does not need to be explicitly introduced in the parameterization. This aligns with the intuition in [2*] (see Section 4.1), where [2*] uses a zero mask as $t$, but $t$ could be omitted entirely from the model. In this regard, while the theorem is unique, practically it does not introduce something very new because one could also parametrise a time-independent model using the formulation in [1*, 2*].
> > > >
> > > > Overall, I find the theorem valuable and insightful, even though it doesn't provide a practical breakthrough. I still recommend acceptance, as the work offers good theoretical contributions and is supported by solid experiments.

---

> ### Author Response · Authors · 2024-11-25
>
> We sincerely appreciate reviewer 4Udp's thorough and timely discussion and apologize for our earlier misunderstanding. We agree that previous works also attempt to remove $t$, and our main contribution lies in providing theoretical and experimental support for why removing $t$ potentially yields better outcomes.
>
> Thank you again for your constructive feedback and acknowledgment of our contributions.

---

### Official Review · Reviewer_rqCn · 2024-11-04

**Soundness:** 3
**Presentation:** 3
**Contribution:** 3
**Rating:** 8
**Confidence:** 3

**Summary:**

The authors express the concrete score for an absorbing (only masked to unmasked) diffusion in terms of a time conditional scalar and prediction of the t=0 data distribution. This significantly simplifies the training of discrete diffusion models as given the state transitions only from masked to unmasked the rate matrix is 0 at all most all positions.

The authors then explain how such a parameterisation can lead to efficient sampling as one may cache the unmasked components rather than recompute.

**Strengths:**

The paper is well explained, the authors make a thorough comparison to concurrent works [1,2] which propose the very much the same methods.

Restricting the generative process from unmasked to masked sounds limiting but greatly simplifies the parameterisation of the concrete score (RADD) as simply learning the masked to unmasked transition.

This RADD reparameterisation also provides insights into heuristic "scaling" in prior works.

The authors provide detailed numerical evidence of how the caching procedure reduces the E-NFE in Fig 1a.

The empirical performance is compelling.

[1]  Shi eta.  Simplified and generalized masked diffusion for discrete data, 2024.
[2] Sahoo et al. Simple and effective masked diffusion language models,  2024

**Weaknesses:**

Although the caching reduces the *effective* NFE, I assume the actual number of function evaluations remains the same given the number of steps, just with fewer outputs per evaluation? This may not reduce workload for jitted - shape static compiled programs like those implemented in jax.

As discussed, many of the ideas are the same as those in concurrent but published works [1,2] from June 2024. Whilst I give the authors the benefit of the doubt given concurrency and explaining similarities, it does temper my score slightly, given it is not clear how the author have attempted to differentiate the paper for this submission.

I will refer to the AC as to whether I should consider these papers, my current score does not reflect these papers.

**Questions:**

See weaknesses.

It is mentioned that the network is time-independent, however, if it is the same as Prop 1 in [1] as discussed in section 5, then it would need to condition on time it seems. Is it really the same as in Prop1 of [1] or can the term in Prop1 of [1] be simplified to be time independent?

[1]  Shi eta.  Simplified and generalized masked diffusion for discrete data, 2024.

---

> ### Author Response · Authors · 2024-11-22
> **Response to Reviewer rqCn**
>
> Thank you for the detailed review and acknowledgment of our contributions.  Below we address specific questions.
>
>
> ## W1:Actual sampling efficiency under cache strategy
> Thank you for your feedback. Your analysis is correct: in static-shape compiled programs, such as those implemented in JAX, our caching strategy does not reduce the workload due to the lack of dynamic batching support. However, PyTorch, which is widely used in both research and industry, supports dynamic shapes. This allows our caching strategy to effectively accelerate sampling, as shown in our results. For futher discussion on mini-batch sampling, please refer to our response to W1.1 for reviewer fLbR.
>
> ## W2 & Q1 : Is RADD's time-independent parameterization unique?
>
> We clarify that our time-independent parameterization is a **unique contribution**. In particular, **the parameterization in [1\*] remains time-dependent, as explicitly shown in their Equation (11)**. In contrast, our approach simplifies the expression to its most fundamental form, completely removing explicit dependency on $t$. For further details, please refer to the common concern 3.
>
> We will make it clearer in the revision.
>
>
> [1*] Shi et al. Simplified and generalized masked diffusion for discrete data, 2024.

---

> > ### Comment · Reviewer_rqCn · 2024-11-27
> >
> > Thank you for the response, I think my score is appropriate.

---

> > > ### Author Response · Authors · 2024-11-27
> > >
> > > Thank you for your feedback. We really appreciate it! In the final revision, we will further polish our paper to incorporate the insights from the rebuttal discussions. Thank you again!

---

### Official Review · Reviewer_fLbR · 2024-11-05

**Soundness:** 3
**Presentation:** 3
**Contribution:** 2
**Rating:** 6
**Confidence:** 5

**Summary:**

This paper studies simplifications of discrete diffusion models with absorbing/masking processes. First, the concrete score in Lou et al. 2024 can be reparameterized as conditional probability of clean data, i.e. p(x_0^i | x_t). Second, when noise samples are not changed during a time-step, cached NN output from previous step can be reused (since the input values are unchanged). Lastly, equivalence between absorbing discrete diffusion and any-order autoregressive models is identified. The simplified training objectives enable better test data perplexity on zero-shot language modeling tasks. The caching trick helps reduce number of function evaluations.

**Strengths:**

The first result on reparameterizing score as conditional denoising probability is the most original contribution. It helps clarify that training concrete score entropy and clean data reconstruction cross-entropy are essentially equivalent up to scalar difference. And this results in better training efficiency compared to SEDD as shown in language modeling experiments. On a side node, the use of clean data reconstruction cross-entropy loss for mask/absorbing discrete diffusion is previously proposed in Austin et al. 2021 for discrete timesteps and Campbell et al. 2024 for continuous time. The connection of this reconstruction distribution and concrete score is newly identified in this paper.

**Weaknesses:**

The second result on caching when samples are not changed during a previous timestep is somewhat well-known.  For example, Chen et al. 2023 proposes to only incur NFE at times when actual transition from mask token to infilled token happens, which also corresponds to how AO-ARM sample one dimension at a step. The caching strategy proposed in this paper becomes less efficient when sampling a batch of samples: in a small timestep when some samples might remain unchanged while some are not, caching cannot be used. Hence I think it should be more efficient to use different $\Delta t$ time steps for different samples in [1], and the $\Delta t$ should be sampled to be the amount of time until next transition from mask to data token for each sample.

The third result on equivalence of absorbing discrete diffusion and AO-ARMs is not new. This has been shown in previous works (Austin et al. 2021, Hoogeboom et al. 2021, Campbell et al. 2024)

[1] Chen, Zixiang, Huizhuo Yuan, Yongqian Li, Yiwen Kou, Junkai Zhang, and Quanquan Gu. "Fast Sampling via De-randomization for Discrete Diffusion Models." *arXiv preprint arXiv:2312.09193* (2023).

[2] Austin, Jacob, Daniel D. Johnson, Jonathan Ho, Daniel Tarlow, and Rianne Van Den Berg. "Structured denoising diffusion models in discrete state-spaces." Advances in Neural Information Processing Systems 34 (2021): 17981-17993.

[3] Hoogeboom, Emiel, Alexey A. Gritsenko, Jasmijn Bastings, Ben Poole, Rianne van den Berg, and Tim Salimans. "Autoregressive diffusion models." *arXiv preprint arXiv:2110.02037* (2021).

[4] Campbell, Andrew, Jason Yim, Regina Barzilay, Tom Rainforth, and Tommi Jaakkola. "Generative flows on discrete state-spaces: Enabling multimodal flows with applications to protein co-design." *arXiv preprint arXiv:2402.04997* (2024).

**Questions:**

- Experiments: For evaluation on language modeling, in G.4 evaluation of generative perplexity, which model is used for evaluating the generation quality? The numbers in Table 3 seem to be much higher than SEDD’s generative perplexity evaluated using GPT-2-Large.
Another ablation might be training both SEDD and RADD to 1000k iterations and compare the performance gap v.s. SEDD and RADD with 400k iterations. Just to see if the gap closes down with more training, or is there a fundamental gap in learning under different parameterization of the model.

- Theory: Clarify which contribution is original and which are already existing in previous literature. Point connection to existing literature when there is. For example, connection to cross-entropy loss proposed in (Austin et al. 2021, Hoogeboom et al. 2021, Campbell et al. 2024); and the connection of caching to derandomized sampling in Chen et al. 2023.

---

> ### Author Response · Authors · 2024-11-22
> **Response to Reviewer fLbR [1/2]**
>
> Thank you for your extremely thorough review and constructive feedback on our paper. Below, we address your concerns and suggestions.
>
> ## W1: Mimi-batch issue and comparison with Chen et al. 2024
>
> ### W1.1: Efficiency of our caching strategy with mini-batch
>
> Our caching strategy enables efficient batch sampling by dynamically adjusting neural network computations to process only the samples that change at each timestep. To illustrate this efficiency, we conducted experiments generating 64 samples under various batch sizes on an NVIDIA 4090 GPU (24GB VRAM, maximum batch size = 8).
>
> The results, summarized in the table below, demonstrate that the average generation time per sample (in seconds) with our caching strategy consistently outperforms SEDD across different batch sizes and timestep configurations:
>
> SEDD-medium:
>
> | Batch Size \Steps | 32   | 64   | 128  | 256  | 512    | 1024   | 2048   | 4096   |
> | ---------------- | ---- | ---- | ---- | ---- | ------ | ------ | ------ | ------ |
> | 1                | 0.98 | 1.83 | 3.57 |  6.95    | 13.80   |27.16| 54.76 | 109.90 |
> | 2                | 0.75 | 1.43 | 2.77 | 5.46 | 10.85 |21.62 | 43.19 | 86.23 |
> | 4                | 0.67 | 1.30 | 2.56 | 5.09 | 10.15 | 20.27 | 40.50 | 80.97 |
> | 8                | 0.70 | 1.34 | 2.65 | 5.25 | 10.46| 20.85 |41.68 | 83.32 |
>
>
> RADD-medium:
>
> | Batch Size \Steps  | 32   | 64   | 128  | 256  | 512  | 1024  | 2048   | 4096   |
> | ---------------- | ---- | ---- | ---- | ---- | ---- | ----- | ------ | ------ |
> | 1                | 0.77 | 1.34 | 2.56 | 4.86 | 8.73 | 14.00 | 20.45 | 30.70 |
> | 2                | 0.60 | 1.08 | 2.05 | 3.98 | 7.32 | 12.32 | 18.88 | 28.97 |
> | 4                | 0.50 | 0.97 | 1.87 | 3.65 | 6.75 | 11.26 | 17.67| 27.67|
> | 8                | 0.51 | 0.97 | 1.90 | 3.71 | 6.76 | 10.87 | 16.76 | 26.58 |
>
> We provide a detailed analysis below:
> - **Batch Size and GPU Utilization**: For a fixed model, increasing the batch size leads to improved GPU utilization before reaching the maximum batch size, reducing the average sampling time per sample. In the case of RADD, sampling time decreases significantly as the batch size increases from 1 to 4. However, the reduction in sampling time becomes minimal beyond a batch size of 4, indicating that GPU utilization has nearly reached its maximum capacity at this point. This demonstrates the scalability of our approach within the limits of the hardware's parallel processing capabilities.
> - **SEDD vs RADD**: Under identical batch sizes and timesteps, RADD consistently outperforms SEDD in sampling speed. This highlights the efficiency of RADD’s design, where the caching mechanism reduces redundant computations and achieves faster generation, especially for larger batch sizes and higher timesteps.
>
> We will add the results in the revision, which improves the quality of the paper.
>
> ## W1.2: Comparison with Chen et al. 2024
>
>
> Our paper and your comment focus on continuous-time setting. As detailed in Common Concern 1, **directly applying the continuous-time samplers from [1\*] to SEDD results in $\text{NFEs} = d$ (equivalent to ARMs). Moreover, Chen et al. [1\*] did not explore methods to reduce NFEs below $d$ for continuous-time models**. In contrast, RADD can directly employ its caching strategy in continuous-time settings, allowing NFEs to be reduced below $d$, as the input to the model remains independent of time.
>
> We will include the discussion in the revised version.
>
>
> ## W2&Q2: Related work
>
> We sincerely thank the reviewer for pointing out the related works. Below, we provide our comparisons and clarifications:
> - **Comparison with Austin et al., 2021**
> Our work provides a rigorous proof of the equivalence, as discussed in detail in the response to Common Concern 2.
> - **Comparison with Hoogeboom et al., 2021**
> We acknowledge that Hoogeboom et al., 2021, establishes the equivalence between ARMs and the ELBO of absorbing diffusion models. However, our approach offers unique contributions, including: an alternative proof of this equivalence leveraging time-independent properties, extending the analysis to demonstrate the equivalence of **four losses** in Eq. (3.7), and conducting a comprehensive study of these losses, presented in Table 1. For more details, please refer to Common Concern 2.
> - **Comparison with Campbell et al., 2024**
> Campbell et al., 2024, is a relevant work on discrete diffusion. However, we did not find a discussion or connection to AO-ARMs in their work.
> - **Comparison with Chen et al., 2023**
> Regarding the connection between caching and derandomized sampling in Chen et al., 2023, please see our detailed response to W1.2.
>
> Further, while our work uses a time-independent cross-entropy loss based on the reparameterization, we do not claim to propose a new loss function.
>
> We are happy to include discussions of these related works in our revised manuscript. These additions will enhance the paper’s quality while leaving our main contributions unchanged.

---

> ### Author Response · Authors · 2024-11-22
> **Response to Reviewer fLbR [2/2]**
>
> ## Q1: Experiments
>
> ### Q1.1: Which model is used for evaluating the generation quality?
>
> For the evaluation of generative perplexity in Section G.4, we also used the GPT-2-Large model. The higher generative perplexity values compared to SEDD [2*] are primarily due to differences in precision during categorical sampling. We explain the precision choices below:
> - Previous works, including SEDD, employed **fp32 precision** for categorical sampling. This introduced precision errors that effectively acted as a form of annealing, resulting in deceptively lower perplexity values (see Appendix G.3).
> - In contrast, we utilized **fp64 precision** to minimize precision errors. While this approach provides more accurate sampling, it inherently results in higher perplexity values due to the lack of annealing effects.
>
> To address your concern, we add new results using **fp32 precision**, where **SEDD’s generative perplexity matches the values reported in their original paper**.  The results are as follows:
>
> RADD-$\lambda$-DCE (medium):
>
> | Metric/Steps | 32  | 64  | 128 | 256 | 512 | 1024 | 2048 | 4096 |
> |--------------|------|------|-----|-----|-----|------|------|------|
> | Avg. NFEs     | 32.00   | 64.00   | 127.96 | 251.35 | 442.84 | 647.48 | 805.98 | 906.13 |
> | PPL          | 126  | 84   | 63  | 51  | 41  | 33   | 27   | 22   |
> | Entropy      | 8.19 | 8.09 | 7.98 | 7.88 | 7.75 | 7.59 | 7.45 | 7.27 |
>
> SEDD (medium):
>
> | Metric/Steps | 32  | 64  | 128 | 256 | 512 | 1024 | 2048 | 4096 |
> |--------------|------|------|-----|-----|-----|------|------|------|
> | Avg. NFEs     | 32   | 64   | 128 | 256 | 512 | 1024 | 2048 | 4096 |
> | PPL          | 125  | 85   | 62  | 51  | 41  | 34   | 27   | 22   |
> | Entropy      | 8.18 | 8.07 | 7.96 | 7.86 | 7.73 | 7.60 | 7.44 | 7.25 |
>
> These results demonstrate that under **fp32 precision**, the perplexity of RADD-$\lambda$-DCE (medium) and SEDD (medium) are similar when using the same number of sampling steps. However, RADD's caching strategy enables it to achieve comparable perplexity with fewer NFEs (Number of Function Evaluations), highlighting its advantage in sampling efficiency. Namley, **the conclusion in fp32 precision holds the same as in fp64 precision.**
>
> ### Q1.2: Training Ablation Experiments
>
> Thank you for the suggestion. We agree that an ablation study would provide valuable insights. Unfortunately, due to time constraints, we are unable to complete this experiment within the rebuttal period. However, we are currently running the experiment and will include the results in the final version of the manuscript.
>
> Nevertheless, we believe that the 400K results, conducted following the settings of SEDD, are sufficient to demonstrate the effectiveness of our approach.
>
> [1*] Chen, Zixiang, Huizhuo Yuan, Yongqian Li, Yiwen Kou, Junkai Zhang, and Quanquan Gu. Fast Sampling via Discrete Non-Markov Diffusion Models, 2024.
>
> [2*] Aaron Lou, Chenlin Meng, and Stefano Ermon. Discrete diffusion modeling by estimating the ratios of the data distribution, 2024.
>
>
> [3*] Austin, Jacob, Daniel D. Johnson, Jonathan Ho, Daniel Tarlow, and Rianne Van Den Berg. "Structured denoising diffusion models in discrete state-spaces." Advances in Neural Information Processing Systems 34 (2021): 17981-17993.
>
> [4*] Hoogeboom, Emiel, Alexey A. Gritsenko, Jasmijn Bastings, Ben Poole, Rianne van den Berg, and Tim Salimans. "Autoregressive diffusion models." arXiv preprint arXiv:2110.02037 (2021).
>
> [5*] Campbell, Andrew, Jason Yim, Regina Barzilay, Tom Rainforth, and Tommi Jaakkola. "Generative flows on discrete state-spaces: Enabling multimodal flows with applications to protein co-design." arXiv preprint arXiv:2402.04997 (2024).

---

> ### Author Response · Authors · 2024-11-25
>
> Dear Reviewer fLbR,
>
> Thank you once again for your constructive feedback. We would like to kindly remind you that we have now included additional experiments  to address your concerns:
> - Validation of the acceleration performance of our cache strategy under varying batch sizes.
> - Generative perplexity results under FP32 precision, aligned with SEDD.
>
> Additionally, we have also added a detailed comparison with all related works including Chen et al. 2024, Austin et al., 2021, Hoogeboom et al., 2021, and Campbell et al., 2024.
>
> ---
>
> As the discussion period is coming to a close in two days, we look forward to your response and would be happy to address any further comments or questions you may have.
>
> Best,
>
> The Authors

---

> ### Author Response · Authors · 2024-11-27
>
> Dear Reviewer fLbR,
>
> Thank you once again for your valuable comments. We believe that our responses have addressed your concerns comprehensively. Could you kindly share whether our replies meet your expectations and if you would consider reevaluating the paper?
>
> Best regards,
>
> The Authors

---

> ### Comment · Reviewer_fLbR · 2024-11-27
>
> Thank you very much for your response.
>
> The clarification of generative perplexity evaluation gap makes sense to me, given the differences between the fp32 and fp64 implementations. My primary concern was in the presentation of the paper's claims. I agree with Reviewer 4Udp that the main novelty of the work lies in establishing the theoretical connection between the parameterization of score entropy (ratio) and clean data prediction. But the practical contribution is less pronounced, as the performance are not very different under one form or the other, especially in terms of generation quality. Additionally, the theory surrounding time-independent clean data reconstruction is relatively well-known and not novel. Therefore, I would recommend placing greater emphasis on the theoretical connection in the reparameterization framework and less on Section 3.3.
>
> Regarding caching, my concern is more about the number of NFEs required as the batch size increases. In such cases, it is highly likely that at least one sample in the batch will require a transition from a mask to a token. This introduces a limit to caching efficiency as batch size grows, as observed in concurrent work [Sahoo et al., 2024].
>
> I have updated my evaluation score and look forward to seeing the revised version address the points I raised, particularly those related to the theoretical presentation.
> [1] Sahoo, Subham Sekhar, Marianne Arriola, Yair Schiff, Aaron Gokaslan, Edgar Marroquin, Justin T. Chiu, Alexander Rush, and Volodymyr Kuleshov. "Simple and Effective Masked Diffusion Language Models." arXiv preprint arXiv:2406.07524 (2024).

---

> ### Author Response · Authors · 2024-11-28
> **relationship with the concurrent work [Sahoo et al., 2024]**
>
> We sincerely appreciate your constructive feedback and the score update. We would like to clarify that the concurrent work [Sahoo et al., 2024] does not **claim omitting $t$ is beneficial for training, nor does it explain why this is the case**.
>
> To provide some context, we first reference the concurrent work [Sahoo et al., 2024] as described in their Sec 3.2.2:
> >"Therefore, we introduce a model $x_\theta(z_t, t): V \times [0, 1] \to \Delta^K$ that approximates $x$ with a neural network. We can also omit explicit dependence of $x_\theta$ on time $t$, which **simplifies sampling, yielding a 2x inference speed-up**”
>
> It is evident that [Sahoo et al., 2024] **approximates** the time-dependent function $x_\theta(z_t, t)$ with a time-independent $\text{NN}(z_t)$ **solely for sampling, without providing any explanation as to why this omission does not impact training**.
>
> Moreover, in Appendix E.5 of [Sahoo et al., 2024], Table 12 empirically demonstrates that removing $t$ results in slightly better PPL on a single task. However, they **do not claim and explain the improvement at all, stating only the following without further analysis**:
> > "We observe that time-conditioning has minimal impact on perplexity."
>
> In conclusion, we believe it is fair to assert that the concurrent work [Sahoo et al., 2024]  does not **claim that omitting $t$ is beneficial for training, nor do they provide any theoretical justification for it**.
>
> In contrast, our work provides a rigorous theoretical foundation demonstrating that **transition probability can be parameterized in a manner that eliminates dependence on $t$** in both the score parameterization (Theorem 1) and the mean parameterization (Eqn. E1 in Appendix E of our submission) as used in [Sahoo et al., 2024].  This theoretical insight **clarifies why our reparameterization on omitting $t$ simplifies the training target and, consequently, benefits the training process**.
>
> Such a theory provides a significant practical improvement. In particular, the *SEDD-Scale* and *RADD-DSE* models in our submission differ only in whether $t$ is removed or retained, which naturally serves as an ablation study on $t$ and a validation of our theoretical insights (see more details in response to Q2 from Reviewer rVfV). The results in Tables 1 and 2 of our submission demonstrate that RADD-DSE consistently and substantially outperforms SEDD-Scale, **providing strong empirical support for our contributions, as also acknowledged by Reviewer 4Udp**.
>
> Once again, we deeply appreciate your valuable feedback. We will revise Section 3.1 as soon as possible as the rebuttal PDF submission deadline is closing soon.

---

> > ### Comment · Reviewer_fLbR · 2024-11-28
> >
> > There seems to be some misunderstanding regarding my previous point about including or excluding $t$ during training. I was not referring to the existing results in [Sahoo et al., 2024]. Since mask tokens already provide sufficient information about the current $x_t$, $t$ itself may be unnecessary. It is nice that your paper contains theory for this. In practice, whether including $t$ benefits training depends on the neural network (NN) architecture, model capacity, and training dynamics. If $t$ provides more benefits than drawbacks—such as offering better context when model capacity is limited—it can always be included in the NN or inferred from the number of mask tokens. However, if the model is not capacity-limited, omitting $t$ might encourage better generalization behavior.
> >
> > The reference to [Sahoo et al., 2024] is about caching: the number of function evaluations (NFE) increases with batch size, as even within a very fine step, a single token change in one sample may still occur and require an additional NFE. I just hope that this can be discussed in a similar way so that the effect of caching can be fully characterized for different sampling settings.

---

> ### Author Response · Authors · 2024-11-28
> **About caching**
>
> Thank you for your timely feedback and acknowledgment of our theory. Regarding the caching mechanism, we clarify that the NFE mentioned in our paper refers to a single sample, but it applies to minibatch cases, as detailed in our response to your W1.1.
>
> In particular, we agree that within a minibatch, some samples may remain unchanged while others may evolve. In our implementation, **we account for this by employing a dynamic batch-size approach**. Only the samples that have changed are passed through the neural network for computation. While this still involves a "batch-level NFE" (your NEF according to our understanding), **the overall computational load is reduced, leading to faster sampling**. In the response to your W1.1, we included a comparison of sampling speeds for batch sizes of 1, 2, 4, and 8 (the maximum batch size per GPU). The results show that caching consistently accelerates sampling.
>
> In comparison, concurrent work  [Sahoo et al., 2024] does not consider dynamic batch size in their implementation, so their practical acceleration falls short of the theoretical potential.
>
> We sincerely appreciate your feedback, which helps make our paper clearer. We have already presented the accelerated sampling results for minibatch scenarios in Appendix J.4 of our revision. We will further incorporate the above discussion to provide additional clarity. Please feel free to reach out with any further questions.

---

### Official Review · Reviewer_rVfV · 2024-11-06

**Soundness:** 3
**Presentation:** 3
**Contribution:** 2
**Rating:** 6
**Confidence:** 4

**Summary:**

Previous work in discrete diffusion language modeling have relied on estimating a time-dependent concrete score. The authors derive a simpler parameterization, RADD, expressed as a time-independent conditional probabilities of the clean data, multiplied by a time-dependent scalar. By deriving a simplified objective, they provide a deeper understanding of discrete diffusion and draw a connection with any-order autoregressive models. By training a time-independent denoising network, they simplify the model architecture by removing additional time-conditioning layers used in prior discrete diffusion language models. The authors show theoretically and experimentally that using a time-independent network can significantly reduce the number of function evaluations at inference. On language modeling benchmarks, their architectural simplifications and optimized training loss results in state-of-the-art performance among discrete diffusion models at the GPT-2 scale.

**Strengths:**

- They provide clear, rigorous analysis showing the equivalence between the concrete score [1], their simplified parameterization, and the any-order autoregressive loss
- The connection between discrete diffusion models and any-order autoregressive models is novel and unexplored in prior work. The authors provide a theoretical and empirical comparison
- They show substantial improvement in sampling speed over SEDD [1] through convincing theoretical and experimental analysis
- They provide thorough comparison to concurrent work [2,3] and explain the added novelty introduced by their paper

[1] Lou, Aaron, et al. "Discrete diffusion language modeling by estimating the ratios of the data distribution."

[2] Shi, Jiaxin. "Simplified and Generalized Masked Diffusion for Discrete Data"

[3] Sahoo, Subham Sekhar. "Simple and Effective Masked Diffusion Language Models."

**Weaknesses:**

- The experimental results are limited. Their likelihood evaluation only includes zero-shot likelihoods using models trained on OpenWebText
- The empirical comparison across RADD models is misleading as they have equivalent objectives (Tables 1, 2, 5). In Table 2, one RADD model underperforms SEDD on LAMBADA while another outperforms.
- Experimental ablations on the architectural and theoretical simplifications are not provided, making it difficult to understand its impact on likelihoods (an ablation on the scaling term is instead provided for SEDD). For example, they claim to design a time-independent network that simplifies learning: it would be valuable to compare RADD with and without time conditioning, while controlling for parameter count
- The sampling speed is not compared with AR, making it unclear whether the speedup improvement has practical significance

**Questions:**

- Does the improvement from including the scaling term in SEDD also hold for RADD?
- If a claim is that the simplification enables a simpler architecture, can the authors provide comparative analysis between RADD with and without time conditioning while controlling for parameter count?
- If the objectives presented are equivalent, why are there differences in the zero-shot likelihoods in Tables 1, 2, 5?
- In Table 4, can the authors compare perplexity from any-order autoregressive generation and parallel generation using the $\tau$-leaping sampler?
- What is the number of tokens used for training, instead of the number of iterations, in order to compare with baselines? In Table 5, the authors report zero-shot likelihood on models trained for 1000K iterations, why do the models in the main paper use 400K iterations?

---

> ### Author Response · Authors · 2024-11-22
> **Response to Reviewer rVfV [1/2]**
>
> Thank you for your detailed review and insightful comments. Below, we address your concerns and suggestions.
>
> ## W1: Limitations of experiments
>
> We utilized OpenWebText and the zero-shot likelihood metric due to their representativeness and widespread use in benchmarking language models [1*, 2*, 3*], ensuring comparability with prior work. Additionally, this choice aligns with SEDD, our primary competitor, to maintain fairness. We are open to conducting additional experiments based on specific recommendations and will include the results in the final version given the time limits during rebuttal.
>
> ## W2 & Q3: Differences in zero-shot likelihoods in Tables 1, 2, and 5
>
> Thank you for your constructive feedback. The equivalence of the objectives we refer to is in terms of their expected values. Specifically, in an ideal scenario with infinite data, the two loss functions would converge to the same solution. However, for models trained on finite datasets using gradient-based optimization methods, the gradient estimation for each loss function differs. Combined with the **non-convex nature of neural networks**, these factors lead the models to converge to different local optima, which explains the discrepancies in zero-shot likelihoods observed in Tables 1, 2, and 5.
>
> We also appreciate your observation that one RADD model underperforms SEDD on LAMBADA while another outperforms it. This inconsistency likely stems from the **inherent stochasticity** and **non-convexity** of the optimization process. Nonetheless, across the five datasets, the overall performance of all RADD loss functions surpasses that of SEDD.
>
> We will clarify it in the final version.
>
> ##  W3 & Q1 & Q2 : Architectures and ablation
>
> ### Part 1: Architecutural comparison between RADD and SEDD
>
> We would like to clarify a potential misunderstanding regarding the architectural differences between RADD and SEDD. The theoretical foundation of RADD demonstrates that modeling the concrete score function **requires both the removal of time conditioning and the inclusion of a scaling term** to achieve optimal performance. The architectural differences between these models are summarized below:
>
> | Model         | Time Conditioning Removed | Scaling Term Applied |
> | :------------ | :-----------------------: | :------------------: |
> | SEDD unscaled |             ✘             |          ✘           |
> | SEDD scaled   |             ✘             |          ✓           |
> | RADD          |             ✓             |          ✓           |
>
> ### Part 2: Ablation of the time condition
>
> It can be seen that adding the time condition back into RADD would essentially revert it to the same model as SEDD-Scale. In our submission, we trained a version of RADD with the same DSE loss as SEDD, referred to as **"RADD-DSE"**. These models have similar parameter counts, with the main difference being the inclusion or exclusion of time conditioning. Therefore, **the results of SEDD-Scale and RADD-DSE in Tables 1 and 2 of our submission serve as an ablation study on the impact of time conditioning**. Notably, RADD-DSE significantly outperforms SEDD-Scale in these comparisons, underscoring the effectiveness of RADD’s architectural design.
>
> We will make it clearer in the revision.
>
> ## W4: AR sampling speed
>
> Thanks for the constructive comments. As highlighted in Fig. 1 of SEDD [3*], SEDD surpasses AR sampling in terms of speed. Moreover, RADD further improves upon SEDD in sampling speed, as demonstrated in Fig. 1b and Table 3 of our submission, indicating that RADD also outperforms AR sampling.
>
> We will include this discussion in the revised version. If the reviewer is interested, we are open to including a direct comparison with ARMs in the final version, as the time constraints of the rebuttal phase limit our ability to provide this analysis now.

---

> ### Author Response · Authors · 2024-11-22
> **Response to Reviewer rVfV [2/2]**
>
> ## Q4: Additional experiments on sampling
>
> Thank you for raising this point. Regarding the $\tau$-leaping sampler mentioned in the question, we are unsure whether you are referring to $\tau$-leaping sampler for discrete diffusion from [4*] or the Tweedie $\tau$-leaping sampler proposed in [3*]. To clarify:
> - **$\tau$-leaping sampler** [4*] is an earlier sampling method.
> - **Tweedie $\tau$-leaping sampler** [3*], which is a more recent and efficient approach, has been adopted in our work.
>
> If the question refers to the Tweedie $\tau$-leaping sampler, we present the following perplexity results for RADD-$\lambda$-DCE medium:
>
> | Steps           | 32  | 64  | 128 | 256 | 512 | 1024 | 2048 | 4096 | 1024 (AO-ARM) |
> |------------------|------|------|------|------|------|-------|-------|-------|-------------------|
> | Perplexity (PPL) | 158  | 113  | 96   | 89   | 84   | 83    | 81    | 81    | 83             |
>
> Except for the last column (corresponding to the AO-ARM results under the `random` row in Table 4), all values were measured using Tweedie $\tau$-leaping. The results demonstrate that with increasing steps, the performance of Tweedie $\tau$-leaping converges and aligns closely with the AO-ARM results. All evaluations were conducted on the same RADD-$\lambda$-medium model to ensure a fair comparison.
>
>
> We will add the new results and dicussion in the revision. If you are referring to the $\tau$-leaping sampler from [4*], please let us know, and we will be happy to perform the necessary experiments.
>
>
>
> ## Q5:Training Tokens and Iterations:
>
>
> Thank you for your question. The training tokens for SEDD and RADD **are the same when using the same number of iterations**. Specifically, for both models:
> - 400K iterations correspond to approximately 105 billion tokens.
> - 1000K iterations correspond to approximately 262 billion tokens.
> It is worth noting that the entire OpenWebText dataset contains only 9 billion tokens, meaning both models cycled through the dataset multiple times during training.
>
> In the main paper, we report results for models trained for 400K iterations to align with the primary baseline, SEDD [3*], which was also trained for 400K iterations. This alignment ensures a fair comparison. Training for 1000K iterations was conducted to observe convergence behavior and evaluate the performance trends of our RADD model more comprehensively.
>
> We will clarify it in the revision.
>
>
> [1*] Aaron Gokaslan and Vanya Cohen. Openwebtext corpus. http://Skylion007.github.io/ OpenWebTextCorpus, 2019
>
> [2*] Gulrajani, I. and Hashimoto, T. Likelihood-based diffusion language models. In Advances in Neural Information Processing Systems, 2023
>
> [3*] Aaron Lou, Chenlin Meng, and Stefano Ermon. Discrete diffusion modeling by estimating the ratios of the data distribution, 2024.
>
> [4*] Andrew Campbell, Joe Benton, Valentin De Bortoli, Tom Rainforth, George Deligiannidis, and Arnaud Doucet. A Continuous Time Framework for Discrete Denoising Models, 2022.

---

> > ### Comment · Reviewer_rVfV · 2024-11-22
> > **Thank you for clarifying**
> >
> > I appreciate the authors' effort to clarify my concerns. I have a better understanding of the scaling term ablation, and appreciate the additional analysis of the $\tau$-leaping sampler compared to AO-ARM. It is promising that the RADD sampler achieves comparable PPL to AO-ARMs.
> >
> > However I still have the following remaining concerns:
> > 1. In terms of limitation of experiments, perhaps the authors can report test perplexity on OWT, or Mauve scores of conditionally generated text to compare with SEDD's Table 5? If the authors think this isn't feasible, I'm open to being wrong on this
> > 2. I understand now that the different objectives may be equivalent in expectation. Am I correct in understanding that they differ in terms of their *variance*? I don't quite follow the provided explanation that the gradient estimation is different. If the authors can provide some additional comment on how the objectives differ in terms of their *variance*, this would be greatly appreciated.

---

> ### Author Response · Authors · 2024-11-22
> **Further clarification**
>
> Thank you for your timely response. We appreciate your recognition of our clarifications and are glad to address the remaining points:
>
> 1. **Test Perplexity on OWT**: OpenWebText [1*] does not include a predefined test set. In line with the methodology followed in SEDD, we did not partition the OpenWebText into a separate test set for evaluation. Consequently, neither SEDD nor RADD could report test perplexity on OWT.
>
> 2. **Mauve Scores of Conditionally Generated Text**: We acknowledge the importance of evaluating conditional generation quality using metrics such as Mauve. While we have been actively working to obtain Mauve scores for comparison with Table 5 of the SEDD paper, based on the official code of SEDD, we have encountered challenges replicating SEDD's reported Mauve results. Notably, other works [5*] have also reported similar difficulties as discussed in Appendix A.4 . We are continuing to resolve these issues and hope to report the results soon.
>
> 3. **Variance of the Objectives**: Yes, different objectives have distinct variances. Namely, the gradients of the losses are different when estimated on finite (i.e. a mini-batch of) samples, influencing training dynamics and convergence speed. This could further lead to discrepancies in zero-shot performance even when training for the same number of steps. To the best of our knowledge, quantifying the variances of such objectives is challenging due to the presence of the neural network. We will add a discussion in the revision.
>
>
> If you have further questions or requirements, we would be happy to continue the discussion. Thank you again for your thoughtful feedback!
>
> [1*] Aaron Gokaslan and Vanya Cohen. Openwebtext corpus. http://Skylion007.github.io/ OpenWebTextCorpus, 2019
>
> [5*] J. Deschenaux and C. Gulcehre, “Promises, outlooks and challenges of diffusion language
> modeling,” 2024, https://arxiv.org/abs/2406.11473

---

> > ### Author Response · Authors · 2024-11-24
> > **Further discussion about the Mauve scores**
> >
> > We appreciate your thoughtful feedback on **Mauve Scores of Conditionally Generated Text**.
> >
> > In line with your suggestion, we investigated MAUVE scores further and observed that they are highly sensitive to hyperparameter variations. For example, even altering the random seed alone can lead to substantial differences in the scores for the same generated text. This sensitivity prevented us from reproducing the results reported in Table 5 of SEDD [1*]. A similar issue has been explored in greater detail, along with additional experiments, in the aforementioned reference [5*].
> >
> > **Please do not hesitate to share any further concerns or questions. We would be glad to address them.**

---

> > > ### Comment · Reviewer_rVfV · 2024-11-24
> > > **Re: clarification/discussion**
> > >
> > > The authors have addressed most of my concerns, I have adjusted my score

---

> > > > ### Author Response · Authors · 2024-11-25
> > > >
> > > > We sincerely thank reviewer rVfV for the constructive comments, timely responses, and score adjustment. We believe the quality and clarity of the paper have been improved.

---

### Official Review · Reviewer_u3V7 · 2024-11-10

**Soundness:** 3
**Presentation:** 2
**Contribution:** 2
**Rating:** 6
**Confidence:** 3

**Summary:**

This paper proposes a reparameterization of the absorbing discrete diffusion, based upon the finding that the concrete score can be decomposed into a time-independent conditional probability of clean data, and a time dependent but data independent term. Therefore, this paper proposes a new network architecture, that removes the time dependency, and add softmax layers to project the output layer to the probability space.

The authors argue that the removed time dependency can accelerate the sampling process by caching the network output before it changes. The total changes would be at most $d$ instead of $n$, and the total number of NFEs reduces to  $n(1 - (1 - \frac{1}{n}))^l)$.

**Strengths:**

The authors proposed a novel reparameterization of absorbing discrete diffusion, decoupled the time-independent and time-dependent part. This discovery allows for better understanding of discrete diffusions, and the previous reparameterization in SEDD.  The proposed reparameterization allows for a simplified network architecture and efficient sampling.

Moreover, the authors proposed the unification of absorbing diffusion with the any-order autoregressive model.

**Weaknesses:**

First of all, the proposed algorithm are specified to absorbing discrete diffusion, and does not fit in the other widely used discrete diffusion model, multinomial diffusion. The authors did not discuss if or why not their methodology can not be applied to multinomial diffusion.

There are existing works that have reduced the sampling complexity of discrete diffusions, including absorbing and multinomial diffusion, e.g., Chen et al. 2024 [1]. [1] also made use of the fact that during the reverse sampling process, computation is only required when tokens changes, thus reducing the computation from number of diffusion steps to number of tokens. Also [1] proposed a similar NFE estimation in Theorem D.1, which is the same as (3.4) in this paper. However, the authors did not discuss or mention the relationship with [1].

Also inspired from [1], the sampling complexity essentially does not comes from whether you have $t$ in your score network or not, but comes from times that you change tokens. Even for SEDD based discrete diffusion network, it is also possible to only calculate when change is required, and skip calculation when $x$ does no change. For example, if using [1] for sampling, does RADD still have any advantage over SEDD?

[1] Fast Sampling via Discrete Non-Markov Diffusion Models.

**Questions:**

See weaknesses.

---

> ### Author Response · Authors · 2024-11-22
> **Response to Reviewer u3V7**
>
> Thank you for your extremely thorough review and constructive feedback on our paper. Below, we address your concerns and suggestions.
>
>
> ## W1: Can the RADD methodology be applied to other discrete diffusion models?
> We appreciate your thoughtful question. RADD is specifically designed for absorbing discrete diffusion, which has demonstrated SOTA performance among discrete diffusion models [1*,2*].  The key intuition behind RADD lies in factoring concrete score into a time-independent term and a time-dependent term. To the best of my knowledge, this specific decomposition is not directly applicable to other models like multinomial diffusion, which have fundamentally different score formulations. We will clarify this in the manuscript.
>
>
> ## W2: Comparison with Chen et al. 2024
>
>
> Thank you for your insightful question. Below, we summarize the key differences, with additional details provided in Common Concern 1. The samplers proposed in Chen et al. (2024) [3*] and in this work are **applicable in different scenarios**, as discussed in the response to W2.2.
>
>
> ## W2.1: Similar NFEs estimation with Chen et al. 2024
>
> Although the samplers in this paper and in Chen et al. (2024) originate from different formulations (detailed in Common Concern 1), the results of Theorem D.1 [3*] for the discrete-time sampler align with those of our Eq. (3.4). However, as discussed later, the two samplers are applicable in different scenarios. If the reviewer has additional insights, we would welcome further discussion.
>
> ## W2.2: If using methods in Chen et al. 2024 for sampling, does RADD still have any advantage over SEDD?
>
> SEDD is a continuous-time model. As outlined in Common Concern 1, **directly applying the continuous-time samplers from [3\*] to SEDD results in $\text{NFEs} = d$ (equivalent to ARMs). Moreover, Chen et al. [3\*] did not explore methods to reduce NFEs below $d$ for continuous-time models like SEDD**. In contrast, RADD can directly employ its caching strategy in continuous-time settings, allowing NFEs to be reduced below $d$, as the input to the model remains independent of time.
>
> Additionally, **RADD not only simplifies but also improves the performance of SEDD (see Table 1 of our submission), providing advantages that extend beyond fast sampling**.
>
> We will include all related discussions in the revised manuscript.
>
>
> [1*] Austin, Jacob, Daniel D. Johnson, Jonathan Ho, Daniel Tarlow, and Rianne Van Den Berg. "Structured denoising diffusion models in discrete state-spaces." Advances in Neural Information Processing Systems 34 (2021): 17981-17993.
>
> [2*] Aaron Lou, Chenlin Meng, and Stefano Ermon. Discrete diffusion modeling by estimating the ratios of the data distribution, 2024.
>
> [3*] Chen, Zixiang, Huizhuo Yuan, Yongqian Li, Yiwen Kou, Junkai Zhang, and Quanquan Gu. "Fast Sampling via Discrete Non-Markov Diffusion Models.", 2024

---

> ### Author Response · Authors · 2024-11-25
>
> Dear Reviewer u3V7,
>
> Thank you once again for your constructive feedback. We would like to kindly remind you that we have clarified the applicable scenarios of RADD and added a detailed comparison with Chen et al. 2024.
>
>
> As the discussion period is coming to a close in two days, we look forward to your response and would be happy to address any further comments or questions you may have.
>
> Best,
>
> The Authors

---

> > ### Author Response · Authors · 2024-11-27
> > **Looking forward to your reply**
> >
> > Dear Reviewer u3V7,
> >
> > Thank you once again for your valuable comments. We believe that our responses have addressed your concerns comprehensively. Could you kindly share whether our replies meet your expectations and if you would consider reevaluating the paper?
> >
> > Best regards,
> >
> > The Authors

---

> > > ### Author Response · Authors · 2024-11-29
> > > **Looking forward to your reply**
> > >
> > > Dear Reviewer u3V7,
> > >
> > > Thank you once again for taking the time to provide your valuable comments. We fully understand that you may currently be very busy. However, considering that the rebuttal period is nearing its end, we feel compelled to kindly remind you for the third time to review our rebuttal and reconsider the evaluation of this manuscript. Thank you very much for your understanding and time.
> > >
> > > Best regards,
> > >
> > > The Authors

---

> ### Author Response · Authors · 2024-12-02
>
> Dear Reviewer u3V7,
>
> Thank you very much for your time and valuable comments. We regret that we have not received your response during the rebuttal period. As the rebuttal phase is set to conclude in one day, we believe that our responses have sufficiently addressed your concerns.
>
> Best regards,
>
> The Authors

---

### Author Response · Authors · 2024-11-22
**Common Concern [1/2]**

We thank all the reviewers for their time and valualbe feedback. We first address the common concerns below.

## Common Concern 1 (from Reviewer u3V7 and Reviewer fLbR) Comparison with Chen et al. 2024 in terms of fast sampling

We thank the reviewers for pointing out the related work [1*]. We begin by summarizing their contributions and then provide a detailed comparison with our work.

Chen et al. 2024[1*] proposes sampling methods for discrete-time and continuous-time diffusion models individually. Consider a sequence $\boldsymbol{x}$ of length $d$.
1. **Discrete-Time Models**: For models trained over discrete timesteps $\mathcal{T}_ {\text{train}} = \{1, \cdots, T\}$,  [1*] proposes to pre-sample the $d$ time points $\tau_ i \in \mathcal{T}_ {\text{train}}$, where each $\tau_ i$ corresponds to a change timepoint for a specific dimension of $\boldsymbol{x}$. These timepoints form a timeset $\mathcal{T}_ {\text{change}} \subset \mathcal{T}_ {\text{train}}$, and updates are only applied at these steps. For absorbing diffusion models, as each token changes only once, $\text{NFEs} = |\mathcal{T}_ {\text{change}}| \leq \min(d, T)$.

2. **Continuous-Time Models**: For models trained over continuous time, $d$ change points $\tau_ i$ are pre-sampled and sorted in ascending order ($\tau_ {n_ 1} < \cdots < \tau_ {n_ d}$). Updates are sequentially applied at these points, resulting in **$\text{NFEs} = d$**, which resembles the sampling process in AO-ARM. **However, how to reduce the NEFs less than $d$ for the continuous-time model has not been investigated in Chen et al. 2024 [1\*].**

In contrast to Chen et al. (2024), which pre-sample specific time points and update tokens only at those predetermined points, RADD leverages a **time-independent parameterization** that updates tokens only when they change. This fundamental difference results in **different applicable  scenarios** for the two methods:
  - Chen et al. 2024 applies to both absorbing and multinomial diffusion models. However, for continuous-time models like SEDD, their sampling method results in **$\text{NFEs} = d$** and, as noted,  how to reduce the NEFs less than $d$ for the continuous-time model has not been investigated.
  - RADD, on the other hand, is specifically designed for absorbing diffusion models. **It is straightfoward to apply the cache strategy of the RADD to continuous-time settings and reduces NEFs less than $d$** because the input of the model is independent of the time.

Besides the above contribution for fast sampling, we emphasize that, as acknowledged by most of the reviewers, the core contribution of this paper lies in the reparameterization theorem, which not only simplifies but also improves the performance of existing MDMs, providing advantages that extend beyond fast sampling.

Nevertheless, we are glad to discuss the related work in our revision.

---

### Author Response · Authors · 2024-11-22
**Common Concern [2/2]**

## Common Concern 2 (from Reviewer fLbR and Reviewer 4Udp): Comparison to prior works concerning equivalence discussion

We sincerely thank the reviewers for pointing out the related work [2*] and [3*]. We provide a detailed comparison below.

### Comparisons with Austin et al. [2*]
- Austin et al. [2*] made an early attempt to explore the connection between absorbing discrete diffusion and AO-ARM. However, their work lacks rigorous proof. Instead, they qualitatively discuss the correlation between the two loss functions. In Appendix A.3 of [2*], they describe the relationship by stating, ***"this looks very similar ... it is not exactly identical."***
- In contrast, our work rigorously establishes this connection. By leveraging our continuous-time framework and the time-independent parameterization presented in Theorem 1, we provide a formal proof demonstrating the equivalence between absorbing discrete diffusion and AO-ARM.

### Comparison with [3*]

We acknowledge that [3*] establishes the equivalence between ARMs and the ELBO of the absorbing diffusion models, corresponding to the $\mathcal{L}_ {\text{DSE}}^T(\boldsymbol{x}_ 0)$ and $\mathcal{L}_ {\text{AO}}(\boldsymbol{x}_ 0)$ terms in Eq. (3.7) of our submission. However, our approach offers unique contributions by:

- Leveraging the time-independent properties under the reparameterization formulation, which provides an alternative proof of this equivalence.
- Extending the analysis to demonstrate the equivalence of **four losses** including $\mathcal{L}_ {\text{DSE}}^T(\boldsymbol{x}_ 0)$ and $\mathcal{L}_ {\text{AO}}(\boldsymbol{x}_ 0)$ in Eq. (3.7), deepening our understanding of absorbing discrete diffusion.
- Conducting a comprehensive study of these losses, presented in Table 1, which has not been previously explored before to our knowledge.

### Conclusion

We are happy to incorporate a discussion of the related works [2*] and [3*] in our revised manuscript and make revisions accordingly. We believe our work complements and extends earlier research by providing a **rigorous and comprehensive perspective** on equivalence discussions in diffusion models.




## Common Concern 3(from Reviewer rqCn and Reviewer 4Udp): Is RADD's time-independent parameterization unique?


We would like to emphasize that our decomposition of the concrete score into time-independent and time-dependent components, followed by the introduction of a time-independent parameterization, represents a **unique contribution** of our work.

To clarify, while there are formal similarities between Proposition 1 in [5*] and our Theorem 1, it is crucial to note that **the parameterization in [5\*] remains time-dependent, as explicitly shown in their Equation (11)**. In contrast, our approach simplifies the expression to its most fundamental form, completely removing explicit dependency on $t$.

This innovation enables our framework to achieve a truly **time-independent** parameterization, which serves as the foundation for subsequent contributions in Sec. 3.2 and Sec. 3.3 of our submission.


[1*] Chen, Zixiang, Huizhuo Yuan, Yongqian Li, Yiwen Kou, Junkai Zhang, and Quanquan Gu. "Fast Sampling via Discrete Non-Markov Diffusion Models.", 2024

[2*] Austin, Jacob, Daniel D. Johnson, Jonathan Ho, Daniel Tarlow, and Rianne Van Den Berg. "Structured denoising diffusion models in discrete state-spaces." Advances in Neural Information Processing Systems 34 (2021): 17981-17993.

[3*] Hoogeboom, Emiel, Alexey A. Gritsenko, Jasmijn Bastings, Ben Poole, Rianne van den Berg, and Tim Salimans. "Autoregressive diffusion models." arXiv preprint arXiv:2110.02037 (2021).

[4*] Andrew Campbell, Joe Benton, Valentin De Bortoli, Tom Rainforth, George Deligiannidis, and A. Doucet. A continuous time framework for discrete denoising models, 2022.


[5*] Jiaxin Shi, Kehang Han, Zhe Wang, Arnaud Doucet, and Michalis K. Titsias. Simplified and generalized masked diffusion for discrete data, 2024.

---

### Author Response · Authors · 2024-11-25
**Summary of Paper Revision**

We thank all reviewers for their constructive feedback and have responded to each reviewer individually. We have also uploaded a **Paper Revision**, including additional results and illustrations. **All changes in revision are marked in blue**.

**For Reviewer u3V7**:

- Section 5 (in Lines 493-495): We explained that our method of RADD is not directly applicable to other discrete diffusions.
- Section 5 (in Lines 496-498), Appendix F: We added a detailed comparison with Chen et al. 2024.

**For Reviewer rVfV**:
- Section 4.3: We explained the ablation experiments on the time-condition and reasons for disparity loss on equivalent loss.
- Section 4.2: We emphasized that SEDD surpasses AR sampling in terms of sampling speed.
- Appendix J.4: We emphasized the comparison between the Tweedie tau-leaping sampler and the AO-ARM sampler.
- Section 4.1 and Appendix  H.2: We report the training tokens and iterations in detail.


**For Reviewer fLbR**:
- Section 4.2 and Appendix J.4 and Table 5: We validate the acceleration performance of our cache strategy under varying batch size.
- Section 5 (in Lines 496-498), Appendix F: We added a detailed comparison with Chen et al. 2024.
- Section 3.3 and Appendix H: We added a discussion about related work on Austin et al., 2021 and Hoogeboom et al., 2021.
- Section 5 (in Lines 495-496): We added a discussion on Campbell et al., 2024.
- Appendix J.4, Table 3, and Table 4: We provided generative perplexity results under fp32 precision aligned with SEDD.
- Section 3.1: We emphasize more on theoretical contributions of our reparameterization framework.

**For Reviewer rqCn**:
- Section 5 (in Lines 518-520): We emphasized the unique contribution of RADD.

**For Reviewer 4Udp**:
- Section 5 (in Lines 518-520): We emphasized the unique contribution of RADD.
- Section 3.3 and Appendix H: We added a discussion about related work on  Hoogeboom et al., 2021.
- Appendix G and Appendix I: We include a section in the appendix discussing the parametrization of AO-ARMs and provide both the training and sampling algorithm of AO-ARMs.

---

### Meta-Review · Area_Chair_LtN7 · 2024-12-20

**Metareview:**

This paper presents a reparameterization of absorbing discrete diffusion called RADD, which simplifies the modeling of the concrete score as a time-independent conditional probability of clean data. This reparameterization eliminates the need for time conditioning in the network, enabling a simpler architecture and faster sampling via caching unchanged tokens. Additionally, the paper establishes a theoretical connection between absorbing discrete diffusion models and any-order autoregressive models (AO-ARMs) Empirical results demonstrate state-of-the-art performance on zero-shot language modeling tasks and reductions in the number of function evaluations.

**Additional Comments On Reviewer Discussion:**

While some reviewers noted overlap with concurrent work, the authors provided comparisons that satisfied the reviewers. Concerns about practical contributions, such as the efficiency of caching in minibatch settings and the impact of removing time conditioning, were addressed through additional experiments and explanations. All reviewers converged on recommending acceptance despite some limitations in novelty.

---

### Decision · Program_Chairs · 2025-01-22

Accept (Poster)